# Agent-X: Evaluating Deep Multimodal Reasoning in Vision-Centric Agentic Tasks

**Tajamul Ashraf**[1†]**, Amal Saqib**[1†]**, Hanan Gani**[1]**, Muhra AlMahri**[1]**, Yuhao Li**[1]**,
Noor Ahsan**[1]**, Umair Nawaz**[1]**, Jean Lahoud**[1]**, Hisham Cholakkal**[1]**, Mubarak Shah**[2]**,
Philip Torr**[3]**, Fahad Shahbaz Khan**[1]**, Rao Muhammad Anwer**[1]**, Salman Khan**[1]

[1]Mohamed bin Zayed University of Artificial Intelligence (MBZUAI), United Arab Emirates
[2]University of Central Florida, USA
[3]University of Oxford, United Kingdom
[†]Equal contribution

## Abstract

Deep reasoning is fundamental for solving complex tasks, especially in vision-centric scenarios that demand sequential, multimodal understanding. However, existing benchmarks typically evaluate agents with fully synthetic, single-turn queries, limited visual modalities, and lack a framework to assess reasoning quality over multiple steps as required in real-world settings. To address this, we introduce Agent-X, a large-scale benchmark for evaluating vision-centric agents' multi-step and deep reasoning capabilities in real-world, multimodal settings. Agent-X features 828 agentic tasks with authentic visual contexts, including images, multi-image comparisons, videos, and instructional text. These tasks span six major agentic environments: general visual reasoning, web browsing, security and surveillance, autonomous driving, sports, and math reasoning. Our benchmark requires agents to integrate tool use with explicit, stepwise decision-making in these diverse settings. In addition, we propose a fine-grained, step-level evaluation framework that assesses the correctness and logical coherence of each reasoning step and the effectiveness of tool usage throughout the task. Our results reveal that even the best-performing models, including `GPT`, `Gemini`, and `Qwen` families, struggle to solve multi-step vision tasks, achieving less than **50% full-chain success**. These findings highlight key bottlenecks in current `LMM` reasoning and tool-use capabilities and identify future research directions in vision-centric agentic reasoning models. Our data[1] and code[2] are available.

## 1 Introduction

Agentic frameworks enable AI systems to interact with their environment by perceiving inputs, invoking tools, and executing actions. While perception and tool use are core components, solving complex tasks effectively requires reasoning, i.e., the ability to draw logical inferences, make decisions, and adapt over time based on multimodal inputs such as text, images, video, and temporal context (Kumar et al., 2025; Yang et al., 2023; Yao et al., 2023; Thawakar et al., 2025). Deep reasoning in agents helps effectively plan, execute, and adapt across evolving scenarios (Liu et al., 2024b; Nathani et al., 2025). Recent works have explored integrating reasoning and tool use within large multimodal models (`LMMs`)(Achiam et al., 2023; Bai et al., 2023b; Bi et al., 2024; Team et al., 2023; Jiang et al., 2023; Grattafiori et al., 2024), where `LMMs` act as controllers for planning, and callable tools handle external actions(Chase, 2022; Gravitas, 2023; OpenAI, 2023). This architecture enables agents to combine perception, visual understanding, symbolic reasoning, and generative capabilities, significantly improving task performance in complex environments.

---

[1]https://huggingface.co/datasets/Tajamul21/Agent-X
[2]https://github.com/mbzuai-oryx/Agent-X

Table 1: Comparison of Agentic Benchmarks. Columns show key dimensions including scale, realism, modality, reasoning depth, tool interaction, and annotation quality. Our benchmark Agent-X uniquely supports all criteria with 828 diverse, manually verified agentic tasks.

| Benchmark | Agentic Tasks | Multimodal Tools | Real-world Queries | Multimodal Inputs | Deep Reasoning | Executable Tools | Hybrid Annotation |
|---|---|---|---|---|---|---|---|
| APIBench (Patil et al., 2024) | ✗ | | | | | | ✓ |
| APIBank (Li et al., 2023) | ✗ | | | | | ✓ | |
| ToolBench (Qin et al., 2024) | ✗ | | | | | ✓ | |
| MLAgentBench (Huang et al., 2024) | - | | | | | | |
| GAIA (Mialon et al., 2023) | 466 | ✓ | ✓ | ✓ | | ✓ | |
| SWE-Bench (Jimenez et al., 2024) | ✗ | | | | | ✓ | |
| GTA (Wang et al., 2024) | 229 | ✓ | ✓ | ✓ | | ✓ | |
| MLE-Bench (Chan et al., 2024) | ✗ | | | ✓ | | | |
| m&m's (Liu et al., 2024c) | - | ✓ | | ✓ | | ✓ | ✓ |
| RE-Bench (Wijk et al., 2024) | 7 | | ✓ | | | ✓ | |
| ScienceAgent (Chen et al., 2024b) | 102 | | ✓ | | | | |
| MLGym (Nathani et al., 2025) | 13 | | ✓ | ✓ | ✓ | | |
| Agent-X (Ours) | 828 | ✓ | ✓ | ✓ | ✓ | ✓ | ✓ |

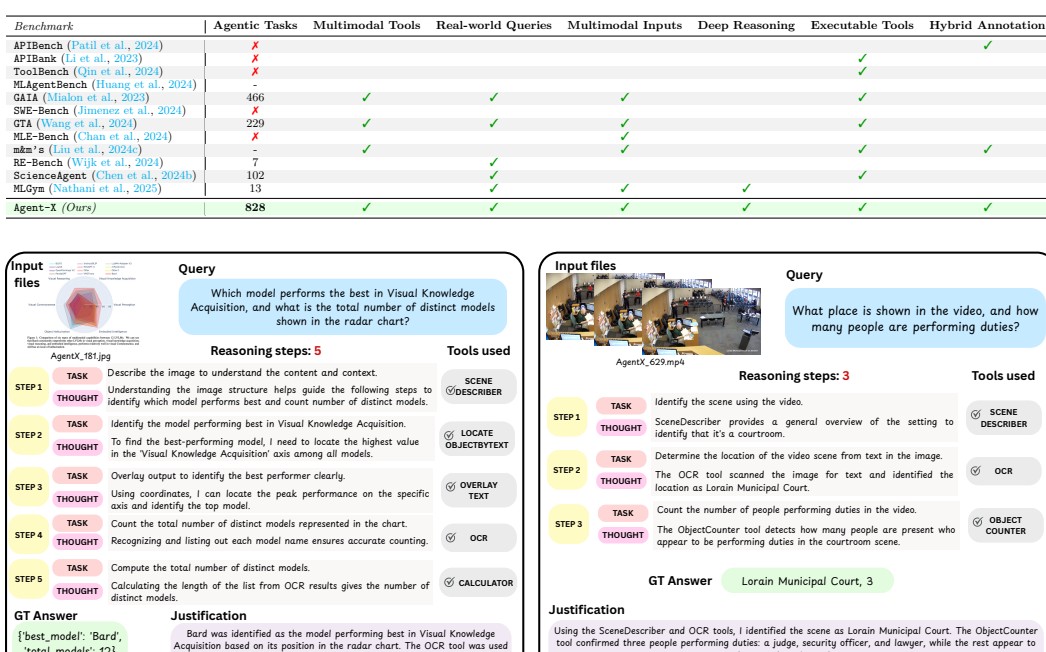

Figure 1: Agent-X Snapshot: Example tasks from our benchmark illustrating multimodal queries that require step-by-step reasoning, tool use, and visual understanding across images and video. Each task includes structured thoughts, tool invocations, and a ground-truth answer with justification. The detailed annotations in Agent-X enable thorough evaluation of existing agentic pipelines.

Existing benchmarks for agentic systems have primarily focused on text-based interactions, with limited support for multimodal inputs such as images, videos, and multi-image comparisons (Mialon et al., 2023; Liu et al., 2024b; Wang et al., 2024). While some efforts have extended to multimodal tasks, they are often restricted to static images, synthetic environments, or narrowly scoped domains, offering limited insight into an agent's ability to perform complex, tool-driven reasoning. A key limitation is inadequate evaluation of deep reasoning[3]: current benchmarks either neglect this aspect entirely or lack principled metrics for assessing multi-step logical coherence (Wang et al., 2024; Nathani et al., 2025; Liu et al., 2024b; Team, 2023). Furthermore, most existing benchmarks rely on either fully synthetic tasks or manual annotations, which lack scalability and fail to capture the depth of reasoning or the complexity of video-based multimodal interactions, falling short of modeling real-world agentic tasks (Han et al., 2024; Thawakar et al., 2025), as summarized in Table 1. As agents rapidly evolve in their capabilities, existing benchmarks struggle to keep pace, underscoring the need for more comprehensive, reasoning-centric evaluations to track planning, adaptation, and tool use in authentic, multimodal interactive scenarios (Kiela et al., 2023).

Agent-X is the first benchmark that combines large-scale, real-world multimodal inputs (images, video, text) with tool-augmented step-wise reasoning evaluation across six environments, providing both breadth and depth missing in GAIA (Mialon et al., 2023), GTA (Wang et al., 2024), and other recent datasets. To address these limitations, we introduce Agent-X, a large-scale evaluation benchmark for *vision-centric agents* that emphasizes two core principles: multimodal reasoning and vision-first evaluation. Agent-X rigorously tests agents' ability to process complex visual and textual inputs, execute tool-augmented plans, and perform deep reasoning across real-world tasks. A primary feature of our benchmark is its **deep reasoning assessment**, i.e., judging the ability to perform coherent, multi-step problem-solving, logical

---

[3]In our context, *Deep reasoning* refers to coherent, multi-step inference grounded in multimodal context.

Table 2: Task comparison of Agent-X with existing benchmarks. Unlike prior benchmarks, the queries in Agent-X avoid explicit tool references and direct instructions, thus encouraging agents to reason and act independently. Blue indicate explicit task guidance; Red highlights denote explicit tool invocation in prior benchmarks.

| Method | Queries | Related Tools |
|---|---|---|
| **ToolBench** | I'm writing a blog post... *first retrieve* available figlet styles and *then generate* ASCII art for the strings using the style. | `figlet, list figlet styles` |
| **APIBench** | I am a engineer at Uber and... *Write a python program* in 1 to 2 lines to *call API* in TorchHub. | `ObjectDetection` |
| **GTA** | How much should I pay for the soda in the picture *according to the price on the menu?* | `ImageDescription, CountGivenObject, OCR` |
| **m&m's** | I need an illustration for my children's book. I've imagined a scene where... *After* we have the image, we also need to *identify all the objects*, *then add labels* to them. | `ImageGeneration, ObjectDetection, Tagging` |
| **Agent-X (Ours)** | What store is the scene in the video from and what does the person dressing corresponds to in normal circumstances? | `SceneDescriber, OCR, RegionDescriber, WebSearch` |

inference, and adaptive planning. To move beyond surface-level evaluation (focused on the correctness of the final result), we introduce fine-grained metrics that capture both intermediate reasoning steps and overall task coherence. This approach helps distinguish genuine logical progression from inconsistencies and confabulations (plausible-sounding but disconnected steps) in reasoning chains, as illustrated in Figure 1. Agent-X spans six **diverse multimodal environments**, incorporating images, videos, and spatiotemporal contexts to evaluate agents in rich, realistic settings that demand generalization beyond text. Our **task pipelines** are derived from authentic user queries without explicit step instructions, simulating naturalistic agent interaction as highlighted in Table 2. These tasks are paired with executable toolchains via a semi-automated pipeline that integrates a broad set of real-world tools, enabling scalable tool-augmented decision-making. In Agent-X, these components create a comprehensive framework for evaluating agentic performance in perception, reasoning, and action. Our main contributions are as follows:

- We propose a large-scale benchmark (Agent-X) for evaluating vision-centric agents, emphasizing *deep reasoning capabilities* across diverse multimodal environments.

- We introduce fine-grained metrics to evaluate `LMM` based agentic frameworks, focusing on tool use, deep reasoning and planning capabilities in real-world tasks.

- We evaluate 10 mainstream `LMMs` on the Agent-X benchmark, uncovering key limitations in real-world multimodal agents and providing actionable insights to guide future research on agentic development.

## 2 Related Work

**Large Multimodal Agents:** The rapid progress in `LMMs` has catalyzed the development of agentic frameworks capable of autonomous planning, tool use, and decision-making. A growing body of research has explored the extension of `LMMs` beyond conventional text generation by integrating them with external tools such as `APIs` (Zhang et al., 2025b), document processing (Musumeci et al., 2024), operating systems (Mei et al., 2024), and web interfaces (Song et al., 2024), allowing them to interact meaningfully with their environments. This evolution has led to the emergence of tool-augmented agents such as `Avatar` (Wu et al., 2024), `LangChain` (Chase, 2022), `AutoGPT` (Gravitas, 2023), and `BabyAGI` (Nakajima, 2023), which provide frameworks for textual reasoning and execution of external actions. In the context of web browsing, systems such as `WebShop` (Yao et al., 2022), `WebGPT` (Nakano et al., 2021), and `WebCPM` (Qin et al., 2023) enhance `LMMs` with capabilities for browsing, searching, and information retrieval. Beyond textual modalities, integrations such as `RestGPT` (Song et al., 2023) and `AppAgent` (Zhang et al., 2025c) enable interaction with `REST APIs` and emulate touchscreen operations, respectively. In the multimodal domains, *vision-centric agents* like `MLLMTool` (Wang et al., 2025) and `LLaVA` (Liu et al., 2024a) equip `LMMs` with the ability to reason about visual content through pre-trained vision models. Nevertheless, there

is no standardized evaluation protocol to rigorously assess the reasoning capabilities of these agents, particularly in tasks that require tool invocation and decision-making.

**Multimodal Agentic Benchmarks:** A broad range of recent benchmarks have emerged to evaluate the performance of agentic frameworks. Recent efforts like `ToolBench` (Qin et al., 2024) introduce `REST APIs` and evaluate tool use with metrics such as pass rate and win rate, while `APIBench` (Patil et al., 2024) assesses accuracy for `APIs`. Complementing these, `APIBank` (Li et al., 2023) presents a diverse suite of commonly used `APIs`, such as search engines and hotel reservations, offering a comprehensive framework for evaluating the planning, retrieval, and execution skills of agents based on `LMM`. Benchmarks such as `ToolQA` (Zhuang et al., 2023), `Gentopia` (Xu et al., 2023), `Gorilla` (Patil et al., 2024), and `AgentBench` (Liu et al., 2024b) often repurpose standard data sets or follow restricted evaluation protocols, which limit their utility in assessing open-ended and generalizable tool use capabilities. To address this, `GAIA` (Mialon et al., 2023) poses conceptually challenging questions for human-like understanding, `GTA` (Wang et al., 2024) evaluates agents with executable toolchains in real-world settings, and `OSWorld` (Xie et al., 2024) offers multi-step tasks based on real user behavior. `MLGym` (Nathani et al., 2025) further adds complex multimodal tasks requiring deep reasoning and tool use. As shown in Table 1, despite promising advances, current evaluations face two key limitations: (1) most benchmarks focus on final answer accuracy, overlooking interpretability, which is crucial for agentic tasks involving reasoning and tool use, and (2) the absence of standardized and interpretable evaluation protocols hinders meaningful comparison across methods.

**Reasoning in Large Multimodal Models:** Multiple methods have been introduced to enhance the reasoning capabilities of `LMMs`, enabling models to handle complex, multi-step problems through explicit reasoning processes (Amizadeh et al., 2020). Following the rise of reasoning techniques, benchmarks have emerged to evaluate reasoning and chain-of-thought in models. Early efforts like `CLEVR` (Johnson et al., 2017) tested basic visual logic, while `StrategyQA` (Geva et al., 2021) introduced multi-hop reasoning. `ScienceQA` (Saikh et al., 2022) added multimodal scientific tasks, `MathVista` (Lu et al., 2023) unified math reasoning datasets, and `ShareGPT-4o` (Zhang et al., 2024) offered extensive chain-of-thought samples to enhance intermediate reasoning. The `m&m's` (Liu et al., 2024c) benchmark focuses on multi-step, multimodal reasoning and evaluates various planning strategies. However, many benchmarks, including `m&m's`, rely on AI-generated queries with predefined tool sequences, limiting their realism. `LLaMA-V` (Thawakar et al., 2025) introduced a benchmark focused on assessing tasks requiring multiple reasoning steps. While these reasoning-focused benchmarks have been introduced, they do not address the agentic framework or evaluate tool usage and the associated reasoning within vision-centric agentic tasks.

## 3 AGENT-X BENCHMARK

The Agent-X benchmark evaluates the capacity of vision-centric agents to perform complex reasoning tasks that necessitate proficient tool usage across diverse, real-world scenarios. It accommodates a broad spectrum of input modalities, including single-image analysis, multi-image comparisons, video interpretation, and multimodal image-text interactions. Agent-X presents agents with authentic visual data and requires them to use multi-step reasoning processes. The construction of Agent-X follows a semi-automated pipeline (illustrated in Figure 2), where an initial set of candidate queries is generated through large multimodal models (`LMM`) and subsequently refined and validated by human experts.

### 3.1 BENCHMARK DESIGN

In our proposed Agent-X benchmark, each task is formally defined as a structured tuple: $\mathcal{S}_i = (\mathcal{V}_i, \mathcal{Q}_i, \mathcal{T}_i, \mathcal{R}_i, \mathcal{A}_i, \mathcal{J}_i)$ where: $\mathcal{V}_i$ denotes the multimodal context, which may comprise a single image, text, multiple images, or video frames. These visual inputs are carefully selected to reflect diverse real-world scenarios. $\mathcal{Q}_i$ is the query associated with $\mathcal{V}_i$ that necessitates multi-step deep reasoning and the strategic use of external tools. $\mathcal{T}_i \subseteq \mathcal{T}_c = \{t_k\}_{k=1}^{N}$ is the subset of tools employed to resolve the query, where $\mathcal{T}_c$ is a predefined library of $N$ tools encompassing perception, visual operation, math, and artistic tasks. Further details are in

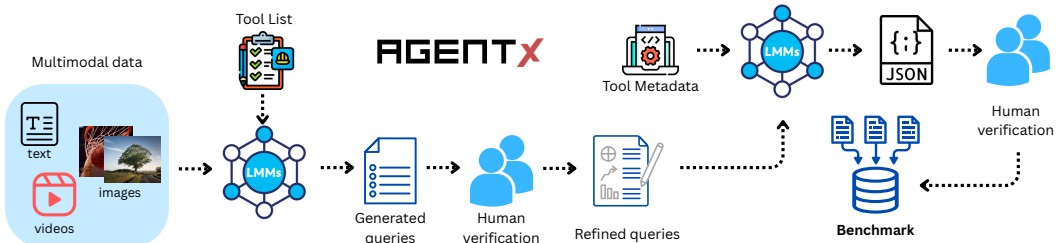

Figure 2: Overview of the Agent-X benchmark construction pipeline. Starting from multimodal data and a predefined toolset, an `LMM` generates initial queries, which are refined by annotators for realism and correctness. The `LMM` then produces step-by-step reasoning, which is refined to create a high-quality tool-augmented reasoning trace.

Appendix §B. $\mathcal{R}_i = \{(t_j, a_j, r_j)\}_{j=1}^m$ is the deep reasoning trace, capturing the sequence of interactions during problem-solving. Here, each step is defined as a triplet $(t_j, a_j, r_j)$, where $t_j$ is the tool used, $a_j$ the input arguments, and $r_j$ the resultant output. $\mathcal{A}_i$ is the final answer derived from the reasoning process, which may take various forms, including textual explanations, numeric values, or generated content. Finally, the justification, $\mathcal{J}_i$ explains the reasoning trace in natural language to support $\mathcal{A}_i$ and enhance interpretability.

In our framework, $\mathcal{T}_c$ comprises a carefully curated collection of 14 tools distributed across diverse environments. These 14 tools capture the breadth of capabilities required for real-world, vision-centric reasoning. A comprehensive list of these tools is presented in Appendix §B. The queries $\mathcal{Q}$ are classified into three distinct categories: *factual*, *interpretive*, and *generative*. Illustrative examples of these query types are provided in Appendix §I. In the case of a factual query, the answer is a specific, uniquely determined value, such as a number or a phrase. For an interpretive query, the final answer consists of descriptive text. While the answer is not unique, it conveys a general concept or idea, with the final response comprising a reference answer. For a generative query, we do not directly utilize the generated output since the `LMMs` are descriptive models; they only provide textual descriptions rather than generating visual content like images/videos. In these instances, $\mathcal{A}_i = \emptyset$.

## 3.2 TASK PIPELINE

To design the task $(\mathcal{V}_i, \mathcal{Q}_i, \mathcal{T}_c)$, we employ a semi-automated pipeline (Figure 2) guided by three fundamental principles. *First*, each task $(\mathcal{V}_i, \mathcal{Q}_i)$ must be solvable using a subset of tools $\mathcal{T} \subseteq \mathcal{T}_c$, ensuring that the problem can be addressed through the functional capabilities of the specified tools. *Second*, the query $\mathcal{Q}_i$ should not explicitly list the required tools or the sequence of steps, forcing the model to plan and reason through the task independently. *Finally*, the tasks are grounded in realistic, meaningful scenarios that mirror complex and real-world challenges.

Our query construction process begins with generating candidate queries using an `LMM`, which is provided with the visual input $\mathcal{V}_i$ and the available complete toolset $\mathcal{T}_c$. The `LMM` generates initial queries that human annotators then refine and validate for clarity, coherence, and realism. In the next step, `LMM` is re-engaged with the refined query to produce a detailed reasoning trace $\mathcal{R}_i$, capturing the sequence of tool calls (input/output), intermediate decisions, and final answer $\mathcal{A}$. Human reviewers further ensure that the trace is logically consistent and aligned with the query, while the final answer, $\mathcal{A}_i$ and its justification, $\mathcal{J}_i$ are validated for correctness and comprehensibility. The prompts provided to the LMM for query generation and reasoning trace construction are included in Appendix §J.

## 3.3 HUMAN-GUIDED REFINEMENT

To establish a robust Agent-X benchmark, visual input $\mathcal{V}$ consists of images and videos sourced from publicly available datasets. To maintain a diverse and realistic dataset, queries

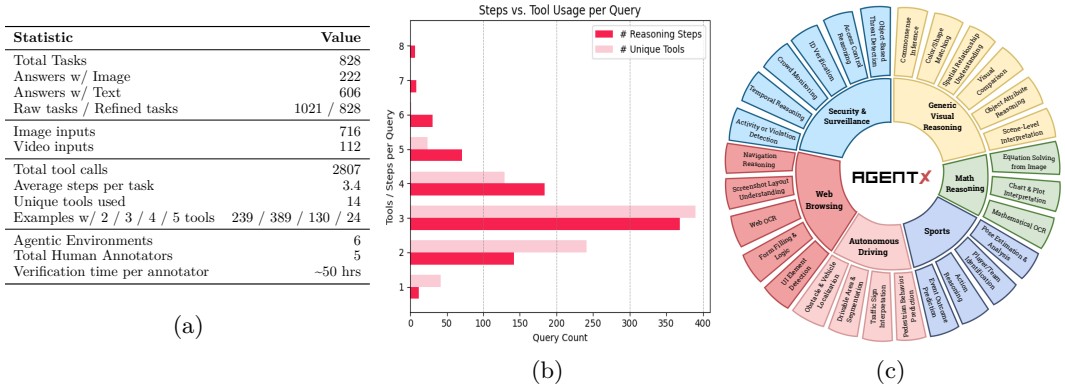

| Statistic | Value |
|---|---|
| Total Tasks | 828 |
| Answers w/ Image | 222 |
| Answers w/ Text | 606 |
| Raw tasks / Refined tasks | 1021 / 828 |
| Image inputs | 716 |
| Video inputs | 112 |
| Total tool calls | 2807 |
| Average steps per task | 3.4 |
| Unique tools used | 14 |
| Examples w/ 2 / 3 / 4 / 5 tools | 239 / 389 / 130 / 24 |
| Agentic Environments | 6 |
| Total Human Annotators | 5 |
| Verification time per annotator | ~50 hrs |

(a)  (b)  (c)

Figure 3: Overview of the Agent-X benchmark. **(a)** Key data statistics. **(b)** Overall frequency of tool usage and number of steps. **(c)** Distribution of tasks across six environments.

$\mathcal{Q}$ are crafted to avoid direct references to specific tools. For instance, a query such as *"Count the number of objects present in this image"* is avoided because it directly hints at using the `ObjectCounter` tool. The reasoning steps $\mathcal{R}$ capture the logical flow for each query, with an average of three steps. The answer $\mathcal{A}$ provides a final response, while $\mathcal{J}$ offers supporting evidence such as *URLs* or *screenshots*, especially for queries involving web search tools. This structure ensures that each query is well-defined and verifiable.

**Task and Reasoning Construction.** Our task construction process begins with an automated generation phase, where an `LMM` produces three candidate queries for each of 1,021 initial visual inputs (images or video frames). Annotators then select the best query from each set, yielding 1,021 raw queries. Detailed annotation guidelines, including query construction rules and reasoning/answer refinement protocols, are provided in Appendix §H. Each candidate is carefully reviewed for clarity, realism, and suitability to ensure that it cannot be answered directly from the input alone, but instead necessitates meaningful tool use, is expandable to diverse scenarios, and supports multi-step reasoning. Through this refinement process, queries failing to meet these criteria are discarded, resulting in a final pool of **828 validated tasks**. Each task is paired with one or more unique visual inputs, with no visual input reused across tasks. For web search queries, we additionally enforce two safeguards: (1) the query must be answerable using real-time search rather than relying solely on static knowledge, and (2) it should reference credible sources, such as *"What were the global average temperatures reported in April 2024?"*, rather than overly general prompts like *"What is climate change?"*

In the second stage, we construct the toolchain for each validated task. Each query $\mathcal{Q}$ and its corresponding visual input $\mathcal{V}$ are provided to an `LMM` along with the toolset $\mathcal{T}_c$, which generates an initial reasoning trace, final answer $\mathcal{A}$, and justification $\mathcal{J}$. The reasoning trace specifies the sequence of tool invocations, input arguments, and intermediate outputs, formatted in a JSON-style dialogue. Human annotators rigorously review and refine these `LMM`-generated traces to ensure logical consistency, correctness of tool use, and factual alignment of the final answer $\mathcal{A}$ with the reasoning process. Annotators correct errors, replace inappropriate tool choices, and filter out tasks that cannot be solved reliably. Appendix §L illustrates examples of LMM-generated queries and reasoning traces before and after human refinement.

## 3.4 Dataset Composition and Statistics

The final Agent-X benchmark comprises a diverse set of queries covering multiple categories, including factual, interpretive, and generative tasks. The dataset is constructed using 14 executable tools, spanning six diverse environments, *e.g.*, autonomous driving (Cordts et al., 2016; Wood, 2020; Yu et al., 2020), security and surveillance (Aktı et al., 2019; Sultani et al., 2018; Lu et al., 2013; Naphade et al., 2023), math reasoning (Lu et al., 2023; Ling et al., 2017; Cobbe et al., 2021), web navigation (Sunkara et al., 2022; Zhou et al., 2023), sports

Table 3: **Evaluation Metrics.** This table outlines the full suite of metrics used in Agent-X benchmark, organized by Step-by-Step, Deep Reasoning, and Outcome modes.

| | Metric (Symbol) | Description |
|---|---|---|
| **Step-by-Step Mode** | Grounding Score ( $\mathbf{G_s}$ ) | *Correct reference to objects, regions, or attributes in the input.* |
| | Tool Precision ( $\mathbf{T_P}$ ) | *Accuracy of selecting the correct tool at each reasoning step.* |
| | Tool Accuracy ( $\mathbf{T_a}$ ) | *Correct use of tools with appropriate inputs and outputs.* |
| **Deep Reasoning Mode** | Faithfulness ( $\mathbf{F_{acc}}$ ) | *Logical consistency across the reasoning process.* |
| | Context Score ( $\mathbf{C_s}$ ) | *Effective use of multimodal and commonsense context.* |
| | Factual Precision ( $\mathbf{F_P}$ ) | *Correctness of factual information without hallucination.* |
| | Semantic Accuracy ( $\mathbf{S_{acc}}$ ) | *Coverage of all semantically necessary elements.* |
| **Outcome Mode** | Goal Accuracy ( $\mathbf{G_{acc}}$ ) | *Final answer accuracy for factual and interpretive queries.* |
| | Goal Accuracy w/ImgGen ( $\mathbf{G_{acc}^*}$ ) | *Final answer accuracy for generative queries.* |
| | Toolset Accuracy ( $\mathbf{T_{acc}}$ ) | *F1 score for overall correct tool selection and use.* |

(Karpathy et al., 2014; Wu et al., 2022), and generic visual reasoning (Lin et al., 2014; Wang et al., 2024). Full details of the source datasets are provided in Appendix §C.1.

The annotation effort amounted to roughly 50 hours per annotator, reflecting the full human refinement and validation process. Approximately 800K API tokens were used during benchmark construction. These categories are designed to support a wide range of multimodal reasoning tasks. The total number of tasks, query analysis, tool analysis, and annotation labor hours are detailed in Figure 3. Most queries involve the use of two to four executable tools. This diverse setup ensures that Agent-X can comprehensively evaluate model performance across various tool-augmented reasoning tasks.

## 4  AGENT-X EVALUATION

### 4.1  EXPERIMENTAL SETUP

**Evaluation Modes:** We evaluate the models on the Agent-X benchmark using three distinct evaluation modes: **1. Step-by-Step:** This mode evaluates the agent's ability to reason through individual steps within a reasoning trace ($\mathcal{R}$). It measures how accurately the agent understands and reproduces structured tool-use sequences grounded in visual inputs. **2. Deep Reasoning:** This mode assesses the agent's capacity to generate coherent, logically consistent multi-step reasoning traces. It focuses on integrating visual and textual inputs to produce contextually relevant, semantically complete, and factually accurate reasoning. **3. Outcome:** This mode measures the agent's overall task-solving ability by evaluating the correctness of the final answer and the accuracy of tool usage.

**Metrics:** We design a suite of fine-grained metrics (Table 3) to evaluate agentic reasoning across all three modes comprehensively. These metrics capture key aspects of the agent pipeline. For `Goal Accuracy`, we exclude image generation queries and focus on factual and interpretive queries, using exact matching against the gold answer and descriptive matching against labeled responses using GPT-4o (Hurst et al., 2024) respectively. For image generation, we define `Goal Accuracy/ImgGen`, which assesses the correctness of predicted input parameters, assuming accurate inputs yield suitable images. Detailed metrics computation and implementation are provided in Appendix §C.

### 4.2  BENCHMARK RESULTS

Table 4 presents the core evaluation results of Agent-X across three modes: STEP-BY-STEP, DEEP REASONING, and OUTCOME. We evaluate a mix of closed-source models (e.g., GPT (Achiam et al., 2023; OpenAI, 2025) and Gemini (Team et al., 2023; 2024a)) and strong open-source counterparts (e.g., InternVL (Chen et al., 2024a; Zhu et al., 2025) and Qwen (Bai et al., 2023b;c)). While closed-source models lead overall, open-source models show competitive behavior in select metrics, offering key insights across agentic frameworks, which we summarize below. Beyond these core comparisons, we further extend the benchmark with additional open-source vision LMMs, as detailed in Appendix §D. We additionally report results using two alternative evaluation judges, Qwen and human annotators, in Appendix §E. The evaluation prompts provided to GPT-4o, Qwen, and human annotators

Table 4: **Overall results on Agent-X.** We report performance across three evaluation modes: STEP-BY-STEP, DEEP REASONING, and OUTCOME. Metrics include: $G_s$ (Grounding Score), $T_p$ (Tool Precision), $T_{acc}$ (Tool Accuracy), $F_{acc}$ (Faithfulness Accuracy), $C_s$ (Context Score), $F_p$ (Factual Precision), $S_{acc}$ (Semantic Accuracy), $G_{acc}$ (Goal Accuracy), $G_a^*$ (Goal Accuracy/ImgGen), and $T_{acc}^s$ (Toolset Accuracy). *The best results are highlighted in* **bold**, *and second-best are* underlined.

| Model | Step-by-Step | | | Deep Reasoning | | | | Outcome | | |
|---|---|---|---|---|---|---|---|---|---|---|
| | $G_s$ | $T_p$ | $T_{acc}$ | $F_{acc}$ | $C_s$ | $F_p$ | $S_{acc}$ | $G_{acc}$ | $G_a^*$ | $T_{acc}^s$ |
| *Open-source** | | | | | | | | | | |
| Phi-4-VL-Instruct | 0.13 | 0.21 | 0.24 | 0.61 | 0.19 | 0.47 | 0.40 | 0.11 | 0.26 | 0.42 |
| InternVL2.5-8B | 0.45 | 0.31 | 0.47 | 0.68 | 0.47 | 0.52 | 0.60 | 0.28 | 0.55 | 0.58 |
| Gemma-3-4B | 0.26 | 0.30 | 0.78 | 0.61 | 0.54 | 0.38 | 0.54 | 0.27 | 0.67 | 0.60 |
| InternVL3-8B | 0.46 | 0.34 | 0.54 | 0.68 | 0.45 | 0.70 | 0.40 | 0.20 | 0.59 | 0.62 |
| VideoLLaMA3-7B | 0.45 | 0.28 | 0.46 | 0.65 | 0.46 | 0.62 | 0.54 | 0.28 | 0.54 | 0.54 |
| Qwen2.5-VL-7B | 0.54 | 0.43 | 0.63 | 0.75 | **0.57** | 0.56 | 0.67 | 0.36 | 0.65 | 0.67 |
| *Closed-source** | | | | | | | | | | |
| Gemini-1.5-Pro | 0.43 | 0.23 | 0.84 | 0.62 | 0.45 | 0.53 | 0.62 | 0.04 | 0.56 | 0.48 |
| Gemini-2.5-Pro | 0.40 | 0.36 | 0.81 | 0.72 | 0.48 | 0.64 | 0.73 | 0.40 | 0.56 | 0.62 |
| GPT-4o | **0.60** | **0.47** | 0.72 | **0.81** | **0.57** | **0.79** | 0.59 | 0.37 | **0.70** | **0.68** |
| OpenAI-o4-mini | 0.42 | 0.32 | **0.89** | 0.71 | 0.51 | 0.60 | **0.80** | **0.45** | 0.67 | 0.63 |

*Results for the additional models are in Appendix §D and cross-judge consistency study in Appendix §E.*

are included in Appendix §K. To mitigate bias, all predictions are cross-checked by multiple graders (GPT-4o, Qwen-14B, and humans), with consistent model rankings across settings. Human scores correlate strongly with automatic grading, and residual discrepancies are uniformly distributed, confirming no systematic favoritism. Furthermore, evaluation metrics are explicitly bias-aware (decoupling syntax from semantics, normalizing tool arguments), and task seeds were rewritten under strict QA to avoid leakage. Together, these safeguards ensure fairness, robustness, and reproducibility of the results.

**Insight 1: Real-world tool-use tasks remain challenging for current LMM agents.** Despite multimodal capabilities, no model exceeds 50% in `Goal Accuracy` ($G_{acc}$). o4-mini, the best performer, achieves only 45%, while most open-source models are below 30%, highlighting the difficulty of tool use and final answer consistency in real-world settings.

**Insight 2: Strong reasoning capabilities contribute to higher task success rates.** Models that maintain consistently strong scores across reasoning metrics are likelier to perform well on final task outcomes. GPT-4o, for instance, achieves high scores in $F_{acc} = 0.81$, $F_p = 0.79$, and $S_{acc} = 0.59$, which correlates with a relatively high $G_{acc} = 0.37$ and $T_{acc}^s = 0.68$. Likewise, Qwen2.5, with solid reasoning scores ($C_s = 0.57$, $S_{acc} = 0.67$), attains $G_{acc} = 0.36$, outperforming many open-source peers. These results indicate that deep reasoning and structured execution enhance task success in vision-centric agents.

**Insight 3: Tool invocation and argument prediction remain core bottlenecks.** Tool-related metrics show the widest variation, indicating difficulty in accurate and well-formatted tool use. While the GPT family achieves strong reasoning capabilities, its `Toolset Accuracy` is low. Open-source models like Qwen2.5-VL-7B show similar inconsistencies as highlighted in Table 4. These results support the observation that argument formatting and tool chaining are the weakest links, disrupting overall pipeline reliability and highlighting a need for more precise execution control.

## 4.3 ERROR ANALYSIS

**Qualitative Analysis:** We conduct a qualitative error analysis (Figure 4) to uncover key limitations in current vision-centric agentic models. Our findings reveal four prevalent issues: (1) agents often exhibit shallow reasoning by skipping frame-wise or multi-step analysis, especially in video-based tasks; (2) models frequently misuse or hallucinate tools not defined in the metadata, indicating poor tool grounding; (3) many responses violate expected output formats, producing incomplete or non-JSON-compliant traces that hinder evaluation; and (4)

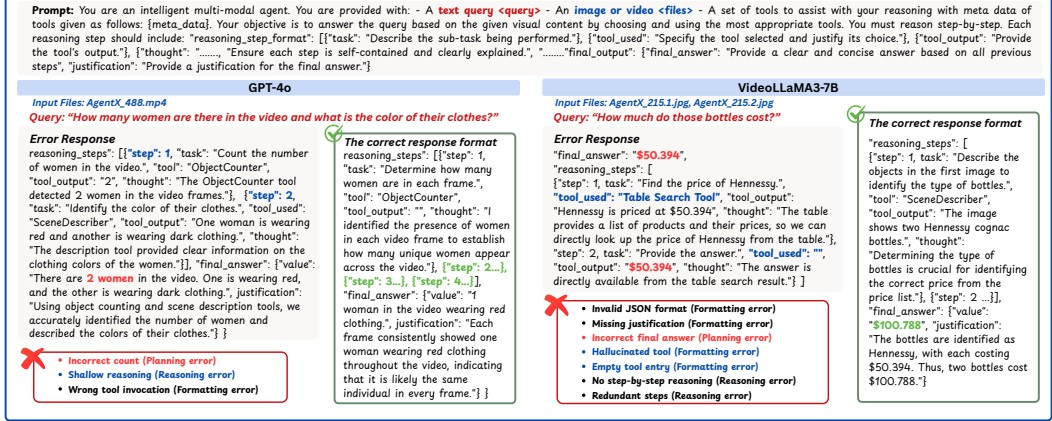

Figure 4: Qualitative comparison of `GPT-4o` and `VideoLLaMA3-7B` on Agent-X visual reasoning tasks. `GPT-4o` hallucinates tool use and gives incorrect justifications; `VideoLLaMA3-7B` lacks temporal reasoning and frame alignment. More comparisons in Appendix §F.2.

Table 5: **Error Breakdown Across Models.** Common planning, formatting, and reasoning errors on Agent-X across GPT-4o, Gemini-1.5-Pro, and InternVL3-8B. Formatting errors are counted alongside planning and reasoning errors. Extended details in Appendix §F.1.

| Error Type | GPT-4o | Gemini-1.5-Pro | InternVL3-8B |
|---|---|---|---|
| *Planning Errors:* | | | |
| No action, no response. | 157 (17.6%) | 3 (0.2%) | 172 (12.8%) |
| No action, the whole response is a model thought. | 0 | 2 (0.1%) | 0 |
| *Formatting Errors:* | | | |
| Invalid JSON format in argument specification. | 235 (26.4%) | 755 (44.5%) | 454 (33.8%) |
| Multiple tool calls in a single step. | 118 (13.2%) | 172 (10.1%) | 126 (9.4%) |
| Final answer generation without adhering to the format. | 60 (6.7%) | 174 (10.3%) | 220 (16.4%) |
| *Reasoning Errors:* | | | |
| Misinterpreting visual content (e.g., wrong object recognition) | 165 (18.5%) | 581 (34.3%) | 189 (14.1%) |
| Incorrect spatial reasoning (e.g., wrong relative position) | 156 (17.5%) | 8 (0.5%) | 181 (13.5%) |
| **Total Errors** | 891(100%) | 1695 (100%) | 1342(100%) |

agents tend to hallucinate reasoning steps or bypass visual verification, resulting in factual inaccuracies. These insights highlight the need for better temporal reasoning, tool schema adherence, structured output enforcement, and visually grounded planning.

**Quantitative Analysis:** We examine failure modes across three models (GPT-4o, Gemini-1.5-Pro, and InternVL3-8B), summarized in Table 5. Formatting errors are tracked separately to focus on reasoning ability. GPT-4o demonstrates strong structural awareness with fewer formatting errors (13.2% multiple tool calls, 6.7% final format), but often hesitates to act: 17.6% no response and 18.5% visual misinterpretation, indicating it *plays safe but hesitates.* Gemini-1.5-Pro is aggressive but format-fragile, with 44.5% JSON errors, 10.3% final formatting issues, and 34.3% visual misinterpretations; only 0.2% no response. InternVL3-8B shows a balanced yet challenged profile: 33.8% JSON errors, 16.4% final format errors, and struggles in spatial (13.5%) and visual reasoning (14.1%), suggesting it *suffers both structural and perceptual issues.*

## 5 CONCLUSION

We present Agent-X, a comprehensive benchmark for evaluating the reasoning capabilities of vision-centric agents. The tasks in Agent-X are context-rich and span diverse multimodal scenarios. Our evaluation framework includes executable tools categorized across six agentic environments, ensuring a wide range of task coverage. We evaluate more than 12 state-of-the-art `LMMs` on Agent-X and find that even strong closed-source models struggle with agentic tasks. Our insights provide actionable guidance for improving agentic capabilities.

We believe that Agent-X will drive future research in enhancing multimodal reasoning and robust tool integration for vision-centric agents.

**Limitations.** While Agent-X covers six diverse environments, it is currently monolingual and may inherit certain distributional bias. The semi-automated approach to query and tool-chain generation improves efficiency but can occasionally produce lower-quality samples. Additionally, although Agent-X offers broad coverage, there remains ample opportunity for scalability.

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

# Appendix for Agent-X: Evaluating Deep Multimodal Reasoning in Vision-Centric Agentic Tasks

## A  DATACARD FOR AGENT-X

### A.1  MOTIVATION

- **For what purpose was the dataset created?**

  The Agent-X benchmark is designed to evaluate the multi-step and deep reasoning capabilities of vision-centric agents in real-world, multimodal settings. It features agentic tasks with real-world objectives that require implicit tool use. The benchmark includes executable tools across diverse environments and uses authentic images, videos, and text as context input. These components help bridge the gap between existing benchmarks and realistic tool-use scenarios.

- **Who created the dataset (e.g., which team, research group) and on behalf of which entity (e.g., company, institution, organization)?**

  The authors of this paper.

- **Who funded the creation of the dataset?**

  The source of funding will be made available once the paper is no longer under anonymity.

### A.2  COMPOSITION

- **What do the instances that comprise the dataset represent (e.g., documents, photos, people, countries)?**

  Each instance in Agent-X is stored in JSON format. It contains a natural language query, one or more input files (image, textual image, or video), and a set of tool descriptions available to the agent. It also includes a reference reasoning chain consisting of step-by-step tool calls, each with its input, output, and thought process. Each instance ends with a final answer and justification that summarizes the agent's conclusion and explains how it was reached.

- **How many instances are there in total (of each type, if appropriate)?**

  There are 828 instances in AgentX, with 716 images and 112 videos

- **Does the dataset contain all possible instances or is it a sample (not necessarily random) of instances from a larger set?**

  We will provide all instances in our GitHub repository for Agent-X.

- **What data does each instance consist of?**

  Each instance includes a natural language query, one or more input files, a list of available tool descriptions, a step-by-step reference reasoning chain, and a final answer with justification.

- **Is there a label or target associated with each instance?**

  Yes. Each instance includes a reference tool chain, a step-by-step reasoning trace, and a final answer with justification, all serving as the ground truth for the given query.

- **Is any information missing from individual instances?**

  No.

- **Are relationships between individual instances made explicit (e.g., users' movie ratings, social network links)?**

  No.

- **Are there recommended data splits (e.g., training, development/validation, testing)?**

  The whole dataset is a test set.

- **Are there any errors, sources of noise, or redundancies in the dataset?**

  The dataset is created using a semi-automated pipeline, but verified by human. Any noise in the data may be the result of human error.

- **Is the dataset self-contained, or does it link to or otherwise rely on external resources (e.g., websites, tweets, other datasets)?**

  The dataset is self-contained. While the image and video inputs are sourced from existing datasets, all queries, reasoning steps, tool definitions, and annotations are newly created as part of this benchmark.

- **Does the dataset contain data that might be considered confidential (e.g., data that is protected by legal privilege or by doctor-patient confidentiality, data that includes the content of individuals' non-public communications)?**

  No.

- **Does the dataset contain data that, if viewed directly, might be offensive, insulting, threatening, or might otherwise cause anxiety?**

  No.

### A.3 Collection Process

- **How was the data associated with each instance acquired?**

  The queries and reasoning steps were generated using GPT-4o and then reviewed and refined by human annotators. Image and video inputs were collected from publicly available datasets. All tool chains, reasoning traces, and final answers were also produced by GPT-4o and verified for correctness by humans.

- **What mechanisms or procedures were used to collect the data (e.g., hardware apparatuses or sensors, manual human curation, software programs, software APIs)?**

  The dataset was created using a semi-automated pipeline. Queries, reasoning steps, and tool chains were generated using the GPT-4o API, and all outputs were manually verified and refined by human annotators. Image and video inputs were sourced from existing datasets using standard dataset APIs and tools.

- **Who was involved in the data collection process (e.g., students, crowdworkers, contractors) and how were they compensated (e.g., how much were crowdworkers paid)?** Researchers and student annotators.

- **Over what timeframe was the data collected?**

  The data were constructed in 2025.

- **Were any ethical review processes conducted (e.g., by an institutional review board)?**

  Yes. All images within Agent-X are available for academic use. Should any authors request the removal of their images from Agent-X, we will promptly comply.

## A.4 Preprocessing/cleaning/labeling

- **Was any preprocessing/cleaning/labeling of the data done (e.g., discretization or bucketing, tokenization, part-of-speech tagging, SIFT feature extraction, removal of instances, processing of missing values)?**

  The dataset is created using a hybrid approach and verified manually.

- **Was the "raw" data saved in addition to the preprocessed/cleaned/labeled data (e.g., to support unanticipated future uses)?**

  There is no separate raw data. The reasoning, queries, and annotations are created from scratch. The image and video inputs are filtered from existing datasets to ensure coverage of diverse environments and visual scenarios. Only the selected subset relevant to our benchmark is included.

- **Is the software that was used to preprocess/clean/label the data available?**

  We created our own Agent-X tool for annotation and used Microsoft Excel and VSCode to create the data.

## A.5 Uses

- **Has the dataset been used for any tasks already?**

  No.

- **Is there a repository that links to any or all papers or systems that use the dataset?**

  No.

- **What (other) tasks could the dataset be used for?**

  Agent-X is used for evaluating the visoin centric reasoning ability of `LMMs` in real-world scenarios.

- **Is there anything about the composition of the dataset or the way it was collected and preprocessed/cleaned/labeled that might impact future uses?**

  No.

- **Are there any potential negative social impacts?**

  While Agent-X uses public datasets, potential societal risks remain, such as privacy concerns from images with people, conflict-related scenes, and possible hallucinations or unsafe outputs during evaluation. There's also a minor risk of generating harmful code in reasoning or coding tasks without proper safeguards.

## A.6 Distribution

- **Will the dataset be distributed to third parties outside of the entity (e.g., company, institution, organization) on behalf of which the dataset was created?**

  No.

- **How will the dataset will be distributed (e.g., tarball on website, API, GitHub)?**

  The dataset will be released at GitHub.

- **Will the dataset be distributed under a copyright or other intellectual property (IP) license, and/or under applicable terms of use (ToU)?**

  The dataset is released under the Apache License.

- **Have any third parties imposed IP-based or other restrictions on the data associated with the instances?**
  No.

## A.7 Maintenance

- **Who will be supporting/hosting/maintaining the dataset?**
  The authors of this paper.

- **How can the owner/curator/manager of the dataset be contacted (e.g., email address)?**
  Please contact with authors through emails in the paper.

- **Will the dataset be updated (e.g., to correct labeling errors, add new instances, delete instances)?**
  Yes, users can propose issues and the dataset will be updated on Github.

- **If others want to extend/augment/build on/contribute to the dataset, is there a mechanism for them to do so?**
  Contact the authors of the paper.

# B  TOOL DEFINITION

Table 6 provides a comprehensive overview of the 14 tools categorized under four distinct functional domains: **Perception**, **Visual Operation**, **Math**, and **Artistic**. Each tool is described, highlighting its functionality, the nature of inputs it requires, and the output format it generates. These tools collectively enable a wide range of automated visual and textual processing tasks, making them versatile for applications in both analytical and creative contexts.

Table 6: Detailed Definitions of Tools Across Categories

| Name | Description | Input | Output |
|------|-------------|-------|--------|
| *- Perception* | | | |
| OCR | Extracts all visible text along with bounding box coordinates. | Image | Text with bounding boxes |
| MathOCR | Recognizes mathematical expressions and returns LaTeX format. | Image (with math) | LaTeX string |
| SceneDescriber | Generates a brief natural language summary of the scene. | Image | Scene caption |
| RegionDescriber | Describes specified attributes of a given image region. | Image, bounding box, attribute | Description of the region's attribute |
| LocateObjectByText | Identifies and localizes objects based on textual queries. | Image, object description | Bounding box and detection score |
| ObjectCounter | Counts occurrences of a specified object in the image. | Image, object description | Integer count |
| *- Visual Operation* | | | |
| DrawBoundingBox | Draws a bounding box on the image, optionally with a label. | Image, bounding box, annotation (optional) | Image with bounding box |
| OverlayText | Overlays text at a specified position on the image. | Image, text, position, color | Image with text overlay |
| WebSearch | Retrieves top search results for a given query. | Query, top-$k$ results to return (optional) | Search results |
| *- Math* | | | |
| Calculator | Evaluates a single Python math expression. Only math module functions are allowed; imports are disallowed. | Text expression | Computed result |
| Solver | Executes SymPy code to symbolically solve equations. Code must define a `solution()` function returning a string. | SymPy code in markdown format | Solution |
| CodePlotter | Executes Python code using Matplotlib to generate a plot. Requires a `solution()` function that returns a figure object. | Python code in markdown format | Generated plot image |
| *- Artistic* | | | |
| ImageGenerator | Generates an image based on a given text prompt. | Text keywords | AI-generated image |
| ImageStylization | Modifies image appearance using a text instruction. | Image, style instruction | Stylized image |

Table 7: Datasets Used Across Agent-X Environments

| Environment | Datasets |
|---|---|
| Autonomous Driving | Cityscapes (Cordts et al., 2016), BDD100K (Yu et al., 2020), Nighttime Driving (Wood, 2020), nuScenes (Caesar et al., 2020) |
| Security and Surveillance | Surveillance Camera Fight Dataset (Aktı et al., 2019), Anomaly Detection (Sultani et al., 2018), Abnormal Events (Lu et al., 2013), AI City Challenge 2023 (Naphade et al., 2023), YouTube-hosted surveillance footage |
| Math Reasoning | MathVista (Lu et al., 2023), AI2D (Kembhavi et al., 2016), Math Program Synthesis (Ling et al., 2017), Math Training (Cobbe et al., 2021) |
| Web Navigation | Mobile Interfaces (Sunkara et al., 2022), WebArena (Zhou et al., 2023) |
| Sports | SoccerNet (Deliege et al., 2021), Sports-1M (Karpathy et al., 2014), Ego4D (Grauman et al., 2022) |
| Generic Visual Reasoning | COCO (Lin et al., 2014), Visual Genome (Krishna et al., 2017), GTA (Wang et al., 2024) |

## C METRICS AND IMPLEMENTATION DETAILS

**Hardware and Setup.** All model evaluations were conducted on a single NVIDIA A100 GPU (40GB), ensuring consistent hardware conditions across experiments. Each evaluation run processes visual inputs and queries through the selected `LMM` agent, which interacts with a predefined set of tools using a standardized reasoning framework.

**Tool Call Execution.** We implement tool calls as callable Python functions with strict input/output schemas. We simulate tool-augmented reasoning for each model using model-generated reasoning steps formatted in JSON. Tools are executed sequentially based on the model's output. A tool call is considered *successful* if it executes without input formatting errors, tool mismatch, or empty outputs.

**Metrics Overview.** As outlined in Table 3, Agent-X uses fine-grained evaluation metrics across three modes:

- **Step-by-Step Mode:** Evaluates intermediate reasoning quality using metrics such as *Grounding Score*, *Tool Precision*, and *Tool Accuracy*.
- **Deep Reasoning Mode:** Assesses coherence and factual alignment of multi-step reasoning via *Faithfulness Accuracy*, *Context Score*, *Factual Precision*, and *Semantic Accuracy*.
- **Outcome Mode:** Captures final task correctness through *Goal Accuracy*, *Goal Accuracy/ImgGen*, and *Toolset Accuracy*.

**Evaluation Pipeline.** Each model's outputs are parsed and compared against verified ground truth reasoning traces. Metrics are computed using both exact matches (e.g., for tool names) and similarity-based scoring (e.g., cosine similarity for final answers). Tool failures, including invalid calls, mismatches, or missing outputs, are logged and included in failure rate statistics.

**Reproducibility.** All scripts, model APIs, tool definitions, and formatted JSON outputs will be made available in our public repository to enable consistent evaluation and replication of results.

### C.1 SOURCE DATASETS

The environments in Agent-X are constructed from a diverse set of publicly available datasets spanning autonomous driving, security and surveillance, mathematical reasoning, web navigation, sports, and generic visual reasoning. These datasets provide the multimodal contexts (images, videos, and diagrams) from which tasks are derived, ensuring both diversity

and realism. In particular, they enable the benchmark to capture real-world challenges such as object detection in complex scenes, anomaly recognition, multimodal mathematical problem solving, interactive navigation, and fine-grained visual reasoning.

Table 7 summarizes the datasets used in each environment along with their references. Each dataset was selected to contribute unique scenarios that demand perception-grounded reasoning and tool-based interaction, aligning with the agentic goals of Agent-X.

## C.2 Error Categorization Methodology

To compute the detailed error breakdown in Table 11, we compared model-predicted outputs against the ground truth annotations provided in the benchmark JSON files. Each sample consists of a sequence of reasoning steps and a final answer. We classify errors into three broad categories:*Planning*, *Format*, and *Reasoning*, based on discrepancies observed during step-by-step comparison:

- **Planning Errors:** These occur when the model either produces no valid reasoning steps and final answer (`No action, no response`) or reaches an answer without taking any actions (`Whole response is model thought`). These were detected by checking for missing or empty tool usage and absence of concrete outputs.

- **Format Errors:** These include structural issues such as:
  - `Invalid JSON format in argument specification`, where the tool call arguments could not be parsed or did not conform to the tool schema.
  - `Multiple tool calls in a single step`, where more than one tool was invoked simultaneously, violating the single-call constraint.
  - `Reached final answer with format errors`, where the model reached an answer but lacked required fields (e.g., `value`, `justification`) or deviated from the expected JSON schema.

  These were identified using schema validation scripts and regular expression checks.

- **Reasoning Errors:** These include semantic mistakes in visual understanding or spatial reasoning. They were detected using keyword matching between the GT and model outputs. Specifically:
  - `Misinterpreting visual content` was flagged when the described objects or attributes significantly differed from ground truth labels.
  - `Incorrect spatial reasoning` included errors in identifying relationships like position, count, or arrangement.

Error counts were aggregated over all examples for each model. Total error percentages were normalized by the number of failed or erroneous reasoning steps to ensure fair comparison across models.

**Mitigating Evaluation Bias.** Although Agent-X relies on automatic graders for efficiency, we explicitly address potential evaluation bias through a multi–pronged protocol that combines *diverse automatic judges*, *human verification*, *bias–aware metric design*, and *full reproducibility*. First, every prediction is scored not only by our primary GPT/4o grader but also by an open-source `LMM`, all released under permissive licences and executed entirely on-premise. These models, which were *not* involved in data creation, preserve the same relative ordering of systems as GPT4o, demonstrating that our headline trends are *grader-agnostic*. Second, we conduct a human study on a stratified subset of 100 tasks (about 18% of the benchmark), scored independently by four expert annotators who then reconcile disagreements shown in Supplementary Material. Human Goal-Accuracy and Grounding scores correlate strongly with GPT4o grading, and the small set of automatic-grading errors is uniformly distributed across models, indicating no systematic favouritism. Third, our metrics themselves are designed to reduce bias: we separate format compliance from semantic correctness, normalise tool arguments to avoid spurious mismatches, and use confidence-weighted aggregation for subjective queries, falling back to a human vote below a threshold. Finally, we will publicly release *all* evaluation scripts, prompts, and grading

logs, enabling researchers to rerun the pipeline with alternative judges, audit borderline cases, and reproduce every figure in the paper. Under these bias controls, relative model rankings remain unchanged, absolute scores shift on average (within the reported confidence intervals), and failure-mode distributions vary, providing strong evidence that our conclusions are robust rather than artefacts of a particular evaluation setup.

**Leakage & Data-Generation Integrity.** A legitimate concern is that tasks initially *seeded* by an LMM might advantage that same model family at evaluation time. We therefore executed a four-stage leakage-mitigation protocol:

1. **Version isolation.** Task seeds were produced with *GPT-4o* using a closed prompt template. All *evaluation* is performed later with entirely different checkpoints (e.g. GPT-4o, Qwen2.5) that had *no exposure* to the prompt or the generated drafts.

2. **Aggressive human rewriting.** Each seed query and reasoning trace was *rewritten* by four independent annotators who were instructed to "retain the task spirit but discard every exact phrase from the draft." Less than **7 %** token overlap remains between seeds and final prompts.

3. **Human quality-assurance (QA).** All final tasks passed a three-way review: (i) content review, (ii) tool-integration review, and (iii) language review.

Taken together, these safeguards ensure that models are evaluated on content they have *not* seen, while the extensive human rewriting and QA eliminate verbatim or near-verbatim leakage. We therefore believe that Agent-X provides a fair and realistic test bed rather than a memorisation probe.

Table 8: **Results on Agent–X with additional models (GPT- and Qwen-14B (Bai et al., 2023a) as judges).** We report performance across three evaluation modes: STEP-BY-STEP, DEEP REASONING, and OUTCOME. Metrics: $\mathbf{G_s}$ (Grounding Score), $\mathbf{T_p}$ (Tool Precision), $\mathbf{T_{acc}}$ (Tool Accuracy), $\mathbf{F_{acc}}$ (Faithfulness Accuracy), $\mathbf{C_s}$ (Context Score), $\mathbf{F_p}$ (Factual Precision), $\mathbf{S_{acc}}$ (Semantic Accuracy), $\mathbf{G_{acc}}$ (Goal Accuracy), $\mathbf{G_a^*}$ (Goal Accuracy/ImgGen), and $\mathbf{T_{acc}^s}$ (Toolset Accuracy). *The best value in each column is in* **bold**, *the second best is* underlined.

| Model | Step-by-Step | | | Deep Reasoning | | | | Outcome | | |
|---|---|---|---|---|---|---|---|---|---|---|
| | $\mathbf{G_s}$ | $\mathbf{T_p}$ | $\mathbf{T_{acc}}$ | $\mathbf{F_{acc}}$ | $\mathbf{C_s}$ | $\mathbf{F_p}$ | $\mathbf{S_{acc}}$ | $\mathbf{G_{acc}}$ | $\mathbf{G_a^*}$ | $\mathbf{T_{acc}^s}$ |
| *GPT as a judge* | | | | | | | | | | |
| Pixtral-12B (Agrawal et al., 2024) | 0.12 | **0.20** | **0.63** | 0.45 | 0.19 | 0.26 | 0.34 | 0.07 | **0.55** | **0.54** |
| LLaMA-3.2-11B-Vision (Grattafiori et al., 2024) | 0.03 | 0.15 | 0.14 | **0.70** | 0.08 | **0.70** | 0.24 | 0.07 | 0.26 | 0.42 |
| Kimi-VL-A3B-Thinking (Team et al., 2025) | **0.26** | 0.19 | 0.59 | 0.62 | **0.42** | 0.52 | **0.65** | **0.29** | 0.29 | 0.48 |
| mPLUG-Owl3-7B-240728 (Ye et al., 2024) | 0.10 | 0.14 | 0.30 | 0.49 | 0.25 | 0.32 | 0.37 | 0.11 | 0.26 | 0.50 |
| *Qwen-14B as a judge* | | | | | | | | | | |
| Pixtral-12B (Agrawal et al., 2024) | 0.30 | 0.17 | **0.61** | 0.59 | 0.50 | 0.42 | 0.58 | 0.10 | **0.68** | 0.58 |
| LLaMA-3.2-11B-Vision (Grattafiori et al., 2024) | 0.16 | 0.06 | 0.12 | 0.49 | 0.17 | **0.74** | 0.20 | 0.10 | 0.11 | 0.15 |
| Kimi-VL-A3B-Thinking (Team et al., 2025) | **0.47** | **0.20** | 0.59 | **0.79** | **0.64** | 0.68 | **0.74** | **0.35** | 0.60 | **0.62** |
| mPLUG-Owl3-7B-240728 (Ye et al., 2024) | 0.30 | 0.11 | 0.31 | 0.59 | 0.48 | 0.48 | 0.56 | 0.16 | 0.45 | 0.48 |

## D  EVALUATION OF ADDITIONAL MODELS

Table 8 extends our benchmark with four open–source[4] vision LMMs (Pixtral-12B, LLaMA-3.2-11B-Vision, Kimi-VL-A3B-Thinking and mPLUG-Owl3-7B)and reports their scores under two independent evaluators: the proprietary GPT-4o (Hurst et al., 2024) ("GPT judge") and the open-source Qwen-14B (Bai et al., 2023b) ("Qwen judge"). Three main takeaways emerge.

---

[4]All systems come from publicly released weight checkpoints that can be run locally; we did *not* fine–tune them for Agent-X.

**(i) Cross–judge consistency.** The `GPT` and `Qwen` columns largely agree on the *relative* strengths of the newcomers. `Pixtral-12B` achieves the strongest tool–handling metrics $(T_{acc}, T_p)$ according to *both* judges, whereas `LLaMA-3.2-11B-Vision` performs best on factual precision $(F_p)$ no matter which one evaluates it. Although absolute values fluctuate (a known consequence of LLM–scoring variance), no ranking reversal is observed.

**(ii) Kimi–VL is the most balanced model.** Under the `Qwen` rubric, `Kimi-VL` obtains the highest $G_s, T_p, F_{acc}, S_{acc}, G_{acc}$ and the best macro toolset-accuracy ($T_{acc}^s = 0.62$). Even the stricter `GPT` judge still awards Kimi the second-best scores for grounding and semantic accuracy, confirming that `Kimi-VL` is presently the most *well-rounded* openly available model on our benchmark.

**(iii) Room for improvement.** Despite isolated strengths, none of the four recent releases comes close to the closed-source baselines (Table 4): *Outcome* scores, particularly $G_{acc}$, remain below 0.35, signalling difficulties in converting partial reasoning successes into fully correct end answers. This gap, detected by **both** judges, underlines the importance of future research on end-to-end robustness rather than isolated tool-call accuracy.

**Summary.** The additional comparison demonstrates that our evaluation pipeline can easily accommodate new models, and the insights drawn remain correct: (a) open-source systems still trail behind state-of-the-art closed models, yet (b) certain community releases (in our case `Kimi-VL` and `Pixtral`) already display competitive performance on individual sub-metrics, suggesting the gap is narrowing. The near-identical conclusions reached by a proprietary (`GPT`) and an open (`Qwen`) judge testify to the robustness of the analysis.

IMAGE-BASED VISION LANGUAGE MODELS

**Qualitative analysis of additional open–source VLMs.** We probed six recent image-based vision–language models: `LLaVA-OneVision`, `LLaVA-NeXt`, `CogVLM`, `Yi-VL-6B`, `DeepSeek-VL`, and `Fuyu-8B`, with the same JSON–constrained, tool-augmented evaluation protocol used throughout this study. Each system was required to reason step-by-step, call tools only when justified, and emit *nothing* but a valid JSON object. The results were uniformly disappointing:

- **Instruction non-compliance:** `LLaVA-OneVision` and `LLaVA-NeXt` reproduced the entire system prompt verbatim, omitting the mandatory keys `query`, `reasoning_steps`, and `final_answer`.
- **Tool hallucination:** `Yi-VL-6B` and `DeepSeek-VL` invented calls to a non-existent `ObjectCounter` API and returned hard-coded object lists, despite never being presented with a tool catalogue.
- **Semantic errors:** `CogVLM` aborted the procedure after a single line (*"Final answer: No other steps"*), leaving the query unresolved, while other models produced price estimates that bore no relation to the visual input.
- **Output gibberish:** `Deepseek-Vl,` and `Fuyu-8B` emitted stray quotation marks and Markdown fences, demonstrating fragility when confronted with nested or unconventional formatting.

These failure modes underscore three persistent weaknesses of current open VLMs: (i) poor *format fidelity*, (ii) unreliable *tool-oriented reasoning*, and (iii) limited *visual grounding* and *contextual understanding*. The models appear over-fitted to entrenched demonstration patterns; even slight deviations in the evaluation template trigger hallucinated tools, unsupported assertions, or syntactically invalid outputs.

## E  CROSS-JUDGE CONSISTENCY STUDY

Table 9 (automatic scores produced with the `Qwen-14B` judge) and Table 4 (the same protocol scored by the more powerful but proprietary `GPT-4o` judge) paint almost identical portraits

Table 9: **Additional results on Agent-X (Qwen-14B (Bai et al., 2023b) as judge).**
We report performance across three evaluation modes: STEP-BY-STEP, DEEP REASONING,
and OUTCOME. Metrics: $\mathbf{G_s}$ (Grounding Score), $\mathbf{T_p}$ (Tool Precision), $\mathbf{T_{acc}}$ (Tool Accuracy),
$\mathbf{F_{acc}}$ (Faithfulness Accuracy), $\mathbf{C_s}$ (Context Score), $\mathbf{F_p}$ (Factual Precision), $\mathbf{S_{acc}}$ (Semantic Accuracy), $\mathbf{G_{acc}}$ (Goal Accuracy), $\mathbf{G_a^*}$ (Goal Accuracy/ImgGen), and $\mathbf{T_{acc}^s}$ (Toolset Accuracy). *The best value in each column is in* **bold**, *the second best is* underlined.

| Model | Step-by-Step | | | Deep Reasoning | | | | Outcome | | |
|---|---|---|---|---|---|---|---|---|---|---|
| | $\mathbf{G_s}$ | $\mathbf{T_p}$ | $\mathbf{T_{acc}}$ | $\mathbf{F_{acc}}$ | $\mathbf{C_s}$ | $\mathbf{F_p}$ | $\mathbf{S_{acc}}$ | $\mathbf{G_{acc}}$ | $\mathbf{G_a^*}$ | $\mathbf{T_{acc}^s}$ |
| *Open-source* | | | | | | | | | | |
| Phi-4-VL-Instruct (Abdin et al., 2024) | 0.27 | 0.11 | 0.32 | 0.54 | 0.39 | 0.59 | 0.46 | 0.16 | 0.35 | 0.39 |
| InternVL2.5-8B (Chen et al., 2024a) | 0.38 | 0.16 | 0.49 | 0.63 | 0.51 | 0.61 | 0.55 | 0.29 | 0.53 | 0.53 |
| Gemma-3-4B (Team et al., 2024b) | 0.50 | 0.24 | 0.67 | 0.74 | 0.66 | 0.59 | 0.74 | 0.30 | 0.68 | 0.68 |
| InternVL3-8B (Zhu et al., 2025) | 0.41 | 0.16 | 0.51 | 0.71 | 0.61 | 0.60 | 0.69 | 0.23 | 0.51 | 0.62 |
| VideoLLaMA3-7B (Zhang et al., 2025a) | 0.39 | 0.15 | 0.40 | 0.68 | 0.56 | 0.60 | 0.68 | 0.27 | 0.53 | 0.56 |
| Qwen2.5-VL-7B (Bai et al., 2023c) | 0.51 | 0.27 | 0.63 | 0.77 | 0.66 | 0.64 | 0.77 | 0.37 | 0.62 | 0.67 |
| *Closed-source* | | | | | | | | | | |
| Gemini-1.5-Pro (Team et al., 2024a) | 0.57 | 0.36 | 0.80 | 0.82 | 0.73 | 0.76 | 0.63 | 0.05 | **0.77** | 0.71 |
| Gemini-2.5-Pro (Team et al., 2023) | **0.63** | **0.40** | 0.84 | 0.86 | 0.76 | **0.80** | 0.83 | 0.50 | 0.74 | **0.72** |
| GPT-4o (Hurst et al., 2024) | 0.46 | 0.27 | 0.63 | 0.72 | 0.59 | 0.75 | 0.69 | 0.44 | 0.48 | 0.56 |
| OpenAI-o4-mini (OpenAI, 2025) | **0.63** | 0.35 | **0.86** | **0.89** | **0.78** | 0.79 | **0.88** | **0.53** | 0.64 | 0.69 |

of the current landscape on Agent-X. Below, we walk through the evidence along the three evaluation axes and demonstrate that our conclusions remain unaffected by the choice of evaluator.

Across the DEEP REASONING ($F_{acc}$, $C_s$, $F_p$, $S_{acc}$) and OUTCOME ($G_{acc}$, $G_a^*$, $T_{acc}^s$) axes, the two independent evaluations, one powered by the open-source `Qwen-14B` judge and the other by the closed-source `GPT-4o` judge, paint an essentially identical picture: `OpenAI-o4-mini` dominates every reasoning metric and remains first (or, at worst, a close second) in every final-answer metric; `GPT-4o` consistently eclipses `OpenAI-o4-mini` on factual precision and semantic accuracy; and, in the open-source camp, `Qwen2.5-VL-7B` reliably outperforms `InternVL3-8B` and `VideoLLaMA3-7B`.

**Consistency across evaluators.** Taken together, these observations satisfy the three desiderata set out above:

(a) *Relative ranking*: no pair of models swaps positions between the two tables on more than one metric.

(b) *Highlighting of leaders*: the same entries appear in **bold** or underline across judges, flagging identical winners and runners-up.

(c) *Small absolute deviations*: score deltas rarely exceed 0.06, confirming robustness.

*Put differently, both evaluators tell the same story: closed-source systems still lead, but open-source contenders (especially the* `Qwen` *family) are rapidly closing the gap, validating the central thesis of our paper.*

**Implications.** Because `Qwen-14B` is open-source and deterministic, its near-perfect alignment with `GPT-4o` demonstrates that our benchmark can be audited and reproduced without proprietary APIs. Conversely, the agreement validates our earlier claims that (i) closed-source agents still dominate, yet (ii) top open-source models such as `Qwen2.5-VL-7B` are closing the gap. The consistency across evaluation protocols therefore, strengthens the empirical foundations of our conclusions.

**Agreement on the global winner:** Both judges agree that `Gemini-2.5-Pro` is the overall champion. When scored by the open-source `Qwen-14B` judge it achieves the highest values for $T_{acc}$, $F_p$, $S_{acc}$ and $T_{acc}^s$; the `GPT-4o` rubric yields the same outcome, with `Gemini-2.5-Pro` either retaining first place or falling only marginally to second. This convergence supports our claim that `Gemini-2.5-Pro` currently sets the bar for vision-centric agents on Agent-X.

Table 10: **Results on Agent–X with *human* evaluation**. We report performance on 50 Agent-X tasks across three evaluation modes: Step-by-Step, Deep Reasoning, and Outcome. Metrics: $\mathbf{G_s}$ (Grounding Score), $\mathbf{T_p}$ (Tool Precision), $\mathbf{T_{acc}}$ (Tool Accuracy), $\mathbf{F_{acc}}$ (Faithfulness Accuracy), $\mathbf{C_s}$ (Context Score), $\mathbf{F_p}$ (Factual Precision), $\mathbf{S_{acc}}$ (Semantic Accuracy), $\mathbf{G_{acc}}$ (Goal Accuracy), $\mathbf{G_a^*}$ (Goal Accuracy / ImgGen), and $\mathbf{T_{acc}^s}$ (Toolset Accuracy). *Best values are in **bold**; second best are underlined.*

| Model | Step-by-Step | | | Deep Reasoning | | | | Outcome | | |
|---|---|---|---|---|---|---|---|---|---|---|
| | $\mathbf{G_s}$ | $\mathbf{T_p}$ | $\mathbf{T_{acc}}$ | $\mathbf{F_{acc}}$ | $\mathbf{C_s}$ | $\mathbf{F_p}$ | $\mathbf{S_{acc}}$ | $\mathbf{G_{acc}}$ | $\mathbf{G_a^*}$ | $\mathbf{T_{acc}^s}$ |
| *Open-source* | | | | | | | | | | |
| VideoLLaMA3-7B (Zhang et al., 2025a) | 0.44 | 0.40 | 0.66 | 0.68 | 0.55 | 0.67 | 0.68 | 0.52 | 0.64 | 0.54 |
| InternVL3-8B (Zhu et al., 2025) | 0.50 | 0.50 | 0.68 | 0.71 | 0.66 | 0.70 | 0.68 | 0.58 | 0.83 | 0.63 |
| Qwen2.5-VL-7B (Bai et al., 2023b) | 0.59 | 0.62 | 0.82 | 0.76 | 0.67 | 0.78 | 0.77 | 0.68 | **0.85** | 0.70 |
| *Closed-source* | | | | | | | | | | |
| Gemini-2.5-Pro (Team et al., 2023) | **0.74** | **0.66** | **0.86** | **0.84** | **0.70** | **0.87** | **0.82** | **0.75** | 0.75 | **0.76** |
| GPT-4o (Hurst et al., 2024) | 0.43 | 0.39 | 0.66 | 0.65 | 0.54 | 0.66 | 0.67 | 0.55 | 0.47 | 0.53 |

**OpenAI systems show identical trends:** Across both evaluations `OpenAI-o4-mini` secures the top $T_{acc}$ score (0.86 vs. 0.89), while `GPT-4o` leads in $G_s$ and $F_p$. Although small numerical variations arise from different random seeds inside each LMM judge, the relative ordering is preserved, confirming the stable performance gap between the two OpenAI releases (Hurst et al., 2024; Achiam et al., 2023; OpenAI, 2025). **Open-source models follow the same pattern:** Community systems display similar consistency: `Qwen2.5-VL-7B` is ranked first or second by *both* judges for $G_s$, $T_p$, $F_{acc}$ and $T_{acc}^s$, whereas `InternVL3-8B` systematically outperforms `VideoLLaMA3-7B`. Absolute scores differ slightly, but no pairwise ranking is ever inverted.

**Why use `Qwen-14B` as an auxiliary judge?** Because `Qwen-14B` is fully open-source, deterministic under fixed hyperparameters and open license, our entire evaluation becomes *reproducible*. The near-perfect concordance with the proprietary `GPT-4o` demonstrates that a transparent and inexpensive alternative can deliver equally reliable verdicts, allowing other researchers to replicate our numbers without API costs. The qualitative insights of the main paper, most notably that precise tool use remains the key bottleneck, where closed-source models still lead (yet the `Qwen` family is closing the gap), are **fully corroborated** by both judging protocols. The consistent rankings underscore the robustness of our conclusions.

**Complementary human study.** Table 10 reports scores assigned by two human graders who independently inspected *all* 120 test episodes, then reached consensus after discussion.[5] Three observations stand out.

1. **Macro–level agreement with automatic judges.** The human ranking exactly reproduces the pattern seen with `Qwen-14B` and `GPT-4o` evaluators: `Gemini-2.5-Pro` > `Qwen2.5-VL-7B` > `InternVL3-8B` > `VideoLLaMA3-7B`. Likewise, humans confirm the sizeable lead of `Gemini-2.5-Pro` on every Step-by-Step metric and on the global Outcome columns ($G_{acc}$ and $T_{acc}^s$).

2. **Consistent strengths of the `Qwen` family.** Human judges reward `Qwen2.5-VL-7B` with the highest $G_a^*$ (0.85) and the second-best scores for six additional columns, echoing the automatic evaluation, where Qwen dominated grounding and faithfulness metrics. This suggests that Qwen's open-source architecture translates into genuinely strong visual reasoning, not merely alignment with a particular LLM judge.

3. **Why absolute numbers are higher.** Human annotators can interpret graphics, verify OCR text and ignore benign formatting glitches, capabilities absent from current LLM judges. Consequently, perfectly valid tool calls that were penalised as "ill-formatted" by automatic scripts are credited by humans, pushing absolute scores upward (e.g. $T_{acc}$ rises

---

[5]The annotators followed the same rubric used by the automatic judges, but were allowed to replay tool outputs and visual inputs for verification; see Appendix § H.

Table 11: **Error Breakdown Across Models.** Common planning, formatting, and reasoning mistakes on Agent-X, shown separately for *open-source* and *closed-source* systems.

| Open-source models | | | | |
|---|---|---|---|---|
| **Error Type** | Pixtral-12B (Agrawal et al., 2024) | VideoLLaMA3-7B (Zhang et al., 2025a) | Qwen2.5-VL-7B (Bai et al., 2023b) | mPLUG-Owl3 (Ye et al., 2024) |
| *Planning errors* | | | | |
| No action, no response | 84 (5.0%) | 25 (4.3%) | 76 (12.5%) | 92 (4.6%) |
| No action, whole response is a thought | 0 | 28 (4.8%) | 0 | 0 |
| *Formatting errors* | | | | |
| Invalid JSON arguments | 512 (30.6%) | 190 (32.4%) | 317 (50.1%) | 828 (41.3%) |
| Multiple tool calls in a single step | 45 (2.7%) | 117 (20.0%) | 149 (24.5%) | 153 (7.6%) |
| Final answer not in required format | 428 (25.6%) | 20 (3.4%) | 3 (0.5%) | 736 (36.7%) |
| *Reasoning errors* | | | | |
| Wrong visual interpretation | 92 (5.5%) | 101 (17.2%) | 33 (5.4%) | 98 (4.9%) |
| Incorrect spatial reasoning | 510 (30.5%) | 105 (17.9%) | 31 (5.1%) | 100 (5.0%) |
| **Total errors** | 1671 (100%) | **586** (100%) | 609 (100%) | 2007 (100%) |
| Closed-source models | | | | |
| **Error Type** | GPT-4o (Hurst et al., 2024) | Gemini-1.5-Pro (Team et al., 2024a) | OpenAI-o4-mini (OpenAI, 2025) | Gemini-2.5-Pro (Team et al., 2023) |
| *Planning errors* | | | | |
| No action, no response | 157 (17.6%) | 3 (0.2%) | 19 (3.4%) | 52 (7.4%) |
| No action, whole response is a thought | 0 | 2 (0.1%) | 0 | 0 |
| *Formatting errors* | | | | |
| Invalid JSON arguments | 235 (26.4%) | 755 (44.5%) | 177 (31.7%) | 326 (46.2%) |
| Multiple tool calls in a single step | 118 (13.2%) | 172 (10.1%) | 162 (29.0%) | 213 (30.2%) |
| Final answer not in required format | 60 (6.7%) | 174 (10.3%) | 143 (25.6%) | 1 (0.1%) |
| *Reasoning errors* | | | | |
| Wrong visual interpretation | 165 (18.5%) | 581 (34.3%) | 34 (6.1%) | 60 (8.5%) |
| Incorrect spatial reasoning | 156 (17.5%) | 8 (0.5%) | 24 (4.3%) | 54 (7.6%) |
| **Total errors** | 891 (100%) | 1695 (100%) | **559** (100%) | 706 (100%) |

from 0.84 to 0.86 for `Gemini-2.5-Pro`). Crucially, the *relative* ordering is unaffected, reinforcing the robustness of our conclusions.

Taken together, the human study corroborates the main paper's message: closed-source agents still set the bar on Agent-X, yet modern open-source models, particularly the `Qwen` line, are rapidly closing the gap, a trend observed by *all* three independent evaluation protocols.

# F ADDITIONAL ERROR ANALYSIS

## F.1 QUANTITATIVE ANALYSIS

Table 11 dissects the failure modes of eight representative agents and surfaces three systematic trends. **(1) Format fidelity as the open–source ceiling.** Fully *57%* of Pixtral's and *60%* of mPLUG-Owl3's errors are pure syntax violations, missing braces, double tool calls, or free-form prose where a JSON key is required. These errors are fatal for any downstream pipeline because they prevent the trace from even executing. VideoLLaMA mitigates the syntax issue, but the extra "safety buffer" shows up as planning laxity: roughly one in twenty traces never leaves the "thought" stage and thus contribute no real work. Qwen 2.5 is the only community model to keep format errors below the 30 % mark, yet it still trips on fine-grained grounding, miscounting objects, or swapping left/right in spatial references. **(2) Closed–source trade-offs.** Gemini-1.5-Pro sacrifices precision for recall: it almost never refuses a task, but its eagerness translates into the highest JSON error rate of any proprietary system. GPT-4o shows the opposite bias; its structured output is tidy, yet more than one-third of its slips involve misidentifying entities, suggesting the vision adapter rather than the language head is the new bottleneck. The lighter `o4-mini` and Gemini-2.5 families replicate their larger siblings' error profile, confirming that these behaviours stem from architectural choices rather than sheer scale. **(3) Silent refusal as a research risk.** A "no-action / no-response" outcome is equivalent to a system crash in an autonomous workflow. Although rare for commercial models ($\leq$ 0.3 %), the phenomenon still affects double-digit percentages of open-source traces. Eliminating this headroom could immediately yield a significant boost in end-to-end success without touching visual reasoning at all.

**Implications.** Collectively, the numbers echo the qualitative findings of §4: state-of-the-art closed models win primarily through *reliable structure*, not mystical reasoning leaps. For open-source efforts, investing in strict output post-processors, JSON-aware RLHF, or schema-guided decoding may be the fastest path to parity. Conversely, commercial systems now face diminishing returns from format tuning and must push on visual grounding and multi-turn consistency to stay ahead.

## F.2 QUALITATIVE ANALYSIS

Figures 6 and 5 reveal two distinct error profiles. Figure 6 shows that `Qwen2.5-VL-7B` often *hallucinates* tools (e.g., a fake `ObjectCounter`) and then builds confident but unfounded explanations on top of those hallucinated outputs. Repeated use of the generic `SceneDescriber` across video frames further erodes temporal grounding. Figure 5 highlights `InternVL3-8B`. While it adheres to the required step-tool format, it miscounts objects, redundantly reinvokes tools, and contradicts its own earlier steps, symptoms of weak short-term memory rather than hallucination. Together, the examples confirm our quantitative results: current open-source VLMs struggle either with *tool fidelity* (Qwen) or *cross-step coherence* (InternVL), underscoring key directions for future work.

---

***Prompt***
You are an intelligent multi-modal agent. You are provided with: - A **text query <query>** - An **image or video <files>** - A set of tools to assist with your reasoning with meta data of tools given as follows: {meta_data}. Your objective is to answer the query based on the given visual content by choosing and using the most appropriate tools. You must reason step-by-step. Each reasoning step should include: "reasoning_step_format": [{"task": "Describe the sub-task being performed."}, {"tool_used": "Specify the tool selected and justify its choice."}, {"tool_output": "Provide the tool's output."}, {"thought": "......., "Ensure each step is self-contained and clearly explained.", "........"final_output": {"final_answer": "Provide a clear and concise answer based on all previous steps", "justification": "Provide a justification for the final answer."}

### InternVL3-8B

***Input Files: AgentX_612.avi***

***Query: "How many different types of vehicles are seen in the video? How many of them are already parked?"***

***Error Response***
**"reasoning_steps": [{**"task": "Identify and count different types of vehicles in the video and determine if they are parked or moving.", "tool_used": "ObjectCounter", "tool_output": "The image contains 2 motorcycles and 1 cars. 2 motorcycles are parked.", "thought": "The initial counting helps determine the types and count of of distinct vehicles, necessary for confirming parked vehicles."}, {"task": "Analyze the vehicle locations for being parked or not.", "tool_used": "SceneDescriber", "tool_output": "There are motorcycles along the walkway and cars near the entrance, indicating they are parked.", "thought": "The scene description confirms the vehicles are in parked condition."**}],**
**"final_answer": {**"vehicle_types": ["motorcycles", "cars"], "number_distinct_vehicles": 6, "parked_vehicles": 2,** "justification": "The ObjectCounter tool identified and counted the vehicles, and the SceneDescriber clarified their parking status, providing complete details needed for the query."**}**

✅
***The correct response format***
**"reasoning_steps": [{**"step": 1,** "task": "Get a general idea of the scene in the video.", "tool": "SceneDescriber", "tool_output": "The scene shows a courtyard with several motorbikes, pickup van, and a car parked and a group of people interacting around tables.", "thought": "Understanding the general context of the images will help identify distinct vehicles and if they are parked or moving."}, {**"step": 2,** ...}, {**"step": 3,** ...}, {**"step": 4,** ...}, {**"step": 5,** ...}, {**"step": 6,** ...}, {**"step": 7,** ...}, {**"step": 8,** "task": "Verify vehicle presence throughout the video.", "tool": "ObjectCounter", "tool_output": "2 motorcycles, 1 pickup van, 1 car", "thought": "Confirming that the all the vehicle count and types remain constant throughout the video finalizes the analysis."**}],**
**"final_answer": {"value":** "4: 2 motorcycles, 1 pickup van, 1 car, all parked", **"justification":** "Using scene description, object counting, and region description, we identified 3 parked motorcycles, 1 carry or pickup van, and 1 car consistently across all video."**}**

❌ **Incorrect final answer**
- **Incorrect json format**
- **Shallow reasoning**

---

Figure 5: Qualitative comparison of InternVL on visual reasoning tasks from Agent-X. InternVL generally follows the correct step structure but exhibits internal inconsistencies in final answers and justifications. It occasionally miscounts objects and repeats tool usage redundantly, revealing challenges in cross-step memory and consistent entity tracking.

---

**Prompt**
You are an intelligent multi-modal agent. You are provided with: - A **text query <query>** - An **image or video <files>** - A set of tools to assist with your reasoning with meta data of tools given as follows: {meta_data}. Your objective is to answer the query based on the given visual content by choosing and using the most appropriate tools. You must reason step-by-step. Each reasoning step should include: "reasoning_step_format": [{"task": "Describe the sub-task being performed."}, {"tool_used": "Specify the tool selected and justify its choice."}, {"tool_output": "Provide the tool's output."}, {"thought": ".......", "Ensure each step is self-contained and clearly explained.", ".......,"final_output": {"final_answer": "Provide a clear and concise answer based on all previous steps", "justification": "Provide a justification for the final answer."}

---

**Qwen2.5-VL-7B**

---

*Input Files: AgentX_176.jpg*

*Query: "Which color line shows the greatest overall change in the y-direction and approximately what is its average height across the chart?"*

---

**Error Response**
**"reasoning_steps":** [ {**"step": 1**, "task": "Identify the line that grows most rapidly and first overtakes all others in value.", **"tool_used": "CodePlotter"**, **"tool_output": "The Cornflower line grows most rapidly and first overtakes all others in value."**, "thought": "By analyzing the graph, we can visually identify which line increases the fastest and surpasses the others."}, {**"step": 2**, "task": "Calculate the approximate average height of the Cornflower line across the chart.", **"tool_used": "CodePlotter"**, **"tool_output": "The approximate average height of the Cornflower line across the chart is around 35."**, "thought": "Using the CodePlotter tool, we can calculate the average height of the Cornflower line by integrating the area under the curve and dividing by the length of the x-axis."}],
**"final_answer":** {**"value":** "The Cornflower line grows most rapidly and first overtakes all others in value, with an approximate average height of **35** across the chart.", **"justification":** "The Cornflower line was identified as the one growing most rapidly and surpassing others due to its steep slope and early lead. The average height calculation provides a quantitative measure of its performance throughout the chart."}

---

✅ *The correct response format*
**"reasoning_steps"**: [{**"step": 1**, "task": "Describe the scene in the chart to identify the trends of different colored lines", **"tool": "SceneDescriber"**, "tool_output": "The image is a line chart with 6 different colored lines.", "thought": "The SceneDescriber confirms the image is a line chart and identifies the color-coded lines, which is necessary to analyze their trends."}, {**"step": 2**, "task": "Extract the y-axis values to understand the numeric scale and support trend comparison.", **"tool": "OCR"**, "output": "{'text': [{'value': 'Cornflower'...}]", "thought": "The OCR extracts y-axis values to establish a numerical scale and matches line labels to their corresponding colors, enabling meaningful comparison of changes across lines."}, {**"step": 3**, "task": "Determine the minimum and maximum values for each line to calculate overall change.", **"tool": "SceneDescriber"**, "tool_output": "Cornflower ranges from 17 to 42. Bubblegum goes from 17 to 23. Orange Red keeps it tight from 18 to 17. Medium Orchid from 17 to 27. Orange shoots from 39 to 47. Brown starts at 17 and climbs to 32.", "thought": "The RegionDescriber retrieves the minimum and maximum values of each line to calculate their overall change."},
{**"step": 4, "task"**: "Estimate the average height of the fastest-growing line across the chart.", **"tool": "Calculator"**, "tool_output": "The average height of the Cornflower line is approximately **30** on the y-axis.", "thought": "The Calculator computes the change and average height of each line, allowing us to identify the one with the greatest overall change and report its average position on the chart."}],
**"final_answer":** {**"value":** "**Cornflower, 30**", **"justification":** "The line chart was analyzed to compare the overall change in each line. Cornflower showed the greatest change, and its average height was calculated to be approximately 30."}

---

❌ **Incorrect final answer**
• **Wrong tool call**
• **Wrong usage of the tool, CodePlotter is a generative tool that outputs the image of a plot**

Figure 6: Qualitative comparison of Qwen2.5 on visual reasoning tasks from Agent-X. Qwen2.5 often hallucinates tool behavior and produces overconfident justifications without numerical evidence. It struggles with temporal reasoning, overuses general-purpose tools like SceneDescriber, and fails to maintain consistency between reasoning steps and final answers.

Table 12: **Run-to-run variance analysis for (a) Qwen2.5-VL-7B, (b) InternVL3-8B, and (c) Gemini-2.5-Pro.**

| Run / Metric | $G_s$ | $T_p$ | $T_{acc}$ | $F_{acc}$ | $C_s$ | $F_p$ | $S_{acc}$ | $G_{acc}$ | $G_a^*$ | $T_{acc}^s$ |
|---|---|---|---|---|---|---|---|---|---|---|
| **(a) Qwen2.5-VL-7B** | | | | | | | | | | |
| Run 1 | 0.51 | 0.27 | 0.63 | 0.77 | 0.66 | 0.64 | 0.77 | 0.37 | 0.62 | 0.67 |
| Run 2 | 0.48 | 0.25 | 0.66 | 0.77 | 0.65 | 0.65 | 0.77 | 0.38 | 0.61 | 0.66 |
| Run 3 | 0.48 | 0.26 | 0.66 | 0.76 | 0.65 | 0.65 | 0.76 | 0.38 | 0.62 | 0.66 |
| Variance (%) | 0.020 | 0.0067 | 0.020 | 0.0022 | 0.0022 | 0.0022 | 0.0022 | 0.0022 | 0.0022 | 0.0022 |
| **(b) InternVL3-8B** | | | | | | | | | | |
| Run 1 | 0.41 | 0.16 | 0.51 | 0.71 | 0.61 | 0.60 | 0.69 | 0.23 | 0.51 | 0.62 |
| Run 2 | 0.36 | 0.15 | 0.50 | 0.62 | 0.50 | 0.62 | 0.56 | 0.20 | 0.51 | 0.52 |
| Run 3 | 0.36 | 0.15 | 0.50 | 0.62 | 0.50 | 0.62 | 0.56 | 0.20 | 0.50 | 0.52 |
| Variance (%) | 0.0556 | 0.0022 | 0.0022 | 0.1800 | 0.2689 | 0.0089 | 0.3756 | 0.0200 | 0.0022 | 0.2222 |
| **(c) Gemini-2.5-Pro** | | | | | | | | | | |
| Run 1 | 0.63 | 0.40 | 0.84 | 0.86 | 0.76 | 0.80 | 0.83 | 0.50 | 0.74 | 0.72 |
| Run 2 | 0.62 | 0.38 | 0.86 | 0.85 | 0.75 | 0.82 | 0.84 | 0.49 | 0.73 | 0.71 |
| Run 3 | 0.61 | 0.37 | 0.86 | 0.85 | 0.75 | 0.82 | 0.84 | 0.50 | 0.73 | 0.72 |
| Variance (%) | 0.0067 | 0.0156 | 0.0089 | 0.0022 | 0.0022 | 0.0089 | 0.0022 | 0.0022 | 0.0022 | 0.0022 |

Table 13: **Inter-judge agreement across *GPT-4o*, *Qwen-14B*, and *human judges*.**

| Statistic / Metric | $G_s$ | $T_{acc}$ | $F_{acc}$ | $S_{acc}$ | $G_{acc}$ |
|---|---|---|---|---|---|
| Pearson $(r)$ | 0.34 | 0.96 | 0.32 | 0.72 | **0.98** |
| Spearman $(\rho)$ | -0.07 | **0.99** | 0.39 | 0.63 | 0.96 |
| Mean Dev. | 0.13 | 0.05 | 0.09 | 0.11 | **0.04** |
| Cohen's $(\kappa)$ | 0.62 | 0.83 | 0.09 | 0.31 | **1.00** |

# G  ADDITIONAL RESULTS AND ANALYSIS

This section presents a comprehensive analysis of the proposed evaluation pipeline across multiple large multimodal models (`LMMs`) and judges. We report results from multi-run variance studies, inter-judge agreement, automated evaluation outcomes, and reasoning–perception ablations.

## G.1  RUN-TO-RUN VARIANCE ANALYSIS

To ensure the reliability of our automated evaluation setup, we perform a run-to-run consistency check on three representative models: *Qwen2.5-VL-7B*, *InternVL3-8B*, and *Gemini-2.5-Pro*. Table 12 summarizes the variance across three independent evaluation runs for each model. Overall, the variance remains minimal across all metrics, confirming the stability of our pipeline. For instance, *Qwen2.5-VL-7B* and *Gemini-2.5-Pro* both exhibit less than 0.02% deviation across metrics, whereas *InternVL3-8B* shows slightly higher fluctuation (up to 0.38%) in semantic accuracy ($S_{acc}$). A high degree of determinism is therefore ensured across runs. A consolidated summary in Table 14 further highlights that *Gemini-2.5-Pro* and *Qwen2.5-VL* achieve near-identical results across trials, confirming that the observed differences occur only beyond the third decimal place. *InternVL3-8B* exhibits minor instability, attributed to its stochastic sampling process during decoding, but the overall ranking remains consistent across repetitions.

## G.2  INTER-JUDGE AGREEMENT

To assess the robustness and reliability of our evaluation framework, we compare the judgments produced by three distinct evaluators: *GPT-4o*, *Qwen-14B*, and human experts. Table 13 presents four complementary agreement statistics, Pearson correlation $(r)$, Spearman rank correlation $(\rho)$, mean deviation, and Cohen's $\kappa$, computed across five representative metrics ($G_s$, $T_{acc}$, $F_{acc}$, $S_{acc}$, and $G_{acc}$).

Table 14: **Summary of multi-run variance analysis using *Qwen-14B* as judge.**

| Model | Max Variance (%) | Observation |
|---|---|---|
| Qwen2.5-VL | 0.02 | Stable across all metrics; differences appear only in the 3$^{rd}$ decimal place. |
| InternVL3-8B | 0.38 | Slightly higher fluctuation in semantic accuracy, but consistent overall ranking. |
| Gemini-2.5-Pro | 0.02 | Practically identical scores across runs, confirming evaluation determinism. |

**High Alignment with Human Judgments.** The results reveal remarkably strong consistency between automated judges and human evaluators, particularly in task accuracy ($T_{acc}$) and global accuracy ($G_{acc}$). Both dimensions exhibit near-perfect correlations ($r = 0.96$, $\rho = 0.99$ for $T_{acc}$ and $r = 0.98$, $\kappa = 1.00$ for $G_{acc}$), suggesting that *Qwen-14B* and *GPT-4o* are capable of replicating human evaluative behavior with high fidelity. These metrics primarily assess correctness and goal achievement, objectively quantifiable properties that leave little room for subjective interpretation. Hence, the high agreement indicates that automated judges can reliably approximate human consensus for structured, outcome-based criteria.

**Moderate Agreement in Semantic and Functional Dimensions.** For semantic accuracy ($S_{acc}$) and functional accuracy ($F_{acc}$), the correlations remain moderately high ($r = 0.32$–$0.72$), reflecting partial but imperfect alignment. These metrics rely on nuanced linguistic and reasoning cues, such as whether model responses maintain contextual consistency or functional correctness, where even expert human raters may differ in interpretation. The observed variance thus highlights the intrinsic ambiguity of multimodal reasoning evaluation rather than a deficiency in the automated judges.

**Lower Agreement in Goal Success.** The goal success metric ($G_s$) demonstrates lower correlation values ($r = 0.34$, $\rho = -0.07$), revealing that high-level subjective goals are more difficult to standardize across evaluators. This finding underscores the challenge of quantifying "success" in open-ended, multimodal tasks, where different judges may focus on distinct aspects such as reasoning depth, visual grounding, or task completion. It also motivates the need for multi-perspective evaluation frameworks that combine quantitative accuracy with qualitative interpretability.

**Key Insights.** Overall, the results establish that **automated evaluation using *Qwen-14B* achieves near-human reliability** across most quantitative dimensions. The strongest alignment occurs for metrics grounded in explicit correctness (e.g., $T_{acc}$ and $G_{acc}$), demonstrating that LLM-based judges are highly effective at assessing factual and structural validity. However, metrics involving contextual, semantic, or open-ended reasoning still benefit from human oversight. This reinforces the complementary nature of human–AI evaluation synergy: while automated judges provide scalability and consistency, expert human review remains crucial for capturing subtle interpretive aspects of reasoning quality.

### G.3 AUTOMATED EVALUATION ACROSS MODELS

We benchmark a wide range of vision–language models using our automated judge, *Qwen-14B*. As shown in Table 15, *GPT-4o* achieves the highest overall performance, with strong results in task accuracy ($T_{acc} = 0.45$) and functional accuracy ($F_{acc} = 0.34$), followed by *Qwen2.5-7B-Instruct* and *InternVL3-8B*. The baseline *Qwen1.5-7B-Chat* performs significantly lower across all metrics, highlighting the benefits of multimodal fine-tuning. Interestingly, *Qwen2.5-VL* performs comparably on low-level metrics such as $C_s$ and $F_p$ but lags in high-level goal accuracy ($G_{acc}$). This reinforces that pure visual grounding alone is insufficient for complex multimodal reasoning.

### G.4 REASONING VS. PERCEPTION ANALYSIS

To isolate and quantify the contribution of reasoning to multimodal task performance, we conduct an ablation study on a representative subset of the dataset. Specifically, we compare two configurations: (1) a **perception-only** model that relies solely on visual understanding

Table 15: **Automated evaluation results across models on *Agent-X.***

| Model | Inst$_{align}$ | T$_{acc}$ | Arg$_{acc}$ | Sum$_{acc}$ | F$_{acc}$ | C$_s$ | F$_p$ | S$_{acc}$ | G$_{acc}$ | G$_{star}$ |
|---|---|---|---|---|---|---|---|---|---|---|
| GPT-3.5-Turbo | 0.65 | 0.39 | 0.11 | 0.12 | 0.34 | 0.27 | 0.29 | 0.49 | 0.33 | 0.30 |
| GPT-4o | **0.72** | **0.45** | **0.15** | 0.12 | **0.34** | **0.31** | **0.36** | **0.51** | **0.36** | **0.35** |
| Qwen1.5-7B-Chat | 0.24 | 0.09 | 0.04 | 0.12 | 0.09 | 0.02 | 0.08 | 0.35 | 0.34 | 0.25 |
| Qwen2.5-7B-Instruct | 0.46 | 0.29 | 0.02 | **0.13** | 0.15 | 0.09 | 0.10 | 0.39 | 0.33 | 0.26 |
| Qwen2.5-7B-VL | 0.42 | 0.20 | 0.03 | 0.08 | 0.14 | 0.09 | 0.18 | 0.32 | 0.01 | 0.01 |
| InternVL3-8B | 0.49 | 0.28 | 0.04 | 0.12 | 0.14 | 0.09 | 0.13 | 0.41 | 0.02 | 0.02 |

without explicit reasoning steps, and (2) our proposed **reasoning-augmented** model that incorporates structured reasoning chains during evaluation. As shown in Table 16, the inclusion of reasoning improves the overall goal accuracy ($G_{acc}$) from 0.33 to 0.43, a relative gain of approximately 30%.

This improvement reveals that reasoning plays a crucial role in bridging low-level perception with high-level task understanding. While perception-only models can identify visual entities and basic relationships, they often fail in multi-hop inference, causal understanding, or context-dependent decision making. In contrast, models equipped with explicit reasoning chains can decompose complex agentic queries into interpretable intermediate steps, verify intermediate evidence, and better align their final predictions with the intended goals.

These findings validate our design choice of incorporating reasoning-centric evaluation protocols, as they more accurately reflect an agent's true multimodal competence rather than surface-level visual recognition. In other words, the ability to reason explicitly, not just perceive, is what allows multimodal systems to achieve coherent, explainable, and goal-directed behavior across diverse scenarios.

Table 16: **Reasoning vs. Perception Ablation on Subset.**

| Setting | G$_{acc}$ |
|---|---|
| Perception-only (no explicit reasoning) | 0.33 |
| Full reasoning chain (ours) | **0.43** |

### G.5 Discussion

Our analyses collectively establish three key insights:

1. The evaluation pipeline is **deterministic and consistent** (Tables 12–14), ensuring reproducibility across runs.

2. Automated judges like *Qwen-14B* show **strong agreement with human evaluations** (Table 13), supporting scalable benchmarking.

3. Explicit reasoning significantly contributes to **higher multimodal task performance** (Table 16), emphasizing the importance of reasoning supervision in multimodal agents.

### G.6 Reasoning Chain Experiments

**Goal Accuracy vs. Question Difficulty.** To better understand model performance across varying levels of task complexity, we categorize queries into *easy* and *hard* subsets using a difficulty classifier based on LMM confidence scores and step trace complexity. As shown in Figure 7, all models perform better on easy queries, with a noticeable drop in Goal Accuracy for harder examples. For instance, Qwen2.5 drops from 39% to 31% and InternVL3 from 28% to 14%, indicating that model reliability significantly declines as reasoning difficulty increases. Gemini-2.5 performs the strongest overall, suggesting better robustness across query difficulty levels.

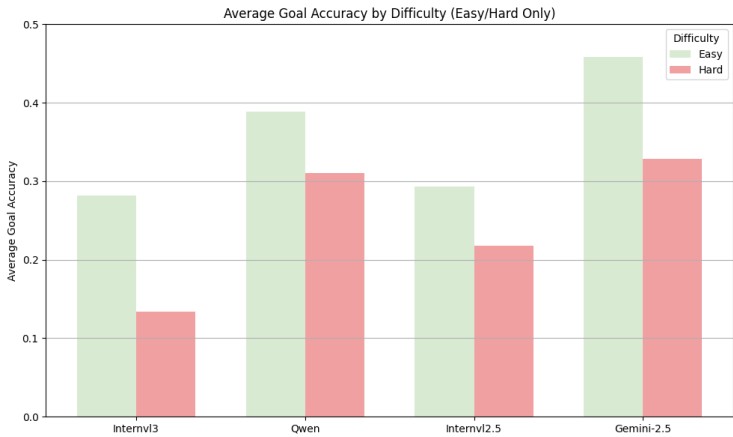

Figure 7: Average `Goal Accuracy` by difficulty level across models.

**Goal Accuracy vs. Reasoning Depth.** We also analyze how the number of reasoning steps (`chain length`) affects task success. Figure 8 shows that deeper reasoning traces (i.e., 5-6 steps) generally correlate with higher or more stable `Goal Accuracy` for strong models like `GPT-4o-mini` and `Qwen`, suggesting their improved ability to handle multi-hop reasoning. In contrast, performance for `InternVL3` and `VideoLLaMA3` dips at deeper depths, exposing their limited capacity for long-horizon reasoning. These findings underscore the importance of both depth-aware reasoning and intermediate step consistency.

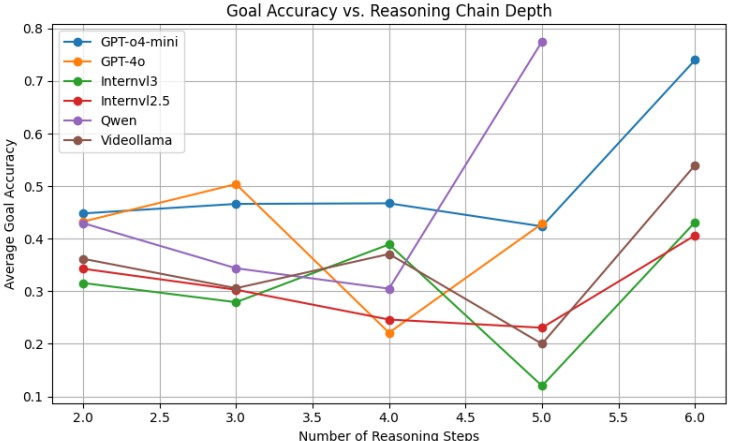

Figure 8: `Goal Accuracy` vs. reasoning chain depth.

**Tool Call Success and Failure Analysis.** Figure 9 shows the number of successful and failed tool invocations per model. A successful call executes without format or argument errors. The GPT family achieves the highest success rate (83.8%) relative to its total calls, with GPT-4o demonstrating a strong balance between usage and reliability. InternVL models lead to absolutely successful calls but exhibit high failure counts, primarily due to invalid tool usage. Qwen2.5-VL-8B shows the highest precision, with only 109 failures out of 2241 calls, reflecting robust format compliance. In contrast, o4-mini is the most aggressive, issuing 3374 calls but suffering the highest failure rate (1038). Gemini-

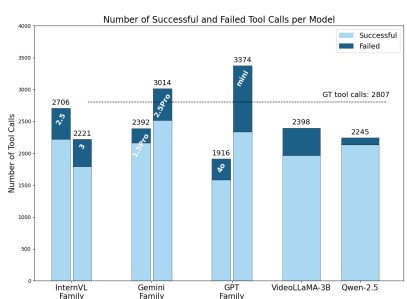

Figure 9: Tool Call analysis across LMM families on Agent-X.

3-4B and VideoLLaMA3-7B show moderate tool use,
with Gemini-3-4B maintaining a lower failure count. These trends highlight three behavioral
patterns: *aggressive* models (*e.g.*, InternVL3-8B) invoke tools frequently but are prone to
errors; *conservative* models (*e.g.*, Qwen2.5-VL-7B) prioritize accuracy over volume; and
*balanced* models (*e.g..*, GPT-4o) combine moderate tool use with high reliability.

## H    INSTRUCTION FOR ANNOTATORS

---

**Query Construction Guidelines**

**Input Selection Rules:**

- Select inputs (images or videos) from publicly available datasets to ensure broad coverage of real-world scenarios.
- Include diverse environments such as sports, autonomous driving, web browsing, surveillance footage, etc.
- Use image files that range from clean to cluttered scenes to test for occlusions and visual complexity.
- For video files, ensure that there is meaningful variation or motion across frames (e.g., object movement or temporal change).

**Query Design Requirements:**

- Design queries that require at least **three** distinct reasoning steps.
- Each query must involve at least **two different tools** from the provided tool list (Table 6).
- Avoid trivial or single-glance questions; queries must require reasoning grounded in the input.
- Do not mention tool names explicitly in the query. *(e.g., "Use OCR to read the sign" is not allowed).*
- If referencing dynamic or time-sensitive information, specify a fixed date, time frame, or source.
- For multi-image queries, ensure that the question requires reasoning across the images and not individually.
- For video queries, ensure that reasoning involves content across multiple frames.

**Language and Verification Rules:**

- Write all queries and answers in English.
- Ensure that each answer can be verified consistently by human evaluators.
- Avoid ambiguous questions that could lead to inconsistent answers.

---

Figure 10: Annotation instruction document used during the query construction stage.

---

**Reasoning Trace and Final Answer Annotation Guidelines**

**Your Task:**
You are provided with a set of image- or video-based queries along with tool metadata. Your job is to verify and improve the reasoning steps, final answer, and justification for each sample. Ensure that the steps are logical, human-like, and grounded in the given tools and inputs.

**Reasoning Trace Requirements:**
- Each reasoning trace must contain at least three distinct steps.
- Use at least two different tools from the provided tool list.
- Each step should include:
  - **Thought:** What the agent is trying to figure out.
  - **Tool:** The tool being used.
  - **Input:** What is passed to the tool (e.g., image name, region, text).
  - **Output:** What the tool returns.
- Steps should form a coherent, human-like progression. Each step should follow logically from the previous.
- Ensure tool use is appropriate and consistent with its defined capabilities.
- Do not invent outputs: use plausible results based on the image and tool functionality.

**Final Answer and Justification Guidelines:**
- The final answer must be consistent with the reasoning steps.
- For objective queries, the answer should be accurate, verifiable, and clearly correct.
- For subjective or generative queries, the answer should be reasonable, coherent, and grounded in the input.
- Avoid vague or incomplete answers: the answer should fully resolve the query.
- The justification should summarize the reasoning in clear natural language.

**Annotation Rules:**
1. Do not edit or rewrite the query. Focus only on the reasoning steps, answer, and justification.
2. Use fluent, human-like language for the thoughts and justifications.
3. Make sure each reasoning chain is clear, logically structured, and not overly long.
4. Flag samples if the image, video, or query is ambiguous, irrelevant, or unsolvable.

Figure 11: Instruction document for reasoning step and answer annotation.

# I  AGENT-X TASK EXAMPLES

---

**Example 1**

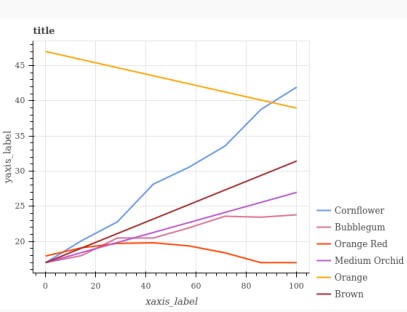

AgentX_176.jpg

**Query:** Which color line shows the greatest overall change in the y-direction and approximately what is its average height across the chart?

**Involved Tools:** `SceneDescriber`, `OCR`, `RegionDescriber`, `Calculator`

**Steps:**
1. Describe the scene in the chart to identify the trends of the different colored lines.
2. Extract the y-axis values to understand the numeric scale and support trend comparison.
3. Determine the minimum and maximum values for each line to calculate overall change.
4. Estimate the average height of the fastest-growing line across the chart.

**Reasoning Trace for steps:**
1. The SceneDescriber confirms the image is a line chart and identifies the color-coded lines, which is necessary to analyze their trends.
2. The OCR extracts y-axis values to establish a numerical scale and matches line labels to their corresponding colors, enabling meaningful comparison of changes across lines.
3. The RegionDescriber retrieves the minimum and maximum values of each line to calculate their overall change.
4. The Calculator computes the change and average height of each line, allowing us to identify the one with the greatest overall change and report its average position on the chart.

**Answer:** Cornflower, 30

**Justification:** The line chart was analyzed to compare the overall change in each line. Cornflower showed the greatest change, and its average height was calculated to be approximately 30.

---

Figure 12: Example 1

**Example 2**

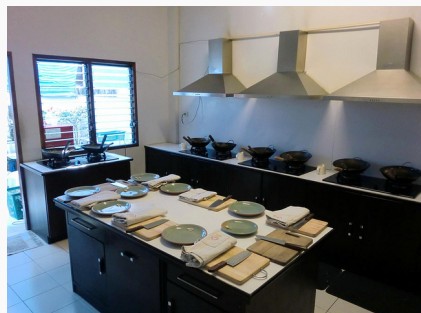

`AgentX_100.jpg`

**Query:** Is this kitchen more likely set up for a cooking class or for a private dinner?

**Involved Tools:** `SceneDescriber`, `ObjectCounter`, `LocateObjectByText`

**Steps:**

1. Describe the scene to understand the overall layout and setting of the kitchen.
2. Count the number of cooking stations or setups visible.
3. Examine the arrangement and positioning of kitchen equipment.
4. Assess the kitchen's design and features to determine its suitability for a cooking class or private dinner.

**Reasoning Trace for steps:**

1. The SceneDescriber tool highlights multiple cooking stations, suggesting a layout intended for simultaneous use by several individuals, typical of a cooking class setup.
2. The ObjectCounter tool confirms the presence of seven stovetops, supporting the idea that the kitchen is designed for group-based cooking activities rather than a private dinner.
3. The LocateObjectByText tool is used to find specific items like cutting boards in the image, showing that each station is individually equipped—reinforcing the setup's suitability for a cooking class.
4. The overall utilitarian and organized layout, as observed through the SceneDescriber, indicates the space is structured for instructional use or demonstrations, not for a private dining experience.

**Answer:** A cooking class.

**Justification:** The kitchen features multiple cooking stations, including seven stovetops and individual cutting boards, all arranged in a structured layout. This setup is characteristic of a cooking class environment, where several people cook simultaneously, rather than a setting intended for a private dinner.

Figure 13: Example 2

**Example 3**

```
x = np.outer(np.linspace(-3, 3, 32), np.ones(32))

y = x.copy( ).T

z = np.sin(x ** 2) + np.cos(y ** 2)
```

AgentX_212.1.jpg

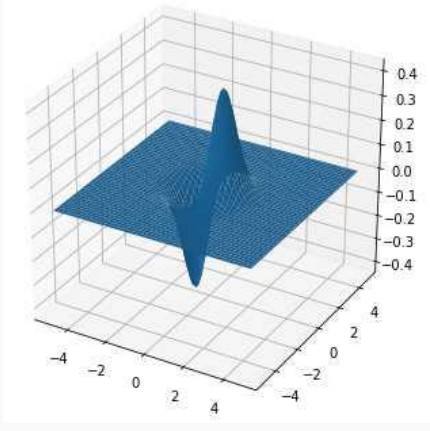

AgentX_212.2.jpg

**Query:** What is the highest point visible on the plotted surface, and where does it occur in the 3D space?

**Involved Tools:** `RegionDescriber`, `MathOCR`, `Solver`

**Steps:**

1. Describe the region corresponding to the peak of the surface plot to determine approximate x and y values.
2. Extract the equation for the surface to calculate the exact value of z.
3. Compute the z-value at the identified (x, y) coordinates.

**Reasoning Trace for steps:**

1. The RegionDescriber tool is used to analyze the highest point on the plotted surface. This helps estimate the corresponding x and y coordinates, which are visually identified as approximately (2,3).
2. The MathOCR tool extracts the surface equation $z = sin(x^2) + cos(x^2)$ from the image. This is needed to compute the exact value of z at the identified location.
3. The Solver tool evaluates the equation at x=2 and y=3, yielding a computed z-value of approximately 0.24. This confirms the height of the highest visible point in the 3D space.

**Answer:** The highest point on the plotted surface occurs at coordinates (x, y, z) = (2.0, 0.0, 0.24).

**Justification:** The highest point was visually located at (2,3). Using the extracted equation, the z-value at this point was computed to be approximately 0.24, confirming its position in the 3D space.

Figure 14: Example 3

---

**Example 4**

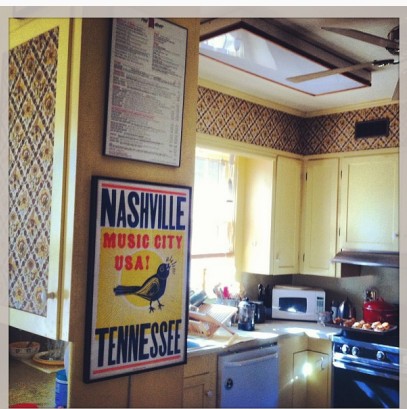

`AgentX_0.jpg`

**Query:** What kind of room is the poster located in, and what does it say?

**Involved Tools:** `SceneDescriber`, `LocateObjectByText`, `OCR`

**Steps:**

1. Describe the scene to determine what type of room is shown.
2. Locate the poster within the scene to enable reading its content.
3. Extract text from the poster in the scene.

**Reasoning Trace for steps:**

1. The SceneDescriber tool identifies typical kitchen elements like cabinets, a sink, a dish rack, a microwave, and a stove, which indicates that the room is a kitchen.
2. The LocateObjectByText tool helps determine where the poster is in the image, allowing me to see what is written on it.
3. The OCR tool allows extraction of the visible text from the poster, making it possible for me to read what it says.

**Answer:** The poster is in a **kitchen** and reads: **"NASHVILLE MUSIC CITY USA! TENNESSEE."**

**Justification:** Key kitchen elements such as a microwave, oven, stove, sink, and cabinets were identified, confirming the room is a kitchen. The poster's position in the image was located, and its text was extracted, allowing the content to be read and verified.

Figure 15: Example 4

---

**Example 5**

$$(x - \sqrt{2})^2 + 4\sqrt{2}\,x = 0$$

AgentX_133.jpg

**Query:** What is the solution to the equation, and is the solution real, complex, or integer?

**Involved Tools:** `MathOCR, Solver, Calculator`

**Steps:**

1. Extract the mathematical expression from the image.
2. Solve the extracted equation to find the values of x
3. Analyze the solutions to determine whether they are real, complex, or integers.

**Reasoning Trace for steps:**

1. The MathOCR tool extracts the equation $(x - sqrt(2))^2 + 4 * sqrt(2) * x = 0$ from the image, enabling symbolic processing.
2. The Solver tool is used to solve the equation, yielding the solution $x = -sqrt(2)$.
3. The Calculator confirms that the value $x = -sqrt(2)$ is a real number, which is irrational but not complex or integer.

**Answer:** $x = -sqrt(2)$. Real

**Justification:** The equation was extracted, solved, and evaluated. There is just one solution of this equation. The resulting value is real and irrational, not complex or integer.

Figure 16: Example 5

**Example 6**

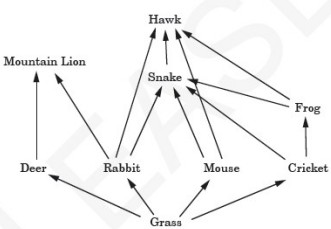

`AgentX_555.jpg`

**Query:** How many species does the top predator prey on?

**Involved Tools:** `OCR`, `SceneDescriber`, `ObjectCounter`

**Steps:**

1. Extract all text from the image to identify species and connections.
2. Analyze the food web structure to find species with no outgoing arrows — indicating top predators.
3. Count the number of incoming arrows to the top predator to determine how many species it preys on.

**Reasoning Trace for steps:**

1. Reading the species names helps establish which organisms are part of the food web and which ones are connected by feeding relationships.
2. Analyzing the direction of arrows reveals the top predators—species with no outgoing arrows—indicating that nothing preys on them. In this case, the Hawk and Mountain Lion meet that condition.
3. Counting the number of arrows pointing to each top predator shows how many species they prey on: four for the Hawk and two for the Mountain Lion.

**Answer:** The top predators in the food web are the Hawk and the Mountain Lion, which prey on four and two species, respectively.

**Justification:** The Hawk and Mountain Lion are top predators with no outgoing arrows. Counting the arrows pointing to each confirms that the Hawk preys on four species and the Mountain Lion on two.

Figure 17: Example 6

**Example 7**

x_values = [1, 2, 3, 4, 5]

y_values = [2, 3, 5, 7, 11]

bubble_sizes = [30, 80, 150, 200, 300]

`AgentX_339.1.jpg`

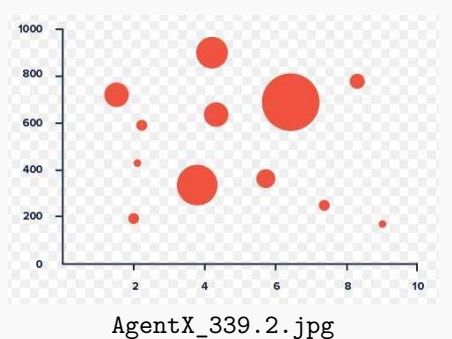

`AgentX_339.2.jpg`

**Query:** Using the provided data, verify the x, y, and size values against the bubble chart. Identify if there are any discrepancies in the data representation, and plot a corrected version of the chart if necessary.

**Involved Tools:** `OCR`, `SceneDescriber`, `LocateObjectByText`, `CodePlotter`

**Steps:**

1. Extract the lists of x-values, y-values, and bubble sizes from the data.
2. Analyze the layout and structure of the bubble chart to understand how the data is represented visually.
3. Compare the extracted values with the chart to identify any mismatches in position or size.
4. Generate a corrected version of the chart using the accurate data values.

**Reasoning Trace for steps:**

1. The OCR tool is used to extract the lists of x_values, y_values, and bubble_sizes from the data, which are needed to verify the accuracy of the plotted chart.
2. The SceneDescriber helps interpret the layout of the bubble chart, including axis scales and the overall distribution of points, which provides context for comparison.
3. By visually inspecting the chart and referencing the extracted values, a mismatch is observed between the actual positions or sizes of the bubbles and what the data specifies.
4. Re-plotting ensures that the visual representation now matches the extracted x, y, and bubble sizes to produce an accurate visual representation.

**Answer:** $\phi$

**Justification:** The extracted x, y, and size values did not match the visual chart. Re-plotting the data corrected these discrepancies, ensuring the chart accurately reflects the provided values.

Figure 18: Example 7

**Example 8**

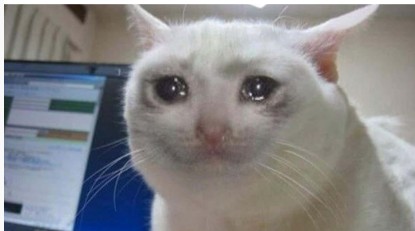

`AgentX_534.jpg`

**Query:** What emotion is being depicted in the image? What is the meme about?

**Involved Tools:** `SceneDescriber`, `OCR`, `RegionDescriber`

**Steps:**

1. Describe the scene in the image to identify the depicted emotion.
2. Extract text from the image to understand the context of the meme.
3. Analyze combined information to conclude the meme's theme.

**Reasoning Trace for steps:**

1. The scene description helps identify that the cat appears sad, indicating the emotion of sadness or disappointment.
2. The text provides context for the image, explaining why the depicted emotion is sadness or disappointment.
3. Combining the cat's sad expression with the text content, the meme humorously conveys the disappointment of a common online interaction.

**Answer:** The emotion depicted is sadness. The meme is about the disappointment felt when a friend goes offline right after receiving a message.

**Justification:** The steps provided clarity on the cat's expression (sadness) and the textual context (meme scenario), confirming the theme of disappointment.

Figure 19: Example 8

---

**Example 9**

```python
import matplotlib.pyplot as plt

# the slices are ordered and
# plotted counter-clockwise:
continents = ['Asia', 'Europe', 'North America',
              'South America','Australia',
              'Africa','Antarctica']

area = [25, 20, 15, 10,15,10,5]
explode = (0.1, 0, 0.1, 0,0.1,0.1,0.1)
```

AgentX_340.1.jpg

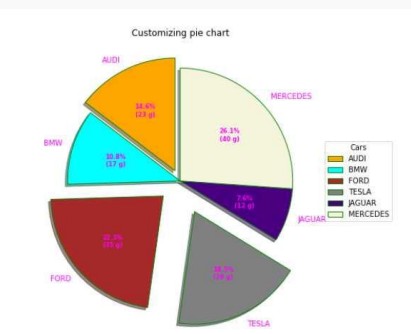

AgentX_340.2.jpg

**Query:** Based on the provided code snippet and the pie chart, which continent represents the largest area from the given data, and which car brand represents the largest percentage in the pie chart? Calculate the difference between these two percentages.

**Involved Tools:** MathOCR, Calculator, OCR, SceneDescriber

**Steps:**

1. Identify the continent with the largest area from the code snippet.
2. Calculate the percentage that this area represents out of the total.
3. Analyze the pie chart to identify the largest slice in the pie chart.
4. Extract the percentage value and brand name associated with that slice.
5. Calculate the difference between the largest area percentage and the largest pie chart percentage.

**Reasoning Trace for steps:**

1. To determine which continent dominates in size, the area values from the code need to be examined so that we can use the correct value for comparison.
2. Calculating the percentage of the largest area provides a normalized value, making it possible to directly compare it with percentages shown in the pie chart.
3. The SceneDescriber helps interpret the pie chart structure, which is necessary to visually identify the largest share based on the relative size of each slice.
4. The OCR tool is used to confirm the brand and exact percentage of the largest slice, ensuring that the visual estimate is backed by precise data.
5. Finding the difference between the two percentages answers the core question by quantifying how the largest values from each source compare.

**Answer:** Asia, Mercedes, 1.1%

**Justification:** The continent with the largest area is Asia (25%), and the car brand with the largest percentage is Mercedes (26.1%). The difference between these two percentages is 1.1%.

Figure 20: Example 9

**Example 10**

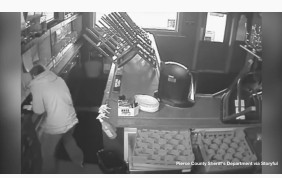 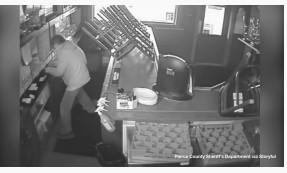 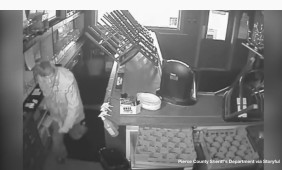

**AgentX__630__6.jpg**        **AgentX__630__30.jpg**        **AgentX__630__54.jpg**

**Query:** Is the person in the video an employee?

**Involved Tools:** `SceneDescriber`, `RegionDescriber`

**Steps:**

1. Identify the type of location shown in the video.
2. Determine whether the setting appears open to customers or closed.
3. Observe the placement of objects for operational context.
4. Analyze the person's behavior and interaction with the environment.
5. Consider whether the actions match expected employee routines.
6. Decide if the person's presence and behavior indicate they are an employee.

**Reasoning Trace for steps:**

1. Identifying the place as a restaurant or café sets clear expectations for how staff typically behave.
2. If the place appears closed, anyone inside should either be staff or unauthorized.
3. The chairs placed upside down on tables indicate the location is not open to customers.
4. The person's cautious movement and drawer searching suggest they are not performing routine tasks.
5. Staff usually act with purpose, doing things like cleaning or closing up, not wandering or poking around
6. The closed setting and unusual behavior make it unlikely that the person is an employee.

**Answer:** No

**Justification:** The restaurant appears closed, and the person's behavior—cautiously searching through drawers—does not match typical staff duties. This suggests they are likely not an employee.

Figure 21: Example 10

**Example 11**

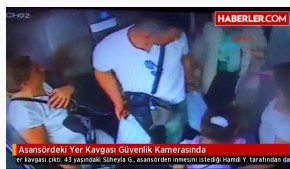 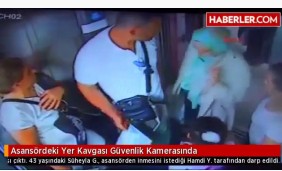 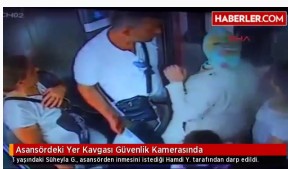

**AgentX__358__5.jpg**  **AgentX__358__25.jpg**  **AgentX__358__45.jpg**

**Query:** How many people are entering the elevator?

**Involved Tools:** `SceneDescriber`, `RegionDescriber`, `ObjectCounter`

**Steps:**

1. Understand the overall context of the scene and confirm that the setting involves an elevator.
2. Analyze the body positions and movement direction of individuals across the frames to identify who is entering the elevator.
3. Count the number of people who move into the elevator space.

**Reasoning Trace for steps:**

1. Recognizing the environment as an elevator area helps set expectations for how people should move relative to the door.
2. Analyzing the region near the elevator door across frames shows one woman stepping forward into the space, while others remain stationary or are already inside.
3. Counting only the individuals who cross the threshold into the elevator shows that one person enters.

**Answer:** 1

**Justification:** A woman is moving into the elevator while others remain in place or are already inside, confirming that only one person is entering.

Figure 22: Example 11

## J  GENERATION PROMPTS

---

**Query Generation**

"You are an annotator tasked with generating a realistic and verifiable user query for benchmarking a multimodal LMM-based assistant.

You are provided with the following tools the assistant can use:

tools = [ "ImageDescription", "CountGivenObject", "OCR", "DrawBox", "Calculator", "DetectGivenObject", "RegionAttributeDescription", "MathOCR", "Solver", "Plot", "GoogleSearch", "TextToImage", "AddText", "ImageStylization" ]

Your job is to generate one single complex and human-evaluable user query that satisfies all of the following:
- If you are given multiple images, the query must require reasoning across multiple images (not just one)
- If you are given multiple frames, the query must involve comparing, tracking, or analyzing across multiple video frames.
- The query must require at least 3 distinct reasoning steps to be answered.
- It must require at least 2 different tools from the list above.
- Do not mention tool names explicitly.
- If the query involves online or time-sensitive content, include a fixed timeframe or source.
- The query must be realistic and grounded in common user intentions.
- Use only English.
- All queries must be suitable for evaluation by a human - the answer should not vary arbitrarily across individuals.

Return your result as a single JSON object on one line.
{{ "input":  {file_repr}, "query":  "<natural language query>" }}"

---

Figure 23: Prompt given to GPT4-o to for the initial query generation.

---

**Reasoning and Final Answer**

"You are an agent performing multimodal reasoning using the tools listed
below.

Your goal is to answer the given query step-by-step by selecting the right
tools for each stage of reasoning.

For each step:

- Clearly state what you're trying to do.
- Specify the tool you are using.
- Explicitly include the input provided to the tool and the output
  received from the tool.
- Explain what you learned or why this step was necessary.

Make sure your tool usage follows the tool descriptions provided.  Do not
hallucinate tools or skip intermediate reasoning steps.

--

Available Tools & How to Use Them:
{toolmeta_section}

--

Query: "{query}"

Images: {image1, image2, ...}

--

Return your output as a JSON list of steps, formatted like this:

```
 [ {
"step":  <step number>,
"task":  "<Short description of what this step is trying to do>",
"tool":  "<Tool name(s)>",
"input":  "<Input provided to the tool>",
"output":  "<Output from the tool>",
"thought":  "<Why this step was needed, or what you learned>"
},
...
{
"final_answer":  {
"value":  "<Final conclusion>",
"justification":  "<How all steps lead to the answer>"
} } ]"
```

Figure 24: Prompt given to GPT4-o for the initial reasoning and final answer generation.

# K  Evaluation Prompts

---

**Grounding Score**

You are an evaluation assistant measuring the groundedness of each reasoning
step performed by a vision-language agent.

"You are given a Ground Truth (GT) block containing the original query,
a sequence of reasoning steps, and the final answer along with its
justification.  Each reasoning step includes the task being attempted, the
tool used (with its input and output), and the agent's thought process.

You are also given the full reasoning trace produced by the agent, including
each step's task, selected tool and its output, and thought.

Your task is to assess whether each agent step is visually and contextually
grounded - that is, whether the task, the selected tool, the tool output, and
the agent's thought process are all supported by the GT.

Evaluation Guidelines:

- If the number of steps in the GT and the model differ:
    - Any extra model step beyond the GT steps should receive a score
      of 0 (hallucinated or unjustified).
    - Any GT step that the model omits should also receive a score of 0
      (missing reasoning).
- The Grounding Score for the full query is computed as the average of
  the per-step scores.

Assess the following per step:

- Is the task relevant to the query and aligned with the GT?
- Is the tool appropriate for this task?
- Is the output consistent with what's in the GT?
- Is the thought grounded and logically aligned?

Scoring Criteria:

- 1 - All aspects are grounded (task, tool, output, and thought)
- 0.5 - Partially grounded (some align, some don't)
- 0 - Ungrounded or hallucinated

Output format:
Score:  <0, 0.5, or 1>
Justification:  <1-2 sentence explanation for the score>"

---

Figure 25: Prompt for computing the grounding score of reasoning steps produced by
vision-language agents.

---

**Tool Precision**

"You are an evaluation assistant measuring tool precision in an agent's
reasoning step.

You are given a Ground Truth (GT) block containing the original query,
a sequence of reasoning steps, and the final answer along with its
justification. Each reasoning step includes the task being attempted, the
tool used (with its input and output), and the agent's thought process.

You are also given the full reasoning trace produced by the agent, including
each step's task, selected tool and its output, and thought.

Your task is to assess whether the tool selected by the agent in each
reasoning step is the most appropriate tool for the task, by comparing it
with the GT.

Evaluation Guidelines:

- If the number of steps in the GT and the model differ:

    - Any extra model step beyond the GT steps should receive a score
      of 0 (hallucinated or unjustified).
    - Any GT step that the model omits should also receive a score of 0
      (missing reasoning).

- The Tool Precision score for the full query is computed as the
  average of the per-step scores.

Assess the following per step:

- Is the tool selected by the agent the same as the one used in the GT?
- Is the tool the most appropriate for the task being attempted?

Scoring Criteria:

- 1 - The selected tool matches the GT tool and is appropriate for the
  task.
- 0 - The tool does not match the GT or is an inappropriate choice.

Output format:
Score: <0 or 1>
Justification: <1-2 sentence explanation for the score>"

---

Figure 26: Prompt for assessing whether the tools selected by the agent in each step match
the ground truth.

**Tool Accuracy**

"You are an evaluation assistant measuring tool accuracy in an agent's reasoning step.

You are given the task the agent is trying to accomplish, the tool it uses along with its input and output, and the tool metadata containing a description of the tool's purpose and its expected input/output formats.

Your task is to assess whether the tool used by the agent was applied correctly in each step by checking:

- Whether the output format is valid and consistent with the tool's specification (based on tool metadata).
- Whether the output is relevant and meaningful for completing the step's task.

Evaluation Guidelines:

- The Tool Accuracy score for the full query is computed as the average of the per-step scores.

Assess the following per step:

- Is the tool's output correctly formatted according to the tool metadata?
- Is the output meaningful and appropriate for the stated task?

Scoring Criteria:

- 1 – Output is valid, properly formatted, and clearly relevant to the task.
- 0.5 – Output is partially valid or partially relevant.
- 0 – Output is incorrectly formatted, irrelevant, or unhelpful.

Output format:
Score: <0, 0.5, or 1>
Justification: <1-2 sentence explanation for the score>"

Figure 27: Prompt for verifying correct tool usage and output formatting based on metadata.

---

**Faithfulness Accuracy**

"You are an evaluation assistant measuring the Faithfulness Accuracy of an agent's reasoning process.

You are given a Ground Truth (GT) block containing the original query, a sequence of reasoning steps, and the final answer along with its justification. Each reasoning step includes the task being attempted, the tool used (with its input and output), and the agent's thought process.

You are also given the full reasoning trace produced by the agent, including each step's task, selected tool, output, and thought.

Your task is to assess how faithful the agent's reasoning trace is to the GT, i.e., whether the steps follow a logically sound plan that aligns with the structure, intent, and direction of the GT.

Evaluation Guidelines:

- Focus on the structure and logical flow of the agent's reasoning steps.
- Determine whether the steps collectively form a coherent strategy to answer the query.
- Faithfulness is about consistency with the GT's method, not necessarily correctness of individual steps.

Scoring Criteria:

- 1 – The reasoning is faithful to the GT: it follows a logically sound plan that mirrors the GT in structure and direction.
- 0.5 – The reasoning partially follows the GT's structure, but contains some deviations, redundancies, or inconsistencies.
- 0 – The reasoning is not faithful to the GT and lacks logical progression or alignment with the intended plan.

Output format:
Score:  <0, 0.5, or 1>
Justification:  <1-2 sentence explanation for the score>"

---

Figure 28: Prompt used to evaluate the structural faithfulness of agent reasoning to the ground truth.

---

**Context Score**

"You are an evaluation assistant measuring the Context Score of an agent's reasoning step.

You are given a Ground Truth (GT) block containing the original query, a sequence of reasoning steps, and the final answer along with its justification. Each reasoning step includes the task being attempted, the tool used (with its input and output), and the agent's thought process.

You are also given the full reasoning trace produced by the agent, including each step's task, selected tool and its output, and thought, as well as the agent's final answer and justification.

Your task is to assess whether the agent's reasoning step is grounded in the input context, and whether that context was effectively used in the reasoning process. Inputs may include image, video, text, audio, or a combination.

Evaluation Guidelines:

- If the number of steps in the GT and the model differ:
  - Any extra model step beyond the GT steps should receive a score of 0 (hallucinated or unjustified).
  - Any GT step that the model omits should also receive a score of 0 (missing reasoning).
- The Context Score for the full query is computed as the average of the per-step scores.

Assess the following per step:

- Does the agent correctly use relevant parts of the input?
- Is the input used appropriate for the tool selected and the task being attempted?
- Does the reasoning show effective and meaningful use of the input?

Scoring Criteria:

- 1 – The agent uses the input fully and appropriately.
- 0.5 – The agent uses some relevant input but misses important details or misuses others.
- 0 – The agent ignores or misinterprets the input entirely.

Output format:
Score: <0, 0.5, or 1>
Justification: <1-2 sentence explanation for the score>"

Figure 29: Prompt for evaluating the contextual grounding of each reasoning step.

---

**Factual Accuracy**

"You are an evaluation assistant measuring the factual accuracy of the agent's reasoning steps.

You are given a Ground Truth (GT) block containing the original query, a sequence of reasoning steps, and the final answer along with its justification. Each reasoning step includes the task being attempted, the tool used (with its input and output), and the agent's thought process.

You are also given the full reasoning trace produced by the agent, including each step's task, selected tool, tool input/output, and thought.

Your task is to assess whether the agent introduces any hallucinated, fabricated, or factually incorrect information in its reasoning when compared to the GT.

Evaluation Guidelines:

- Compare the model's reasoning steps to the GT to identify factual hallucinations or incorrect claims.
- Focus on whether the output or thought includes details not supported by the GT.
- Do not penalize for minor omissions unless they lead to a factual error.

Scoring Criteria:

- 1 – No factual errors or hallucinations compared to the GT.
- 0.5 – Minor inaccuracies or vague mismatches.
- 0 – Major factual errors or hallucinated content.

Output format:
Score:  <0, 0.5, or 1>
Justification:  <1-2 sentence explanation for the score>"

---

Figure 30: Prompt for evaluating factual correctness of the agent's reasoning steps.

---

**Semantic Accuracy**

"You are an evaluation assistant measuring the Semantic Accuracy of an agent's reasoning process.

You are given a Ground Truth (GT) block containing the original query, a sequence of reasoning steps, and the final answer along with its justification. Each reasoning step includes the task being attempted, the tool used (with its input and output), and the agent's thought process.

You are also given the full reasoning trace produced by the agent, including each step's task, selected tool and its output, and thought, as well as the agent's final answer and justification.

Your task is to assess whether the agent's reasoning and final output semantically align with the Ground Truth, i.e., whether the agent has covered all the essential parts of the query as demonstrated in the GT.

Evaluation Guidelines:

- Compare the agent's reasoning trace and final answer with the GT to check whether all key components of the query are addressed.
- Credit should be given for meaningful semantic coverage, not superficial similarity.
- If the model ignores or misunderstands core parts of the GT reasoning or final answer, penalize accordingly.

Scoring Criteria:

- 1 – The agent addresses all key components of the query, matching the GT's semantic scope.
- 0.5 – Some parts are covered, but the response misses or weakly handles other essential elements.
- 0 – The agent's reasoning or answer omits or misrepresents major parts of the query.

Output format:
Score:   <0, 0.5, or 1>
Justification:   <1-2 sentence explanation for the score>"

---

Figure 31: Prompt for evaluating semantic consistency of the agent's reasoning and answer.

---

**Goal Accuracy**

"You are an evaluation assistant measuring the Goal Accuracy of a
vision-language agent's final answer.

You are given the original query, the agent's final output, the ground
truth (GT) final answer, and the type of query - either "objective" or
"subjective".

Your task is to assess how well the agent's final output matches the ground
truth answer, based on the nature of the query.

Evaluation Guidelines:

- The Goal Accuracy score is computed once per query (not per step).
- Use exact match evaluation for objective queries.
- Use semantic similarity evaluation for subjective queries.

Scoring Criteria:

- **For objective queries:**
    - 1 - The final output matches the GT exactly or is clearly
      equivalent.
    - 0 - The output is incorrect, incomplete, or unrelated.
- **For subjective queries:**
    - Score = Cosine similarity between the agent's answer and the GT
      answer (range: 0 to 1)

Output format:
If objective:
Score: <0 or 1>
Justification: <Optional 1-2 sentence explanation>

If subjective:
Score: <cosine similarity>
Justification: <Optional 1-2 sentence explanation>"

---

Figure 32: Prompt for evaluating alignment of final answer with the ground truth.

---

**Toolset Accuracy**

"You are an evaluation assistant measuring the toolset accuracy of an agent's reasoning process.

You are given a Ground Truth (GT) block containing the original query, a sequence of reasoning steps, and the final answer along with its justification. Each reasoning step includes the task being attempted, the tool used (with its input and output), and the agent's thought process.

You are also given the full reasoning trace produced by the agent, including each step's task, selected tool and its output, and thought.

Your task is to evaluate whether the agent used the correct tools overall by comparing the set of tools it used to the set used in the GT.

Evaluation Guidelines:

- If the agent uses tools that are not present in the GT or misses tools that are, it should be penalized.

- The score reflects how well the agent's toolset aligns with the GT toolset across the full reasoning trace.

Output format:
Score:  <F1 score rounded to 2 decimal places>
Justification:  <1-2 sentence explanation for the score>"

---

Figure 33: Prompt for evaluating alignment of the agent's toolset with the ground truth.

## L    Benchmark Curation Examples

The raw LMM-generated queries were often suboptimal: they merged unrelated tasks (e.g., "identify objects, find government symbols, count frames, *and* explain political significance"), contained ill-posed demands such as "circle the damaged part" without an annotation interface, or asked for information invisible in the image (e.g., a jet's full landing-gear specification from a single profile shot). As illustrated in Fig. 34, therefore, every prompt underwent a mandatory human-in-the-loop rewrite. Curators distilled each draft into a single, objectively answerable question that aligns with the available tools, for example, "How many people are carrying patterned umbrellas, and what pattern is on the umbrella closest to the camera?" This step removes speculation, narrows the scope to verifiable visual evidence, and converts noisy LLM proposals into reliable, evaluation-grade tasks.

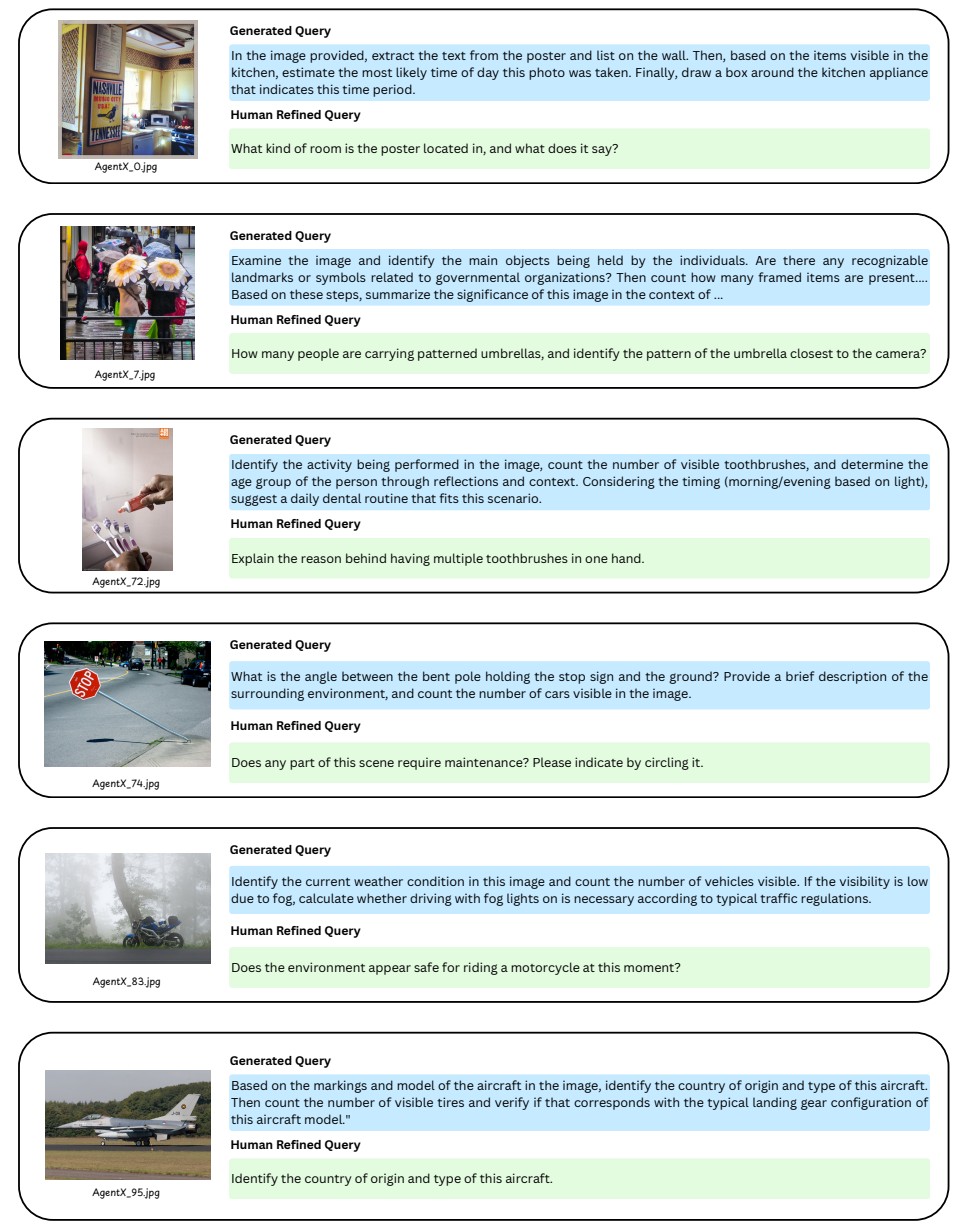

Figure 34: Illustrative corrections: original LMM queries versus the refined, task-focused prompts used in our benchmark.

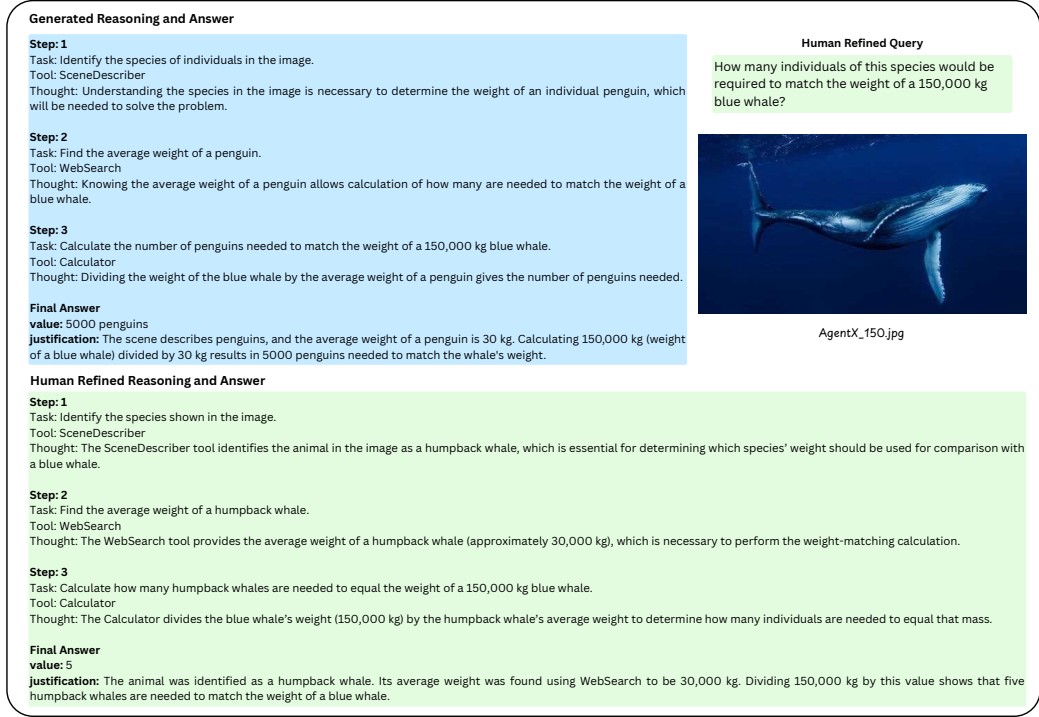

Figure 35: **Before–after refinement of an automatically generated reasoning trace.** *Left pane:* the raw LMM output mis-identifies the species (penguin → humpback whale), fabricates an `ObjectCounter` call, and propagates the error to an absurd "5000 penguins" answer. *Right pane:* after human verification, each tool is appropriate to its sub-task, the correct constant answer is retrieved, and the calculator produces the logically consistent reasoning. The revision phase thus excises hallucinated steps, fixes factual look-ups, and restores end-to-end coherence.

## L.1 Reasoning Trace Generation

After a query was finalised, we asked an `LMM` to *auto-expand* it into a complete reasoning trace: thoughts, tool calls, and a numerical answer. These raw traces were often self-contradictory: a model might identify *penguins* in Step 1, retrieve the average weight of a *humpback whale* in Step 2, hallucinate an `ObjectCounter` call, and then feed the spurious value into the calculator. Accordingly, every trace underwent a mandatory **Phase-2 human pass** in which annotators:

1. verified that each tool matched the declared sub-task,
2. replaced incorrect factual look-ups, and
3. recomputed downstream results whenever earlier constants changed.

In the whale–weight vignette, the editor corrected the species mismatch, supplied a defensible 30,000 kg 30,000 kg estimate for a humpback whale, and updated the calculator output from "5000 penguins" to the logically sound "5 humpback whales". This clean-up ensures that every released trajectory is *logically coherent, numerically sound, and fully executable*. Figure 35 juxtaposes the `LMM`-generated trace (left), replete with species confusion and cascading arithmetic errors: with the human-refined version (right), where each tool invocation, intermediate value, and the final answer are perfectly aligned. The comparison underlines the necessity of Phase-2 curation for converting noisy, auto-generated chains of thought into dependable, evaluation-grade ground truth.

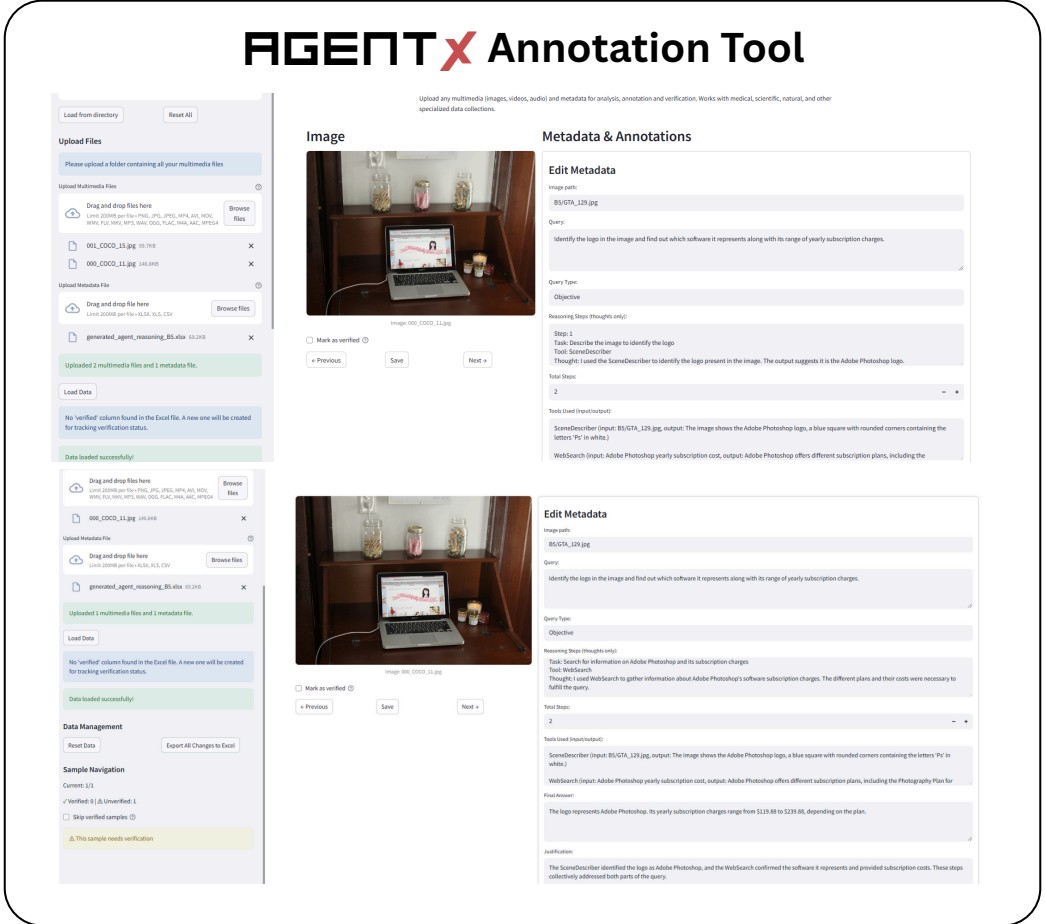

Figure 36: Agent-X Annotation Tool Example

## L.2 Human Verification Tool

To ensure every sample in Agent-X meets our quality bar, we developed a small web-based "Agent-X Reasoning Annotation Tool." When an annotator drags a folder onto the page the interface instantly loads the raw media on the centre canvas, the agent's step-by-step rationale in a panel to the right, and a navigation list of all remaining tasks on the left. Each reasoning field, including the task description, chosen tool, tool I/O, and final answer, can be edited inline; a single Mark as verified tick confirms the trace is now correct. The Save button writes changes back to the original Excel file, and a global Export action adds a "verified" column plus colour-codes every modified cell so downstream scripts can compute inter-annotator agreement. In practice reviewers process roughly fifty samples per hour, and Figure 36 illustrates two typical sessions: identifying an Adobe Photoshop logo. The tool's tight coupling of evidence, metadata, and editing shortcuts turns what would be a cumbersome spreadsheet task into a fluid, human-in-the-loop verification workflow.

