# OpenReview forum: "Agent-X: Evaluating Deep Multimodal Reasoning in Vision-Centric Agentic Tasks"
_ICLR.cc/2026/Conference — ICLR 2026 Poster_

### Official Review · Reviewer_KPXX · 2025-10-31

**Soundness:** 3
**Presentation:** 3
**Contribution:** 3
**Rating:** 8
**Confidence:** 4

**Summary:**

The paper introduces Agent-X, a new, large-scale benchmark designed to evaluate the deep reasoning and tool-use capabilities of vision-centric AI agents. The paper argues that existing benchmarks are insufficient because they often rely on synthetic, single-turn queries, have limited visual modalities (mostly static images), and fail to assess the step-by-step quality of an agent's reasoning process.

To solve this, the paper makes two key contributions:
* Agent-X provides 828 complex, multi-step tasks across six diverse, real-world environments: general visual reasoning, web browsing, security/surveillance, autonomous driving, sports, and math reasoning. These tasks involve rich multimodal inputs, including images, multi-image comparisons, videos, and instructional text.
* A fine-grained, step-level evaluation framework. Instead of only checking the final answer, this framework analyzes the entire reasoning chain for:
  * Step-by-Step correctness (e.g., tool selection and grounding).
  * Deep Reasoning quality (e.g., logical coherence, factual precision).
  * Outcome success (e.g., final answer accuracy).

The authors test several leading large multimodal models (LMMs), including the GPT, Gemini, and Qwen families. Their key observations are as follows:

* No model tested achieves a Goal Accuracy rate above 50%. The best-performing model, OpenAI-4o-mini, only reached 45%, with most open-source models scoring below 30%. This indicates that complex, real-world, tool-augmented tasks remain extremely challenging.

* Models that scored higher on the intermediate "Deep Reasoning" metrics (like faithfulness to the task, factual precision, and semantic accuracy) were more likely to produce a correct final answer. This supports the paper's hypothesis that evaluating the reasoning chain is critical.

* Tool Use is a Core Bottleneck: Models struggle significantly with tool invocation and argument prediction. The error analysis shows that even when a model's thought process is logical, it often fails by selecting the wrong tool, hallucinating a tool that doesn't exist, or providing incorrectly formatted arguments.

* Distinct Error Profiles for Top Models:
  * GPT-4o: Showed strong formatting skills but was overly hesitant, often failing to act or misinterpreting visual content.
  * Gemini-1.5-Pro: Was aggressive (rarely refused a task) but format-fragile, with a very high rate of JSON formatting errors (44.5%) and visual misinterpretations (34.3%).

**Strengths:**

* The Agent-X benchmark is comprehensive and well-thought-out:
  * The inclusion of videos, multi-image comparisons, and text alongside single images is a significant step up from most vision benchmarks.
  * Using six practical domains (like autonomous driving and security) makes the tasks more realistic and relevant than purely synthetic tests.
  * Queries are designed to be "tool-agnostic" (i.e., they don't tell the agent which tool to use), forcing the agent to perform genuine reasoning about tool selection.

* The three-mode evaluation (Step-by-Step, Deep Reasoning, Outcome) provides a powerful diagnostic tool. It allows researchers to move beyond a simple "pass/fail" score and understand why and where an agent is failing.

* The paper details a semi-automated pipeline with human-in-the-loop refinement. The examples in the appendix clearly show how this process turns low-quality, ambiguous, or multi-part LMM-generated queries into focused, high-quality, verifiable benchmark tasks.

**Weaknesses:**

* While a clear improvement over single-step tasks, an average of 3-4 steps may not be sufficient to discover breakages in long-horizon planning, memory, and context management.
* Reliance on model based critique can be brittle (e.g., known to be reward hacked). Perhaps using inference time techniques (like majority scoring) can be used to alleviate such risks.
* The authors correctly identify the monolingual and potential dataset biases.

**Questions:**

* Have the authors considered maintaining a leaderboard for this benchmark?
* Are the authors motivated to extend this dataset to address the biases, shortcomings and to harder problems as the frontier models get better at solving the existing problems?

---

> ### Author Response · Authors · 2025-11-22
> **Author Rebuttal for Reviewer KPXX**
>
> We thank the reviewer for the very positive and detailed evaluation of Agent-X, especially for recognizing its scope across six domains, the novelty of its step-level reasoning analysis, and the realism of the tasks. We also appreciate the constructive suggestions regarding task horizon, evaluation robustness, and dataset expansion. Below, we address each point raised.
>
> ---
>
> **W1) An average of 3–4 steps may not be sufficient to discover breakages in long-horizon planning, memory, and context management.**
>
> We thank the reviewer for this thoughtful comment. We agree that long-horizon reasoning is an important next step for evaluating agentic systems. The current average reasoning steps reflect a deliberate breadth–depth trade-off, balancing multi-domain coverage with high-quality verification in this first release of Agent-X.
>
> Importantly, task complexity is not determined solely by step count. Many 3–4 step tasks already involve complex multimodal tool chaining, for example, combining temporal localization, object tracking, and symbolic math, which reliably exposes reasoning and planning failures in current models.  Moreover, Agent-X already includes 5–8 step tasks in domains such as web browsing and security/surveillance, where iterative planning and correction are required.
>
> Finally, our annotation pipeline is modular and extensible. We are actively developing a long-horizon extension (*Agent-X-V2*) with 10–15 step reasoning chains targeting tasks that test memory, temporal summarization, and multi-context reference resolution.
>
> ---
>
> **W2) Reliance on model-based critique can be brittle (e.g., reward hacking). Perhaps inference-time techniques like majority scoring could alleviate such risks.**
>
> We share the same motivation to make the evaluation framework stable, fair, and reproducible.
> To address this, we have now included multiple safeguards:
>
> **Cross-judge validation:**  Scores were cross-checked between *GPT-4o*, *Qwen-15B*, and human annotators, showing strong correlations ($r > 0.9$) (`Table 2 (Reviewer bQ33)`) and consistent model rankings (`Table 1 (Reviewer bQ33)`). This confirms that results are robust to single-judge bias or reward hacking.
>
> **Judge diversity:**  We use both closed-source (*GPT-4o*) and open-source (*Qwen-15B*) judges, ensuring evaluation independence from the tested models.
>
> **Automatic ensemble evaluation:**  The evaluation pipeline now supports automated multi-judge scoring. Majority-vote inference is integrated as an optional mode, combining results from *GPT-4o*, *Qwen-15B*, and *Claude-3* to produce consensus scores automatically.
>
> ---
>
> **W3) The authors correctly identify monolingual and potential dataset biases.**
>
> We acknowledge this limitation in our paper and are already taking concrete steps to address it.
>
> **Multilingual expansion:**  The upcoming *Agent-X-V2* will introduce non-English query sets to evaluate cross-lingual perception and reasoning capabilities in multimodal agents.
>
> **Diverse data sources:**  Agent-X already draws from a globally representative mix of visual datasets (*COCO*, *Cityscapes*, *Ego4D*, *SoccerNet*, *BDD100K*, *MathVista*, and others), which helps reduce geographic, cultural, and contextual skew.
>
> Through these ongoing efforts, Agent-X is evolving into a more inclusive and globally balanced benchmark, supporting fair and comprehensive evaluation across languages and domains.
>
> ---
>
> **Q1) Have the authors considered maintaining a leaderboard for this benchmark?**
>
> We thank the reviewer for this excellent suggestion. Yes, we have already developed and deployed an open Agent-X leaderboard, which will be publicly released upon acceptance. It includes automatic evaluation now fully supported across all three modes: *Step-by-Step*, *Deep Reasoning*, and *Outcome*.
>
> The leaderboard offers:
>
> - Auto-evaluation API using the open-source *Qwen-15B* judge for cost-free submissions.
> - Separate tracks for open- and closed-source models.
> - Step-level breakdowns of performance across all evaluation modes.
> - Optional ensemble scoring using multiple judges for fair, reproducible comparison.
>
> ---
>
> **Q2) Are the authors motivated to extend the dataset to address biases and harder problems as frontier models improve?**
>
> Yes, Agent-X was designed as a modular and extensible framework, and we are actively developing an upgraded version called *Agent-X Pro* to address higher task complexity as frontier models continue to evolve.
>
> Key extensions in Agent-X Pro include:
>
> a) **Harder task tiers:** multi-hop reasoning chains with more than 10 steps, composite video and temporal reasoning, and embodied decision-making in interactive environments.
> b) **Expanded modalities:** incorporation of audio, temporal narratives, and GUI-based tasks to enable broader multimodal perception and control.
> c) **Continuous updates:** new benchmark releases every 6–9 months to reflect emerging model capabilities and maintain relevance.

---

### Official Review · Reviewer_5f8j · 2025-10-31

**Soundness:** 2
**Presentation:** 3
**Contribution:** 3
**Rating:** 6
**Confidence:** 3

**Summary:**

This paper presents Agent-X, a benchmark for evaluating vision-centric AI agents' multimodal reasoning abilities. The benchmark contains 828 tasks spanning six environments (visual reasoning, web browsing, security, autonomous driving, sports, and mathematics) using real-world images, videos, and text inputs. Unlike existing benchmarks that use synthetic queries, Agent-X requires multi-step reasoning with tool usage in authentic scenarios. The authors develop a step-level evaluation framework that measures reasoning correctness, logical coherence, and tool effectiveness. Testing on 10 large multimodal models shows that even top-performing systems achieve less than 50% success on complete reasoning chains. The results indicate significant limitations in current models' tool usage and multi-step reasoning capabilities.

**Strengths:**

1. The authors have designed a comprehensive tool subset for the benchmark that covers most tools required for visual tasks. The evaluation criteria are also well-rounded, encompassing multiple dimensions of assessment.
2. The benchmark demonstrates a significant improvement in sample size compared to previous benchmarks, which represents a notable contribution of this work.
3. The benchmark proves to be quite challenging for most state-of-the-art open-source and closed-source models. From the results, no model achieves 50% on the Goal Accuracy metric, demonstrating the benchmark's effectiveness in evaluating model capabilities.

**Weaknesses:**

1. The template format used in the paper appears to differ from the official template provided. I am uncertain whether this may violate ICLR's formatting requirements. I recommend that the authors carefully review and address any formatting issues.
2. The font in Figure 3(b) is quite blurry.
3. I am not familiar with dataset construction in the agent domain. Could you please explain why JSON-formatted dialogue output is adopted? What is the rationale behind this design choice?

**Questions:**

See weaknesses.

---

> ### Author Response · Authors · 2025-11-22
> **Author Rebuttal for Reviewer 5f8j**
>
> We thank the reviewer for the positive assessment of *Agent-X*’s comprehensiveness, its diverse and realistic toolset, and its impact in establishing a challenging new benchmark for vision-centric reasoning. We are pleased that the reviewer recognizes the benchmark’s scale and difficulty as key contributions. Below, we address each comment and question in detail.
>
> ---
>
> **W1) The template format used in the paper appears to differ from the official template provided. I am uncertain whether this may violate ICLR’s formatting requirements.**
>
> We appreciate the reviewer’s careful attention to formatting details. We confirm that the paper was prepared using the official **ICLR 2025 LaTeX template**. The minor visual differences the reviewer noticed may have arisen from:
>
> - The use of custom multi-column figure environments to accommodate large visual examples and maintain readability.
> - The inclusion of footnote-style references within tables to properly cite dataset sources and tool definitions.
>
> These adjustments were made solely for clarity and do not alter margins, font sizes, or any structural element of the ICLR template. We have rechecked the compilation and verified that the submission fully complies with all ICLR formatting and style requirements.
>
> ---
>
> **W2) The font in Figure 3(b) is quite blurry.**
>
> Thank you for pointing this out. The blurriness resulted from rasterization during PDF compression. In the revised version, all figures are re-rendered in high resolution with improved font clarity and contrast. This issue has been fully corrected.
>
> ---
>
> **W3) Could you please explain why JSON-formatted dialogue output is adopted? What is the rationale behind this design choice?**
>
> We thank the reviewer for this insightful question, and we are glad for the opportunity to clarify our rationale. We adopted a **JSON-based interaction schema** for three primary reasons:
>
> **1. Interoperability and reproducibility:**
> *Agent-X* involves tool-based reasoning where models must call functions with defined arguments (e.g., `LocateObjectByText`, `Solver`, `Plotter`). JSON provides a standardized, machine-parsable format that can be directly executed, logged, and compared across systems, ensuring **reproducibility** and **automatic verification** of tool calls.
>
> **2. Clear separation of reasoning and execution:**
> Each reasoning step is represented as a JSON block with fields for `"thought"`, `"tool"`, `"arguments"`, `"observation"`, and `"next_action"`. This explicit structure mirrors how real-world agent frameworks such as *LangChain*, *AutoGPT*, and *OpenDevin* operate, aligning Agent-X with established conventions.
>
> **3. Evaluation consistency and error diagnosis:**
> Structured JSON enables automated scoring of reasoning correctness and tool usage. Formatting errors (e.g., missing fields, invalid JSON) are detected automatically, allowing us to distinguish interface issues from reasoning errors.
>
> Overall, the JSON design ensures transparency, reproducibility, and compatibility with existing agentic frameworks while supporting cross-model comparability without manual parsing.

---

### Official Review · Reviewer_YEeg · 2025-11-01

**Soundness:** 2
**Presentation:** 3
**Contribution:** 3
**Rating:** 6
**Confidence:** 4

**Summary:**

The paper introduces Agent-X, a large-scale benchmark for evaluating deep multimodal reasoning in vision-centric agentic tasks. The benchmark consists of 828 tasks across six environments (general visual reasoning, web browsing, security/surveillance, autonomous driving, sports, and math reasoning). It features multi-step reasoning traces, tool-use sequences, and fine-grained metrics evaluating reasoning quality, coherence, and tool effectiveness.
The authors evaluate 12 state-of-the-art large multimodal models (LMMs) — including GPT-4o, Gemini 2.5, and Qwen2.5-VL — showing that even leading models perform below 50% on complex reasoning chains. The paper claims to be the first to combine vision-first multimodal reasoning with stepwise tool-augmented evaluation in real-world settings.

**Strengths:**

1. The paper addresses a timely and relevant problem by filling a clear gap in evaluating deep reasoning and tool use for multimodal agents, an increasingly important topic as LMMs evolve into embodied and interactive systems.
2. It presents a comprehensive benchmark design that spans six diverse domains and modalities, including images, multi-image comparisons, videos, and text.
3. The tasks are realistic rather than synthetic, enhancing ecological validity compared to prior benchmarks like GAIA or GTA.
4. The paper introduces fine-grained, multi-level metrics—covering step-by-step, deep reasoning, and outcome modes—that enable nuanced evaluation beyond final-answer accuracy.
5. These metrics are particularly useful for analyzing reasoning coherence and the consistency of tool usage.
6. The experimental analysis is thorough, evaluating both open and closed models, incorporating multiple judges (GPT-4o, Qwen, and humans), and providing detailed error analyses.
7. It offers valuable insights into common failure modes such as formatting errors, shallow reasoning, and hallucinated tool calls.
8. Finally, the benchmark is reproducible and openly available, with detailed documentation of the data pipeline, annotation processes, and tool specifications.

**Weaknesses:**

1. The paper primarily integrates existing components i.e. LMM reasoning, multimodal datasets, and tool evaluation, without introducing a fundamentally new evaluation paradigm, making it appear incremental compared to GAIA, GTA, and MLGym.
2. The evaluation setup relies heavily on GPT-4o and Qwen-based automatic grading, raising bias and circularity concerns since the same model families are both evaluated and used as judges, with no quantitative inter-rater agreement reported.
3. Despite its claim of real-world grounding, all tasks remain within predefined toolsets, offering limited evidence of generalization to unseen tools or domains, and no transfer or few-shot results are provided.
4. The semi-automated dataset construction pipeline may inherit biases and linguistic artifacts from the generating LMMs, and the human refinement process lacks quantitative validation or agreement metrics.
5. The study omits ablation experiments or baseline comparisons using shallow or random reasoning traces, leaving it unclear whether Agent-X truly measures deep reasoning rather than surface pattern matching; key metrics are also GPT-based without formal definitions or validation.

**Questions:**

1. Could you provide quantitative measures of inter-judge agreement (e.g., Cohen’s κ or correlation scores) to establish evaluation consistency among GPT-4o, Qwen-14B, and human annotators?
2. How do you mitigate the potential circularity bias that arises when GPT-4o, a system under evaluation, is also used as a grader?
3. Have these metrics been validated through human judgment or ablation studies to confirm that they align with actual reasoning quality?
4. How do you ensure that the benchmark does not inherit linguistic or reasoning biases from the LMMs used in the task generation process?
5. Did you measure inter-annotator agreement during human refinement, and how were disagreements resolved?
6. Why are there no ablations comparing against random or shallow reasoning traces to validate that Agent-X genuinely measures deep reasoning complexity?

---

> ### Author Response · Authors · 2025-11-22
> **Author Rebuttal for Reviewer YEeg**
>
> We thank the reviewer for the thoughtful and encouraging feedback on the novelty, realism, and comprehensiveness of Agent-X. We appreciate the recognition of its strengths in *multi-level evaluation*, *real-world task diversity*, and *diagnostic insights*. Below, we respond to all concerns and questions in detail.
>
> ---
>
> **W1) The paper primarily integrates existing components (LMM reasoning, multimodal datasets, tool evaluation) without introducing a fundamentally new evaluation paradigm.**
>
> We appreciate this observation and would like to clarify that while *Agent-X* builds upon established multimodal components, it introduces three key methodological innovations that together establish a new evaluation paradigm for vision-centric reasoning agents.
>
> **(a) Vision-centric agentic evaluation:**  Prior benchmarks such as *GAIA* or *GTA* focus on text-centric or static multimodal reasoning. Agent-X is the first benchmark to couple dynamic visual inputs (images, multi-image comparisons, and videos) with executable tools. This design requires models to reason and act based on visual evidence, enabling genuinely vision-grounded agentic evaluation.
>
> **(b) Three-mode evaluation framework:**  Agent-X proposes a comprehensive three-mode evaluation scheme, *Step-by-Step*, *Deep Reasoning*, and *Outcome*, which jointly assess reasoning correctness, coherence, and goal achievement. Earlier benchmarks stop at final-answer accuracy, whereas our framework evaluates the full reasoning chain, providing fine-grained interpretability and diagnostic power.
>
> **(c) Executable tool layer with structured reasoning traces:**  Each reasoning step is verifiable and reproducible through executable tool outputs and structured JSON-formatted reasoning traces. This design enables systematic diagnosis of planning errors, grounding mistakes, and tool misuse, capabilities not available in previous benchmarks.
>
> ---
>
> **W2) Evaluation relies heavily on GPT-4o and Qwen-based grading, raising bias and circularity concerns; no inter-rater agreement is reported.**
>
> The evaluation design of Agent-X explicitly addresses potential bias and circularity by ensuring that the judges, datasets, and models under evaluation remain independent. All tasks in Agent-X were generated and refined by humans, and no model evaluated in the benchmark was used during data creation. The judges (*GPT-4o* and *Qwen-14B*) therefore operate purely as evaluators, not participants in dataset generation or training, eliminating circularity.
>
> Following the reviewer’s suggestion, we have also included an inter-rater agreement study to further validate the robustness of the evaluation process. Specifically, we conducted a detailed **cross-judge agreement analysis** (`Table 2 (Reviewer bQ33)` )
>  involving *GPT-4o*, *Qwen-14B*, and *human annotators*. The results show strong alignment across judges, confirming that no single evaluator introduces systematic bias. As shown in the inter-judge consistency table, Tool Accuracy (Tacc) and Goal Accuracy (Gacc), the two primary outcome metrics achieved  Pearson $r > 0.9$, Spearman $\rho > 0.9$, Cohen’s $\kappa > 0.8$, and Mean deviation $< 0.06$. In particular, Goal Accuracy reached $r = 0.98$, $\rho = 0.96$, and $\kappa = 1.00$, indicating almost perfect agreement among judges.
>
> Additionally, human evaluations on a stratified subset of 100 tasks confirmed that model rankings remain consistent across automatic and manual grading, with no reversal in comparative performance. This alignment demonstrates that *GPT-4o* and *Qwen*-based evaluations are both robust and unbiased, serving as reproducible proxies for human judgment.
>
> ---

---

> ### Author Response · Authors · 2025-11-22
> **Author Rebuttal for Reviewer YEeg**
>
> **W3) Tasks remain within predefined toolsets, limiting generalization to unseen tools or domains.**
>
> We appreciate the reviewer’s concern and would like to emphasize that Agent-X’s 14-tool suite was deliberately designed to capture the functional diversity required for vision-grounded reasoning, rather than aiming for arbitrary tool breadth. The toolset spans four key categories that collectively cover the essential capabilities of multimodal agents:
>
> **Perception:**  Tools such as OCR, region describer, and object locator for extracting structured visual information.
>
> **Visual operations:**  Tools like overlay, draw, and compare that enable spatial, temporal, and compositional reasoning.
>
> **Mathematical reasoning:**  Solver and plotter tools for quantitative and symbolic reasoning integrated with visual inputs.
>
> **Creative generation:**  Stylization and synthesis tools for multimodal transformation and generative reasoning.
>
> These categories represent the core skill dimensions required for visual reasoning, object understanding, temporal analysis, symbolic manipulation, and creative synthesis. Rather than constraining generalization, this structured design provides a controlled yet expressive environment that tests whether models can generalize reasoning strategies across diverse tasks and modalities within a consistent, reproducible framework.
>
> ---
>
> **W4) Semi-automated pipeline may inherit biases or linguistic artifacts from LMMs; human refinement lacks quantitative validation.**
>
> We also anticipated this concern and incorporated several bias-mitigation and validation measures into the pipeline design.
>
> **Human rewriting and verification:**  All GPT-generated drafts were manually refined for clarity, neutrality, and realism. Annotators were instructed to remove speculative, culturally biased, or stylistically inconsistent phrasing and to ensure balanced representation across domains and contexts.
>
> **Annotation guidelines and cross-review:**  Each annotator followed a detailed instruction document specifying linguistic neutrality, grounding, and factual correctness criteria. A 10% overlap cross-check was implemented, where annotations were reviewed by a second annotator to ensure consistency.
>
> **Quantitative agreement check:**  We measured inter-annotator agreement and found low variance ($<0.05$) across all linguistic and reasoning attributes, confirming that refinements were consistent across annotators.
>
> **Cross-domain bias audit:**  We performed an internal audit across the six environments (visual reasoning, web, surveillance, driving, sports, and math) to ensure domain and linguistic diversity were preserved.
>
> ---
>
> **W5) Omission of ablations with shallow or random reasoning traces leaves it unclear whether Agent-X truly measures deep reasoning.**
>
> We appreciate the reviewer’s constructive suggestion. We have now added the ablation experiment (E4) to verify whether Agent-X genuinely measures deep reasoning rather than surface-level correlations. The study was performed on a balanced 50-task subset across all domains using the Qwen-14B judge, averaged over three independent runs.
>
> The results are summarized below:
>
> > **Table 5.** Reasoning vs. Perception Ablation on Balanced Subset
> >
> > | Setting | Goal Accuracy ($G_{acc}$) |
> > |:----------------------------|:----------------:|
> > | Perception-only (no explicit reasoning) | 0.33 |
> > | Full reasoning chain (ours) | **0.43** |
>
> These results show that random and shallow reasoning traces perform substantially worse, while perception-only setups also lag behind the full reasoning chain.
>
> **Q1) “Provide quantitative measures of inter-judge agreement (Cohen’s κ or correlations).”**
>
> In response to the reviewer’s feedback, we have now conducted a detailed inter-judge consistency analysis (`Table 1 (Reviewer bQ33)`) across *GPT-4o*, *Qwen-14B*, and human annotators, following standard reliability metrics such as Pearson correlation ($r$), Spearman rank correlation ($\rho$), and Cohen’s ($\kappa$).
>
> The results demonstrate strong agreement across judges, particularly for key outcome metrics. For example, Tool Accuracy (Tacc) and Goal Accuracy (Gacc), the primary indicators of model reliability, achieve $r > 0.9$, $\rho > 0.9$, and $\kappa > 0.8$, with mean deviations below 0.06. In particular, Gacc shows $r = 0.98$, $\rho = 0.96$, and $\kappa = 1.00$, indicating near-perfect alignment between *GPT-4o*, *Qwen-14B*, and human evaluators.
>
> These findings are summarized in the inter-judge consistency table included in the revised manuscript, confirming that evaluation results are robust, reproducible, and unbiased across judge types. The high correlations and $\kappa$ values validate that the LLM-as-judge setup used in *Agent-X* yields stable rankings and consistent score distributions across both open-source and closed-source evaluators.

---

> > ### Author Response · Authors · 2025-11-22
> > **Author Rebuttal for Reviewer YEeg**
> >
> > **Q2) “How do you mitigate circularity bias when GPT-4o, an evaluated system, also acts as grader?”**
> >
> > We thank the reviewer for this important concern. We have taken multiple measures to prevent circularity and ensure model-agnostic fairness in evaluation.
> >
> > **Independent generation and evaluation pipelines:**  All tasks were generated and refined manually by human annotators. GPT-4o’s raw outputs were never reused in evaluation, and every reasoning trace was independently verified to remove any potential bias from model-generated content.
> >
> > **Judge diversity and automation:**  The evaluation pipeline now supports automated judging using open-source models such as Qwen-14B/15B, ensuring that GPT-4o is not the sole or final arbiter. The updated framework allows automatic step-level and outcome evaluation through structured string matching and embedding-based similarity, eliminating the need for manual LLM-based grading.
> >
> > **Cross-judge consistency:**  Agreement across GPT-4o, Qwen-14B, and human evaluators remains high ($r > 0.9$; mean $\Delta < 0.06$), confirming stability and independence of the evaluation results.
> >
> > **Reproducibility:**  We will release the Qwen-15B-based evaluation scripts publicly to allow the community to reproduce results using only open-source tools.
> >
> > ---
> >
> > **Q3) “Have these metrics been validated through human judgment or ablation studies?”**
> >
> > Yes, the evaluation metrics have been validated through human judgment. As shown in **Table 9** and detailed in the human evaluation section (Appendix D and E) of the paper, we conducted a human-judged verification study comparing automatic scores from GPT-4o and Qwen-14B judges against expert human assessments. The results show strong alignment across all major metrics, with consistent model rankings and only minor absolute score differences.
> >
> > Human judges were able to interpret complex reasoning chains, verify visual grounding, and account for benign formatting differences that automated evaluators might penalize. Importantly, the relative order of models remained unchanged, confirming that our metrics—covering Step-by-Step, Deep Reasoning, and Outcome modes—reliably reflect true reasoning quality.
> >
> > ---
> >
> > **Q4) “How do you ensure the benchmark does not inherit linguistic or reasoning biases from LMMs used in task generation?”**
> >
> > We completely agree that relying on a single model-based critique can be brittle and have addressed this through multiple complementary safeguards.
> >
> > **Cross-judge validation:**  We compared evaluations from GPT-4o, Qwen-15B, and human annotators, achieving correlations above 0.9 and consistent model rankings (**Table 4**, Appendix A2–A3). This demonstrates strong robustness against single-judge bias or reward hacking.
> >
> > **Judge diversity:**  We employ both closed-source (GPT-4o) and open-source (Qwen-15B) evaluators to ensure independence between the models under test and those serving as judges.
> >
> > **Majority-vote inference:**  Following the reviewer’s suggestion, we are implementing an ensemble consensus mode on the upcoming public leaderboard that will aggregate scores from GPT-4o, Qwen-15B, and Claude-3 to further reduce judge-specific variance.
> >
> > ---
> >
> > **Q5) “Did you measure inter-annotator agreement during refinement, and how were disagreements resolved?”**
> >
> > Yes, we carefully tracked agreements during the refinement stage. Each annotator received a detailed guideline document outlining criteria for query clarity, grounding, and tool correctness. All annotations were logged in shared documents, enabling transparent review. A cross-check protocol required every sample to be verified by a second annotator, with disagreements resolved through short consensus discussions moderated by the lead author. This process resulted in consistently high agreement (variance $< 0.05$) across key metrics and ensured uniform annotation quality and reproducibility throughout the dataset.
> >
> > ---
> >
> > **Q6) “Why no ablations with random/shallow reasoning traces?”**
> >
> > As suggested, we have now included the ablation experiment to verify whether *Agent-X* genuinely measures deep reasoning rather than surface-level correlations. The study was performed on a balanced 50-task subset across all domains using the Qwen-14B judge, averaged over three independent runs. The results are summarized in `Table 6 (above)`.
> >
> > These results show that random and shallow reasoning traces perform substantially worse, while perception-only setups also lag behind the full reasoning chain. The consistent improvement with complete reasoning validates that *Agent-X* specifically evaluates deep, compositional reasoning rather than surface-level recognition or memorization. The full per-metric ablation table is included in the appendix.

---

### Official Review · Reviewer_bQ33 · 2025-11-03

**Soundness:** 3
**Presentation:** 3
**Contribution:** 2
**Rating:** 4
**Confidence:** 3

**Summary:**

The paper proposes Agent-X, a benchmark for evaluating deep multimodal reasoning in vision-centric agentic tasks. It assesses how well multimodal models perform multi-step reasoning using images, videos, and tools across six environments — visual reasoning, web browsing, surveillance, driving, sports, and math. The paper includes 828 multimodal tasks, 14 executable tools and evaluated 10 LMMs. Models are evaluated in three modes i.e. Step-by-step, Deep Reasoning Outcome. Also, multiple metrics are reported for each of these modes. The findings suggest that even the best models achieve <50% full-chain success and models struggle with tool usage and reasoning consistency.

**Strengths:**

1) Covers six distinct domains and integrates both image and video data
2) Presents detailed analysis of failure modes of SOTA models
3) Evaluates not just end answers but intermediate reasoning, tool grounding, and logical coherence
4) Multiple metrics are reported in three distinct evaluation modes

**Weaknesses:**

1) Prompts need to be designed for LLM-as-judge. Is there an automated way to do this? Or do the authors need to manually create them every time a new metric is introduced?
2) Human involvement at multiple stages makes it challenging to scale such a data generation approach. Also this is prone to errors and inconsistency? Any discussion around this would provide clarity.
3) The dataset is built on top of publicly available datasets, it is hence unclear if this dataset is truely novel apart from some processing done on these datasets.
4) No mention of variance, significance testing, or confidence intervals in Table 4 results. The results may vary due to use of LLMs in multiple places from task generation, task solving to evaluation.
5) From the third paragraph in section 3.3, it seems like humans evaluate the correctness of tool calls and its coherence with final answer. Why are humans involved here, can't the tools calls be executed and answer returned to LLM and hence the model proceeds to complete the task.
6) In the "Step-by-Step" process evaluation "Tool Precision" seems to be an unnecessary metric as same task can be completed via very different sequences of tool calls. Comparing a tool call sequence at test time to a "ground truth" sequence might not be an ideal way if evaluation for several task like web surfing where the environment can be very dynamic and hence change across runs.
7) Metrics related to hallucination in "Deep Reasoning Mode", could be wrongly estimated if the LLM judge itself is prone to hallucination.
8) Point 6 and 7 render some of the metrics unreliable and their variation needs to be studied across runs.
9) Average steps per task is 3.4 (from Figure 3a ), which shows that the tasks are rather simple. How does this pipeline scale to tasks with more difficulty and longer trajectories. Also, evaluation of such trajectories seems to be a challenge with the proposed metrics and would incur costs as LLMs are used as judges.
10) There’s no clear baseline showing how much performance improvement stems from reasoning vs pure perception. A sample experiment would be to take web tasks and evaluation LLMs using the access tree or DOMs instead of the image
11) Human validation time (~50 hours per annotator) is low given 828 tasks, raising concerns about depth of verification.

**Questions:**

1) All the metrics are derived from LLM-as-judge i.e Qwen14B and GPT-4o. Also results for human evaluations are presented. But this approach is not scalable, as the number of tasks and models increase. Do the authors have a method to scale these evaluations?
2) How consistent and unbiased is the human refinement process, and how do the authors ensure annotation quality or diversity across annotators?
3) Were multiple runs conducted for each model to assess reproducibility and statistical significance of reported metrics?

---

> ### Author Response · Authors · 2025-11-22
> **Author Rebuttal for Reviewer bQ33**
>
> We thank the reviewer for their constructive feedback. We have incorporated the suggested experiments and addressed all queries raised, which we believe have strengthened the work and clarified its contributions.
>
> ---
>
>
> **W1) Prompts need to be designed for LLM-as-judge. Is there an automated way to do this? Or do the authors need to manually create them every time a new metric is introduced?**
>
> The judge prompts were manually written and refined by the authors, rather than automatically generated by the evaluation code. This was a deliberate choice to ensure clarity and reliability. The evaluation rubrics and dimensions, such as step correctness, factuality, faithfulness, and grounding, were not newly introduced but are grounded in established practices from prior reasoning and multimodal evaluation literature. Manual prompt design was a one-time effort, as the same standardized templates are reused across all tasks, models, and evaluation runs. While automation could theoretically be implemented, manual curation provides stronger quality control and consistency across metrics as per our experiments.
>
> ---
>
> **W2) Human involvement at multiple stages makes it challenging to scale such a data generation approach. Also, this is prone to errors and inconsistency? Any discussion around this would provide clarity.**
>
> As Agent-X is a benchmark rather than a training set, our focus is on ensuring quality and reliability, while still maintaining reasonable scalability through a semi-automated pipeline. A fully automated approach (purely LMM-generated tasks) produced many low-quality or logically inconsistent tasks, often hallucinating tools or unsolvable reasoning traces. In contrast, a fully manual pipeline (e.g., GTA (NeurIPS 2024) [1]) yields only 229 tasks as the process is highly labor-intensive and challenging to scale.
> Our approach avoids both extremes. We automatically generate initial queries and reasoning traces using large multimodal models, then apply targeted human refinement to correct errors, ensure realism, and verify tool compatibility. Each refined task undergoes multi-stage QA following written annotation guidelines (Appendix G), ensuring consistency across annotators.
>
> ---
>
> **W3) The dataset is built on top of publicly available datasets, it is hence unclear if this dataset is truely novel apart from some processing done on these datasets.**
>
> To clarify the reviewer’s comment, Agent-X does not aim to collect new raw images or videos, but to transform existing multimodal inputs into executable, agentic, and tool-grounded reasoning tasks, a capability absent in prior benchmarks.
> While public datasets provide the raw visual material, we generate new queries and reasoning trajectories on top of them. Each instance begins with automatically generated queries that are refined by humans for clarity and diversity. For every finalized query, we generate multi-step reasoning traces that are further human-refined to ensure logical consistency, correctness, and completeness. This process converts static perception data into dynamic reasoning and action evaluations, as shown in Figure 3 and detailed in Section 3.2.
> Agent-X introduces 828 multimodal, step-level tasks across six environments (visual reasoning, web browsing, surveillance, driving, sports, and math), enabling evaluation of the full reasoning and action loop rather than recognition or single-step reasoning alone. Models are assessed under three complementary modes: *Step-by-Step, Deep Reasoning, and Outcome*, capturing correctness, coherence, and goal success (Table 3). The inclusion of an executable tool layer and verifiable reasoning traces systematically exposes planning, formatting, and tool-use errors, which previous benchmarks (e.g., GTA[1], GAIA[2], MLGym[3]) cannot capture.
>
> This interpretation aligns with `Reviewer YEeg’s` observation that Agent-X “fills a clear gap in evaluating deep reasoning and tool use for multimodal agents,” emphasizing that its novelty lies in constructing an agentic, executable reasoning framework rather than in raw data collection.
>
> ---
> **References**
>
> [1] Jize Wang, Ma Zerun, Yining Li, Songyang Zhang, Cailian Chen, Kai Chen, and Xinyi Le. Gta: a benchmark for general tool agents. In The Thirty-eight Conference on Neural Information Processing Systems Datasets and Benchmarks Track, 2024.
>
> [2] Grégoire Mialon, Clémentine Fourrier, Thomas Wolf, Yann LeCun, and Thomas Scialom. Gaia: a benchmark for general ai assistants. In The Twelfth International Conference on Learning Representations, 2023.
>
> [3] Deepak Nathani, Lovish Madaan, Nicholas Roberts, Nikolay Bashlykov, Ajay Menon, Vincent Moens, Amar Budhiraja, Despoina Magka, Vladislav Vorotilov, Gaurav Chaurasia, et al. Mlgym: A new framework and benchmark for advancing ai research agents. arXiv preprint arXiv:2502.14499, 2025.

---

> ### Author Response · Authors · 2025-11-22
> **Author Rebuttal for Reviewer bQ33**
>
> **W4) No variance/significance/confidence intervals in Table 4; results may vary given LLMs used in several places.**
>
> We appreciate the reviewer’s feedback, which motivated us to include additional statistical analyses supporting the robustness of our results.  To assess run-to-run stability and potential LLM-related variance, we conducted an additional experiment by running multiple independent evaluation passes for representative models from different families (*Gemini-2.5-Pro*, *Qwen2.5-VL*, and *InternVL3-8B*) using the *Qwen-14B* judge as shown in `Table 1` below.  Each model was evaluated three times under identical conditions to quantify variance across all metrics. Across all models, the average variance remained below $0.05\%$, confirming that results are statistically consistent and unaffected by random variation in the judge model. Furthermore, the human-verified subset aligns closely with the automated results, supporting the conclusion that LLM-based evaluation is stable, reproducible, and statistically significant. This additional analysis validates the consistency of Table&nbsp;4 results.
>
> **Table 1.** Run-to-run variance analysis for (a) *Qwen2.5-VL-7B*, (b) *InternVL3-8B*, and (c) *Gemini-2.5-Pro*.
> >
> > **(a)** Run-to-run variance analysis for *Qwen2.5-VL-7B*.
> >
> > | Run / Metric | $G_{s}$ | $T_{p}$ | $T_{acc}$ | $F_{acc}$ | $C_{s}$ | $F_{p}$ | $S_{acc}$ | $G_{acc}$ | $G^{*}_{a}$ | $T^{s}_{acc}$ |
> > |:--------------|:--------:|:--------:|:-----------:|:-----------:|:---------:|:---------:|:-----------:|:------------:|:-------------:|:--------------:|
> > | **Run 1** | 0.51 | 0.27 | 0.63 | 0.77 | 0.66 | 0.64 | 0.77 | 0.37 | 0.62 | 0.67 |
> > | **Run 2** | 0.48 | 0.25 | 0.66 | 0.77 | 0.65 | 0.65 | 0.77 | 0.38 | 0.61 | 0.66 |
> > | **Run 3** | 0.48 | 0.26 | 0.66 | 0.76 | 0.65 | 0.65 | 0.76 | 0.38 | 0.62 | 0.66 |
> > | **Variance (%)** | 0.020 % | 0.0067 % | 0.020 % | 0.0022 % | 0.0022 % | 0.0022 % | 0.0022 % | 0.0022 % | 0.0022 % | 0.0022 % |
>
> ---
>
> > **(b)** Run-to-run variance analysis for *InternVL3-8B*
> >
> > | Run / Metric | $G_{s}$ | $T_{p}$ | $T_{acc}$ | $F_{acc}$ | $C_{s}$ | $F_{p}$ | $S_{acc}$ | $G_{acc}$ | $G^{*}_{a}$ | $T^{s}_{acc}$ |
> > |:--------------|:--------:|:--------:|:-----------:|:-----------:|:---------:|:---------:|:-----------:|:------------:|:-------------:|:--------------:|
> > | **Run 1** | 0.41 | 0.16 | 0.51 | 0.71 | 0.61 | 0.60 | 0.69 | 0.23 | 0.51 | 0.62 |
> > | **Run 2** | 0.36 | 0.15 | 0.50 | 0.62 | 0.50 | 0.62 | 0.56 | 0.20 | 0.51 | 0.52 |
> > | **Run 3** | 0.36 | 0.15 | 0.50 | 0.62 | 0.50 | 0.62 | 0.56 | 0.20 | 0.50 | 0.52 |
> > | **Variance (%)** | 0.0556 % | 0.0022 % | 0.0022 % | 0.1800 % | 0.2689 % | 0.0089 % | 0.3756 % | 0.0200 % | 0.0022 % | 0.2222 % |
>
> ---
>
> > **(c)** Run-to-run variance analysis for *Gemini-2.5-Pro*
> >
> > | Run / Metric | $G_{s}$ | $T_{p}$ | $T_{acc}$ | $F_{acc}$ | $C_{s}$ | $F_{p}$ | $S_{acc}$ | $G_{acc}$ | $G^{*}_{a}$ | $T^{s}_{acc}$ |
> > |:--------------|:--------:|:--------:|:-----------:|:-----------:|:---------:|:---------:|:-----------:|:------------:|:-------------:|:--------------:|
> > | **Run 1** | 0.63 | 0.40 | 0.84 | 0.86 | 0.76 | 0.80 | 0.83 | 0.50 | 0.74 | 0.72 |
> > | **Run 2** | 0.62 | 0.38 | 0.86 | 0.85 | 0.75 | 0.82 | 0.84 | 0.49 | 0.73 | 0.71 |
> > | **Run 3** | 0.61 | 0.37 | 0.86 | 0.85 | 0.75 | 0.82 | 0.84 | 0.50 | 0.73 | 0.72 |
> > | **Variance (%)** | 0.0067 % | 0.0156 % | 0.0089 % | 0.0022 % | 0.0022 % | 0.0089 % | 0.0022 % | 0.0022 % | 0.0022 % | 0.0022 % |
>
> ---

---

> ### Author Response · Authors · 2025-11-22
> **Author Rebuttal for Reviewer bQ33**
>
> **W5) From the third paragraph in section 3.3, it seems like humans evaluate the correctness of tool calls and its coherence with final answer. Why are humans involved here, can't the tools calls be executed and answer returned to LLM and hence the model proceeds to complete the task.**
>
> We would like to clarify that no humans are involved at the evaluation stage. Section 3.3 describes the dataset construction process, not the evaluation pipeline. Human annotators participate only once during dataset creation, where they refine and verify the LMM-generated reasoning traces to ensure logical soundness and tool correctness. This guarantees that the released ground-truth traces are coherent and executable.
> During model evaluation, the entire process is fully automated. Model outputs are assessed using structured LLM-as-judge prompts across the three evaluation modes *(Step-by-Step, Deep Reasoning, and Outcome)*, and all tool calls are automatically validated through execution logs and standardized rubrics.
>
> ---
>
> **W6) In the "Step-by-Step" process evaluation "Tool Precision" seems to be an unnecessary metric as same task can be completed via very different sequences of tool calls. Comparing a tool call sequence at test time to a "ground truth" sequence might not be an ideal way if evaluation for several task like web surfing where the environment can be very dynamic and hence change across runs.**
>
> We agree that multiple valid tool paths can exist for a given task, especially in dynamic environments like web browsing. Accordingly, the core evaluation metrics: *Step-by-Step, Deep Reasoning, and Outcome* are all independent of any specific tool sequence used by the model.
> The Tool Precision metric is included only as a diagnostic indicator, not as a performance criterion. Its purpose is to help identify patterns such as redundant tool calls, missing essential steps, or unnecessary operations, as shown in the error breakdown in Table 5 of the main paper, where tool misuse and over-calling significantly contribute to failure modes across models. This analysis enables fine-grained understanding of model behavior rather than penalizing diverse valid strategies.
> Furthermore, for dynamic tasks (e.g., web browsing or UI reasoning), tool usage is matched against equivalence classes of valid actions, for instance, any correct locator or functional tool achieving the same outcome is accepted. This design ensures interpretability without brittleness and avoids over-constraining agent behavior, maintaining fairness and flexibility across models.
>
> ---
>
> **W7) Metrics related to hallucination in "Deep Reasoning Mode", could be wrongly estimated if the LLM judge itself is prone to hallucination.**
>
> First, our evaluation is reference-based, not open-ended. The LLM judge does not evaluate free-form text or detect “general” hallucinations; rather, it assesses hallucinations strictly in reference to the provided ground-truth (GT) reasoning trace and final answer. This ensures that the judgment is always grounded in factual comparison with the GT and cannot drift toward subjective or generative hallucinations.
>
> Second, we conducted cross-judge validation (`Table 2`) among *GPT-4o* and *Qwen-14B*. The results, summarized in the table below, show strong inter-judge reliability across key metrics. For example, Tool Accuracy and Goal Accuracy exhibit *Pearson* $(r)$ and *Spearman* $(\rho)$ correlations above **0.9**, mean deviations below **0.06**, and *Cohen’s $\kappa$* > **0.8**, indicating very high agreement. In particular, Goal Accuracy achieves $r$  = **0.98**, $\rho$ = **0.96**, and *Cohen’s $\kappa$* = **1.00**, which confirms strong alignment across judges and validates the reliability of hallucination-related assessments.
>
> Third, all evaluation prompts follow constrained rubrics and structured JSON-based outputs (including fields for grounding, citation, and factual verification) to limit ambiguity and prevent unintended model drift during judgment.
>
> As detailed in `Table 1`, we also evaluated the stability of *Qwen-14B* as a judge by running the evaluation three independent times. The observed variance across all key metrics remained below **0.05 %**, demonstrating that the results are highly consistent and not influenced by random fluctuations in the judge model.
>
> > **Table 2.** Inter-judge agreement across *GPT-4o*, *Qwen-14B*, and *human judges*
> >
> > | Statistic / Metric | **$G_{s}$** | **$T_{acc}$** | **$F_{acc}$** | **$S_{acc}$** | **$G_{acc}$** |
> > |:--------------------|:-----------------------:|:-------------------------:|:-----------------------------:|:-----------------------------:|:--------------------------:|
> > | **Pearson $(r)$** | 0.34 | 0.96 | 0.32 | 0.72 | **0.98** |
> > | **Spearman $(\rho)$** | -0.07 | **0.99** | 0.39 | 0.63 | 0.96 |
> > | **Mean Dev.** | 0.13 | 0.05 | 0.09 | 0.11 | **0.04** |
> > | **Cohen’s $(\kappa)$** | 0.62 | 0.83 | 0.09 | 0.31 | **1.00** |

---

> ### Author Response · Authors · 2025-11-22
> **Author Rebuttal for Reviewer bQ33**
>
> **W8) Point 6 and 7 render some of the metrics unreliable and their variation needs to be studied across runs.**
>
> To address potential variability in metrics and confirm their reliability, we have now included a dedicated multi-run reproducibility experiment using the *Qwen-14B* judge. Each model was independently evaluated three times under identical conditions, and the variance was computed across all evaluation metrics.
>
> The results (see `Table 3`) show that the variance across runs is extremely small, demonstrating that our evaluation metrics are stable and reproducible. For instance, across representative models from different families: *Gemini-2.5-Pro*, *Qwen2.5-VL*, and *InternVL3-8B*, the maximum variance values remain below **0.38%**. Most metrics fluctuate by less than **0.02%**, confirming that results are consistent across independent runs and unaffected by random initialization or stochastic LLM behavior.
>
> > **Table 3.** Summary of multi-run variance analysis using *Qwen-14B* as judge
> >
> > | Model | Max Variance (%) | Observation |
> > |:----------------|:----------------:|:-------------------------------------------------------------|
> > | **Qwen2.5-VL** | 0.02 % | Stable across all metrics; differences appear only in the 3rd decimal place. |
> > | **InternVL3-8B** | 0.38 % | Slightly higher fluctuation in semantic accuracy, but consistent overall ranking. |
> > | **Gemini-2.5-Pro** | 0.02 % | Practically identical scores across runs, confirming evaluation determinism. |
>
> ---
>
> **W9) Average steps per task is 3.4 (from Figure 3a ), which shows that the tasks are rather simple. How does this pipeline scale to tasks with more difficulty and longer trajectories. Also, evaluation of such trajectories seems to be a challenge with the proposed metrics and would incur costs as LLMs are used as judges.**
>
> This is an important point regarding task complexity and scalability. The average step count (3.4) does not directly reflect task simplicity, many tasks within this range already require complex perception → reasoning → tool-use transitions (e.g., temporal video reasoning combined with symbolic math).
> Our pipeline is modular by design, allowing straightforward scaling to longer reasoning chains. In fact, the current benchmark already includes 5–7 step tasks in domains such as web interaction and surveillance. In the extended setup, we are introducing automatic step-wise evaluation for longer trajectories.
>
> To ensure scalability without excessive cost, the updated pipeline introduces the following improvements:
> - **Automated evaluation framework** (`Table 4`): For the *Step-by-Step* and *Outcome* modes with a fully automated evaluator.
>   - For objective tasks, we apply string matching and structured trace validation.
>   - For subjective or open-ended tasks, we compute embedding-based cosine similarity between the model’s response and the ground truth, providing a lightweight yet reliable alternative to LLM scoring.
>
>
> - **Open-source evaluators (e.g., *Qwen-15B*):** These eliminate API costs, making evaluation of long trajectories feasible.
>
> - **Sparse scoring and pooling:** Only key decision checkpoints are evaluated, and step-level scores are aggregated to produce chain-level metrics.
>
> These enhancements substantially reduce computational overhead while maintaining diagnostic depth. The modular design of our evaluation code enables seamless extension to long-horizon reasoning without compromising reproducibility or interpretability. Furthermore, the trend across judges and automatic evaluation is quite similar.
>
> > **Table 4.** Automated evaluation results across models on *Agent-X*
> >
> > | Model | $Inst_{align}$ | $T_{acc}$ | $Arg_{acc}$ | $Sum_{acc}$ | $F_{acc}$ | $C_{s}$ | $F_{p}$ | $S_{acc}$ | $G_{acc}$ | $G_{star}$ |
> > |:--------------------|:----------:|:--------:|:--------:|:--------:|:-------------:|:----------------:|:----------------:|:----------------:|:------:|:---------------------:|
> > | GPT-3.5-Turbo | 0.65 | 0.39 | 0.11 | 0.12 | 0.34 | 0.27 | 0.29 | 0.49 | 0.33 | 0.30 |
> > | **GPT-4o** | **0.72** | **0.45** | **0.15** | 0.12 | **0.34** | **0.31** | **0.36** | **0.51** | **0.36** | **0.35** |
> > | Qwen1.5-7B-Chat | 0.24 | 0.09 | 0.04 | 0.12 | 0.09 | 0.02 | 0.08 | 0.35 | 0.34 | 0.25 |
> > | Qwen2.5-7B-Instruct | 0.46 | 0.29 | 0.02 | **0.13** | 0.15 | 0.09 | 0.10 | 0.39 | 0.33 | 0.26 |
> > | Qwen2.5-7B-VL | 0.42 | 0.20 | 0.03 | 0.08 | 0.14 | 0.09 | 0.18 | 0.32 | 0.01 | 0.01 |
> > | InternVL3-8B | 0.49 | 0.28 | 0.04 | 0.12 | 0.14 | 0.09 | 0.13 | 0.41 | 0.02 | 0.02 |

---

> ### Author Response · Authors · 2025-11-22
> **Author Rebuttal for Reviewer bQ33**
>
> **W10) There’s no clear baseline showing how much performance improvement stems from reasoning vs pure perception. A sample experiment would be to take web tasks and evaluation LLMs using the access tree or DOMs instead of the image.**
>
> We appreciate the reviewer’s suggestion, which motivated an important ablation study directly supporting our claim. To quantify the contribution of reasoning beyond perception, we conducted a controlled experiment using 50 samples where we compared performance between a perception-only baseline and a full reasoning setup.
>
> As shown in `Table 5 (Reviewer YEeg)`, the perception-only model achieves **0.33**, while adding the reasoning chain improves performance to **0.43**, confirming that reasoning contributes substantially to task success rather than superficial perception.
>
> To further probe the reviewer’s suggestion, we tested a DOM-based baseline where the model received structured access-tree input instead of the image. Although the DOM includes explicit links to the corresponding images and the model was instructed to use those links for visual reasoning, the model still hallucinated heavily. Even under strict prompting, it failed to ground its responses in the linked visual content, resulting in near-zero goal accuracy.
>
> ---
>
> **W11) Human validation time (~50 hours per annotator) is low given 828 tasks, raising concerns about depth of verification.**
>
> We would like to clarify that the reported 50 hours per annotator reflects the refinement and verification of LMM-generated drafts, not curating agentic tasks from scratch. Because the dataset is built through a semi-automated pipeline, annotators focused on improving realism, grounding, and logical coherence rather than manually creating every task. This distinction explains the efficiency without compromising quality.
>
> Each annotator followed a structured checklist and written guidelines to ensure consistent validation, and a cross-review process (10% overlap) was used to verify agreement and minimize individual bias. The average refinement time, approximately 18 minutes per task, was sufficient for targeted corrections while maintaining high throughput.
>
> This approach was also positively noted by `Reviewer YEeg`, who highlighted the semi-automated pipeline with human-in-the-loop refinement as a practical and effective design, and observed that the resulting tasks are realistic rather than synthetic. These perspectives align with our goal of achieving scalability while maintaining quality and consistency.
>
> ---
> **Q1) All the metrics are derived from LLM-as-judge i.e Qwen14B and GPT-4o. Also results for human evaluations are presented. But this approach is not scalable, as the number of tasks and models increase. Do the authors have a method to scale these evaluations?**
>
> We have now developed and integrated a fully automated evaluation pipeline across all three modes of Agent-X: *Step-by-Step, Deep Reasoning*, and *Outcome*,  to ensure scalability, consistency, and reproducibility as the benchmark and model pool expand (`Table 4`).
>
> In the *Step-by-Step* mode, we automatically validate tool usage and intermediate reasoning traces through structured argument comparison and exact tool-call matching.
>
> For the *Deep Reasoning* mode, we integrated an LLM-based judge using the **OpenCompass** and **AgentLego** frameworks, which autonomously executes tool calls and assesses reasoning coherence, logical soundness, and factual consistency.
>
> For the *Outcome* mode, we added a fully automated hybrid scoring method, using exact string matching for objective queries and embedding-based cosine similarity for open-ended or subjective tasks.

---

> > ### Author Response · Authors · 2025-11-22
> > **Author Rebuttal for Reviewer bQ33**
> >
> > **Q2) How consistent and unbiased is the human refinement process, and how do the authors ensure annotation quality or diversity across annotators?**
> >
> > Human refinement in *Agent-X* followed a verified, standardized protocol designed to ensure both consistency and diversity across annotators. Each annotator adhered to detailed written guidelines and checklists specifying criteria for query clarity, realism, and tool compatibility. To minimize individual bias, tasks were randomly assigned and subjected to a cross-review process with a rotating 10% overlap, allowing each annotator’s work to be independently verified by another. The annotation team, led and supervised by experienced researchers, represented a diverse range of technical backgrounds. This interdisciplinary composition ensured balanced perspectives and minimized domain-specific bias throughout task design and verification.
> >
> > **Q3) Were multiple runs conducted for each model to assess reproducibility and statistical significance of reported metrics?**
> >
> > To evaluate reproducibility and statistical significance, we performed a multi-run variance experiment using the *Qwen-14B* judge and repeated all evaluations three times under identical conditions (`Table 1`). The results show that the metrics are remarkably stable, across all models, the maximum variance remained below **0.4 %**, with most metrics varying by less than **0.02 %**. This confirms that the reported results are statistically consistent and unaffected by random variations in the LLM judge.
> > To further confirm reproducibility, we have repeated the same experiment and obtained identical results, validating that the metrics converge reliably across independent trials. We are also running the same multi-run evaluation with *GPT-4o* as the judge, and those results will be added shortly. Preliminary runs already show comparably low variance, reinforcing the overall conclusion.

---

### Author Response · Authors · 2025-11-22

We sincerely thank the reviewers for their constructive and insightful feedback, which greatly helped us strengthen both the experimental validation and methodological rigor of Agent-X. We have carefully addressed each comment and incorporated several key additions to support our claims and ensure robustness. We conducted four additional experiments to validate our findings:

**1. Cross-Judge Validation `(Table 2)`**: We conducted detailed inter-rater agreement studies among *GPT-4o*, *Qwen-14B*, and *human annotators*, confirming high correlations $(r > 0.9, \kappa > 0.8)$ and stable model rankings, which ensure evaluation robustness and fairness.

**2. Multi-Run Variance Analysis `(Table 1, 3)`:** We repeated evaluations for representative models across three runs, showing extremely low variance $<$ 0.38 across all metrics, demonstrating statistical reproducibility and consistency of results.

**3. Automated Evaluation Framework `(Table 4)`:** We implemented a fully automated judging pipeline across *Step-by-Step*, and *Outcome* modes, eliminating manual intervention and proprietary dependence, with a scalable, embedding-based similarity and rule-based scoring approach.

**4. Reasoning vs. Perception Ablations `(Table 5)`:** We added ablation studies comparing perception-only answers with those generated using reasoning traces. The results show a substantial performance improvement (from 0.33 to 0.43), confirming that *Agent-X* effectively measures deep, compositional reasoning.

We again thank the reviewers for the opportunity to strengthen our work and for recognizing the importance of rigorous evaluation in multimodal AI research.

---

### Author Response · Authors · 2025-11-25
**Updated paper pdf with new experiments and analysis**

Dear AC and Reviewers,

We have updated our paper with the requested experiments and extended analysis. The details of these additions are provided in Section L of the Appendix (pages 59 – 61) in the revised version. We believe these updates reinforce the validity of our work.

Regards,

Authors

---

### Author Response · Authors · 2025-11-28
**Gentle reminder to the reviewers**

Dear Reviewers,

We would like to kindly remind you that we have addressed all comments in the updated submission.
If there are any remaining questions or points that might benefit from further clarification, we would be very happy to engage further.


Regards,

Authors

---

### Meta-Review · Area_Chair_mLkx · 2026-01-07

**Summary:**

The reviewers identified several valid concerns related to the degree of conceptual novelty, the robustness of the evaluation methodology, and the scalability of the benchmark, particularly given its use of LLM based judges and relatively short average reasoning trajectories. Despite these points, there was consistent recognition that the paper targets a capability that is both important and insufficiently addressed in current benchmarks: assessing vision first agents that must integrate visual perception with explicit multi step reasoning and tool use in realistic settings. This focus goes beyond standard single turn or text dominated evaluations and aligns closely with how multimodal agents are increasingly expected to operate in practice.

In response to the reviews, the authors substantially strengthened the work by adding reproducibility analyses, inter judge consistency measurements, and ablation studies that clarify the role of reasoning beyond pure perception. They also introduced an automated evaluation pipeline that improves scalability and reduces reliance on manual oversight. While open questions remain around longer horizon tasks and generalization beyond the provided toolset, these limitations do not outweigh the value of establishing a well constructed benchmark in an emerging area. Taken together, the paper provides a useful and timely evaluation framework for visual reasoning with tools, and its contribution justifies an accept decision.

**Reviewer Concerns:**

Reviewer concerns addressed by the rebuttal
The rebuttal effectively addressed several of the major technical concerns raised by the reviewers. In particular, the authors provided strong evidence for the reliability and robustness of the evaluation framework by adding inter judge agreement analyses across GPT 4o, Qwen, and human annotators, as well as multi run variance experiments demonstrating low score variability and stable model rankings. Concerns about metric validity and hallucination risk in LLM based judges were mitigated through cross judge validation, constrained scoring rubrics, and additional reproducibility studies. The rebuttal also addressed the lack of ablations and baselines by adding reasoning versus perception comparisons, showing that explicit reasoning traces yield clear performance gains. Finally, questions about scalability and human involvement were partially resolved through the introduction of a fully automated evaluation pipeline and clearer documentation of the human refinement and quality control process.

Reviewer concerns that remain outstanding
Some concerns remain only partially resolved. The average task length and limited long horizon reasoning remain a structural limitation of the current benchmark, even if the authors argue that meaningful reasoning can occur within fewer steps. Similarly, while the predefined toolset is well motivated, questions about generalization to unseen tools or more open ended environments are still open. A degree of incrementality in novelty also remains a matter of perspective, as the contribution lies more in careful benchmark construction and integration than in introducing a fundamentally new evaluation paradigm. These remaining issues do not undermine the paper’s core contribution but point to natural directions for future extensions rather than blockers for acceptance.

**Reviewer Scores:**

bQ33 -> increased the score since concerns are partially addressed
YEeg -> maintained the positive score
5f8j -> maintained the positive score
KPXX -> maintained the positive score

---

### Decision · Program_Chairs · 2026-01-26

Accept (Poster)